# On Provable Benefits of Depth in Training Graph Convolutional Networks

**Weilin Cong**
Penn State
wxc272@psu.edu

**Morteza Ramezani**
Penn State
morteza@cse.psu.edu

**Mehrdad Mahdavi**
Penn State
mzm616@psu.edu

## Abstract

Graph Convolutional Networks (GCNs) are known to suffer from performance degradation as the number of layers increases, which is usually attributed to over-smoothing. Despite the apparent consensus, we observe that there exists a discrepancy between the theoretical understanding of over-smoothing and the practical capabilities of GCNs. Specifically, we argue that over-smoothing does not necessarily happen in practice, a deeper model is provably expressive, can converge to global optimum with linear convergence rate, and achieve very high training accuracy as long as properly trained. Despite being capable of achieving high training accuracy, empirical results show that the deeper models generalize poorly on the testing stage and existing theoretical understanding of such behavior remains elusive. To achieve better understanding, we carefully analyze the *generalization capability* of GCNs, and show that the training strategies to achieve high training accuracy significantly deteriorate the *generalization capability* of GCNs. Motivated by these findings, we propose a decoupled structure for GCNs that detaches weight matrices from feature propagation to preserve the expressive power and ensure good generalization performance. We conduct empirical evaluations on various synthetic and real-world datasets to validate the correctness of our theory.

## 1 Introduction

In recent years, Graph Convolutional Networks (GCNs) have achieved state-of-the-art performance in dealing with graph-structured applications, including social networks [28, 21, 51, 12, 44], traffic prediction [10, 45, 34, 31], knowledge graphs [52, 53, 43], drug reaction [14, 16] and recommendation system [2, 60]. Despite the success of GCNs, applying a shallow GCN model that only uses the information of a very limited neighborhood on a large sparse graph has shown to be not effective [23, 6, 20, 9, 46]. As a result, a deeper GCN model would be desirable to reach and aggregate information from farther neighbors. The inefficiency of shallow GCNs is exacerbated even further when the labeled nodes compared to graph size is negligible, as a shallow GCN cannot sufficiently propagate the label information to the entire graph with only a few available labels [35].

Although a deeper GCN is preferred to perceive more graph structure information, unlike traditional deep neural networks, it has been pointed out that deeper GCNs potentially suffer from over-smoothing [35, 41, 26, 4, 58], vanishing/exploding gradients [33], over-squashing [1], and training difficulties [62, 38], which significantly affect the performance of GCNs as the depth increases. Among these, the most widely accepted reason is "over-smoothing", which is referred to as a phenomenon due to applying multiple graph convolutions such that all node embeddings converge to a single subspace (or vector) and leads to indistinguishable node representations.

The conventional wisdom is that adding to the number of layers causes over-smoothing, which impairs the *expressiveness power* of GCNs and consequently leads to a poor *training accuracy*. However, we observe that there exists a discrepancy between theoretical understanding of their inherent capabilities

35th Conference on Neural Information Processing Systems (NeurIPS 2021).

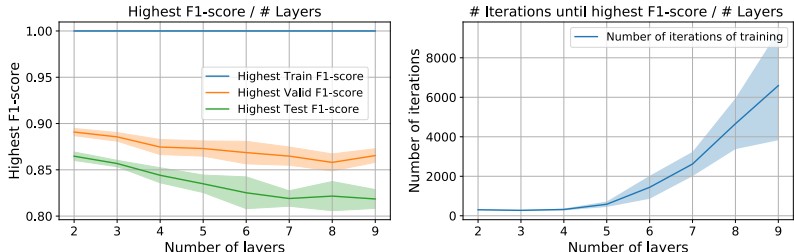

Figure 1: Comparison of F1-score for GCN with different depth on *Cora* dataset, where deeper models can achieve high training accuracy, but complicate the training by requiring more iterations to converge and suffer from poor generalization.

and practical performances. According to the definition of over-smoothing that the node representation becomes indistinguishable as GCNs goes deeper, the classifier has difficulty assigning the correct label for each node if over-smoothing happens. As a result, the training accuracy is expected to be decreasing as the number of layers increases. However, as shown in Figure 1, GCNs are capable of achieving high *training accuracy* regardless of the number of layers. But, as it can be observed, deeper GCNs require more training iterations to reach a high training accuracy, and its *generalization performance* on evaluation set decreases as the number of layers increases. This observation suggests that the performance degradation is likely due to inappropriate training rather than the low expressive power caused by over-smoothing. Otherwise, a low expressiveness model cannot achieve almost perfect training accuracy simply by proper training tricks alone.[1] Indeed, recent years significant advances have been witnessed on tweaking the model architecture to overcome the training difficulties in deeper GCN models and achieve good generalization performance [38, 6, 33, 62].

**Contributions.** Motivated by aforementioned observation, i.e., still achieving high training accuracy when trained properly but poor generalization performance, we aim at answering two fundamental questions in this paper:

Q1: *Does increasing depth really impair the expressiveness power of GCNs?*
In Section 4, we argue that there exists a discrepancy between over-smoothing based theoretical results and the practical capabilities of deep GCN models, demonstrating that over-smoothing is not the key factor that leads to the performance degradation in deeper GCNs. In particular, we mathematically show that over-smoothing [41, 26, 35, 4] is mainly an artifact of theoretical analysis and simplifications made in analysis. Indeed, by characterizing the representational capacity of GCNs via Weisfeiler-Lehman (WL) graph isomorphism test [39, 55], we show that deeper GCN model is at least as expressive as the shallow GCN model, the deeper GCN models can distinguish nodes with a different neighborhood that the shallow GCN cannot distinguish, as long as the GCNs are properly trained. Besides, we theoretically show that more training iterations is sufficient (but *not necessary* due to the assumptions made in our theoretical analysis) for a deeper model to achieve the same training error as the shallow ones, which further suggests the poor training error in deep GCN training is most likely due to inappropriate training.

Q2: *If expressive, why then deep GCNs generalize poorly?*
In Section 5, in order to understand the performance degradation phenomenon in deep GCNs during the evaluation phase, we give a novel generalization analysis on GCNs and its variants (e.g., ResGCN, APPNP, and GCNII) under the semi-supervised setting for the node classification task. We show that the generalization gap of GCNs is governed by the number of training iterations, largest node degree, the largest singular value of weight matrices, and the number of layers. In particular, our result suggests that a deeper GCN model requires more iterations of training and optimization tricks to converge (e.g., adding skip-connections), which leads to a poor generalization. More interestingly, our generalization analysis shows that most of the so-called methods to solve over-smoothing [47, 61, 6, 29] can greatly improve the generalization ability of the model, therefore results in a deeper model.

---

[1]The results in Figure 1 can be reproduced by removing both the dropout and weight decay operation. These two augmentations are designed to improve the generalization ability (i.e., validation/testing accuracy) of the model but might hurt the training accuracy, because of the randomness and also an extra regularization term on the model parameters which are introduced during training. A simple experiment using DGL can be found here.

The aforementioned findings naturally lead to the algorithmic contribution of this paper. In Section 6, we present a novel framework, *Decoupled GCN* (DGCN), that is capable of training deeper GCNs and can significantly improve the generalization performance. The main idea is to isolate the expressive power from generalization ability by decoupling the weight parameters from feature propagation. In Section 7, we conduct experiments on the synthetic and real-world datasets to validate the correctness of the theoretical analysis and the advantages of DGCN over baseline methods.

## 2 Related works

**Expressivity of GCNs.** Existing results on expressive power of GCNs are mixed, [39, 37, 7, 5] argue that deeper model has higher expressive power but [41, 26, 25] have completely opposite result. On the one hand, [39] shows a deeper GCN is as least as powerful as the shallow one in terms of distinguishing non-isomorphic graphs. [37] shows that deep and wide GCNs is Turing universal, however, GCNs lose power when their depth and width are restricted. [7] and [5] measure the expressive power of GCNs via its subgraph counting capability and attribute walks, and both show the expressive power of GCNs grows exponentially with the GCN depth. On the other hand, [25] studies the infinity wide GCNs and shows that the covariance matrix of its outputs converges to a constant matrix at an exponential rate. [41, 26] characterize the expressive power using the distance between node embeddings to a node feature agnostic subspace and show the distance is decreasing as the number of layers increases. Details are deferred to Section 4 and Appendix B. Such contradictory results motivate us to rethink the role of over-smoothing on expressiveness.

**Generalization analysis of GCNs.** In recent years, many papers are working on the generalization of GCNs using uniform stability [50, 63], Neural Tangent Kernel [15], VC-dimension [49], Rademacher complexity [18, 42], algorithm alignment [56, 57] and PAC-Bayesian [36]. Existing works only focus on a specific GCN structure, which cannot be used to understand the impact of GCN structures on its generalization ability. The most closely related to ours is [50], where they analyze the stability of the single-layer GCN model, and show that the stability of GCN depends on the largest absolute eigenvalue of its Laplacian matrix. However, their result is under the inductive learning setting and extending the results to the multi-layer GCNs with different structures is non-trivial.

**Literature with similar observations.** Most recently, several works have similar observations on the over-smoothing issue to ours. [62] argues that the main factors to performance degradation are vanishing gradient, training instability, and over-fitting, rather than over-smoothing, and proposes a node embedding normalization heuristic to alleviate the aforementioned issues. [38] argues that the performance degradation is mainly due to the training difficulty, and proposes a different graph Laplacian formulation, weight parameter initialization, and skip-connections to improve the training difficulty. [59] argues that deep GCNs can learn to overcome the over-smoothing issue during training, and the key factor of performance degradation is over-fitting, and proposes a node embedding normalization method to help deep GCNs overcome the over-smoothing issue. [30] improves the generalization ability of GCNs by using adversarial training and results in a consistent improvement than all GCNs baseline models without adversarial training. All aforementioned literature only gives heuristic explanations based on the empirical results, and do not provide theoretical arguments.

## 3 Preliminaries

**Notations and setup.** We consider the semi-supervised node classification problem, where a self-connected graph $\mathcal{G} = (\mathcal{V}, \mathcal{E})$ with $N = |\mathcal{V}|$ nodes is given in which each node $i \in \mathcal{V}$ is associated with a feature vector $\mathbf{x}_i \in \mathbb{R}^{d_0}$, and only a subset of nodes $\mathcal{V}_{\text{train}} \subset \mathcal{V}$ are labeled, i.e., $y_i \in \{1, \ldots, |\mathcal{C}|\}$ for each $i \in \mathcal{V}_{\text{train}}$ and $\mathcal{C}$ is the set of all candidate classes. Let $\mathbf{A} \in \mathbb{R}^{N \times N}$ denote the adjacency matrix and $\mathbf{D} \in \mathbb{R}^{N \times N}$ denote the corresponding degree matrix with $D_{i,i} = \deg(i)$ and $D_{i,j} = 0$ if $i \neq j$. Then, the propagation matrix (using the Laplacian matrix defined in [28]) is computed as $\mathbf{P} = \mathbf{D}^{-1/2} \mathbf{A} \mathbf{D}^{-1/2}$. Our goal is to learn a GCN model using the node features for all nodes $\{\mathbf{x}_i\}_{i \in \mathcal{V}}$ and node labels for the training set nodes $\{y_i\}_{i \in \mathcal{V}_{\text{train}}}$, and expect it generalizes well on the unlabeled node set $\mathcal{V}_{\text{test}} = \mathcal{V} \setminus \mathcal{V}_{\text{train}}$.

**GCN architectures.** In this paper, we consider the following architectures for training GCNs:

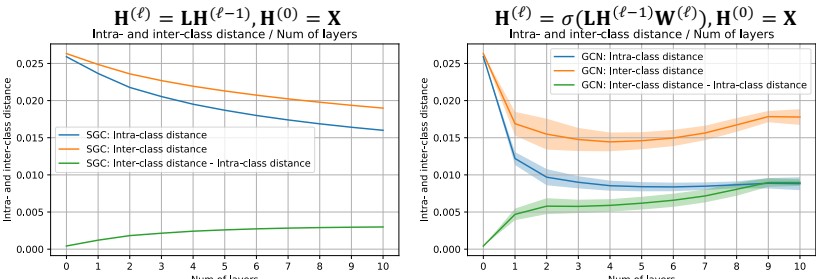

Figure 2: Comparison of intra- and inter-class normalized node embeddings $\mathbf{H}^{(\ell)}/\|\mathbf{H}^{(\ell)}\|_{\mathrm{F}}$ pairwise distance on *Cora* dataset. See Appendix A for more evidences.

- *Vanilla GCN* [28] computes node embeddings by $\mathbf{H}^{(\ell)} = \sigma(\mathbf{P}\mathbf{H}^{(\ell-1)}\mathbf{W}^{(\ell)})$, where $\mathbf{H}^{(\ell)} = \{\mathbf{h}_i^{(\ell)}\}_{i=1}^N$ is the $\ell$th layer node embedding matrix, $\mathbf{h}_i^{(\ell)} \in \mathbb{R}^{d_\ell}$ is the embedding of $i$th node, $\mathbf{W}^{(\ell)} \in \mathbb{R}^{d_\ell \times d_{\ell-1}}$ is the $\ell$th layer weight matrix, and $\sigma(\cdot)$ is the ReLU activation.
- *ResGCN* [33] solves the vanishing gradient issue by adding skip-connections between adjacency layers. More specifically, ResGCN computes node embeddings by $\mathbf{H}^{(\ell)} = \sigma(\mathbf{P}\mathbf{H}^{(\ell-1)}\mathbf{W}^{(\ell)}) + \mathbf{H}^{(\ell-1)}$, where node embeddings of the previous layer is added to the output of the current layer to facilitate the training of deeper GCN models.
- *APPNP* [29] adds skip-connections from the input layer to each hidden layer to preserve the feature information. APPNP computes node embeddings by $\mathbf{H}^{(\ell)} = \alpha_\ell \mathbf{P}\mathbf{H}^{(\ell-1)} + (1-\alpha_\ell)\mathbf{H}^{(0)}\mathbf{W}$, where $\alpha_\ell \in [0,1]$ balances the amount of information preserved at each layer. By decoupling feature transformation and propagation, APPNP can aggregate information from multi-hop neighbors without significantly increasing the computation complexity.
- *GCNII* [6] improves the capacity of APPNP by adding non-linearty and weight matrix at each individual layer. GCNII computes node embeddings by $\mathbf{H}^{(\ell)} = \sigma\big((\alpha_\ell \mathbf{P}\mathbf{H}^{(\ell-1)} + (1-\alpha_\ell)\mathbf{H}^{(0)})\bar{\mathbf{W}}^{(\ell)}\big)$, where $\bar{\mathbf{W}}^{(\ell)} = \beta_\ell \mathbf{W}^{(\ell)} + (1-\beta_\ell)\mathbf{I}$, constant $\alpha_\ell$ same as APPNP, and constant $\beta_\ell \in [0,1]$ restricts the power of $\ell$th layer parameters.

## 4 On true expressiveness and optimization landscape of deep GCNs

**Empirical validation of over-smoothing.** In this section, we aim at answering the following fundamental question: *"Does over-smoothing really cause the performance degradation in deeper GCNs?"* As first defined in [35], *over-smoothing* is referred to as a phenomenon where all node embeddings converge to a single vector after applying multiple graph convolution operations to the node features. However, [35] only considers the graph convolution operation *without* non-linearity and the per-layer weight matrices. To verify whether over-smoothing exists in normal GCNs, we measure the pairwise distance between the normalized node embeddings with varying the model depth.[2] As shown in Figure 2, without the weight matrices and non-linear activation functions, the pairwise distance between node embeddings indeed decreases as the number of layers increases. However, by considering the weight matrices and non-linearity, the pairwise distances are actually increasing after a certain depth which *contradicts* the definition of over-smoothing that node embeddings become indistinguishable when the model becomes deeper. That is, graph convolution makes adjacent node embeddings get closer, then non-linearity and weight matrices help node embeddings preserve distinguishing-ability after convolution.

Most recently, [41, 26] generalize the idea of over-smoothing by taking both the non-linearity and weight matrices into considerations. More specifically, the expressive power of the $\ell$th layer node embeddings $\mathbf{H}^{(\ell)}$ is measured using $d_{\mathcal{M}}(\mathbf{H}^{(\ell)})$, which is defined as the distance of node embeddings to a subspace $\mathcal{M}$ that only has node degree information. Let denote $\lambda_L$ as the second largest eigenvalue of graph Laplacian and $\lambda_W$ as the largest singular value of weight matrices. [41, 26] show that the expressive power $d_{\mathcal{M}}(\mathbf{H}^{(\ell)})$ is bounded by $d_{\mathcal{M}}(\mathbf{H}^{(\ell)}) \leq (\lambda_W \lambda_L)^\ell \cdot d_{\mathcal{M}}(\mathbf{X})$, i.e., the expressive power of node embeddings will be exponentially decreasing or increasing as the number

---

[2]The pairwise distances are computed on the normalized node embeddings $\mathbf{H}^{(\ell)}/\|\mathbf{H}^{(\ell)}\|_{\mathrm{F}}^2$ to eliminate the effect of node embedding norms.

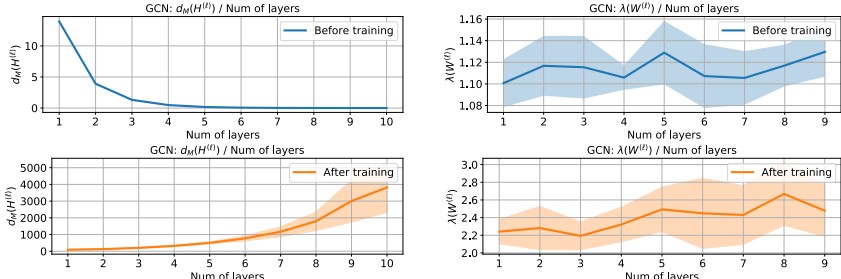

Figure 3: Compare $d_{\mathcal{M}}(\mathbf{H}^{(\ell)})$ and $\lambda_W(\mathbf{W}^{(\ell)})$ on both *trained*- (using 500 gradient update) and *untrained*-GCN models on *Cora* dataset.

of layers increases, depending on whether $\lambda_W \lambda_L < 1$ or $\lambda_W \lambda_L > 1$. They conclude that *deeper GCN exponentially loss expressive power* by assuming $\lambda_W \lambda_L < 1$. However, we argue that this assumption does not always hold. To see this, let suppose $\mathbf{W}^{(\ell)} \in \mathbb{R}^{d_{\ell-1} \times d_\ell}$ is initialized by uniform distribution $\mathcal{N}(0, \sqrt{1/d_{\ell-1}})$. By the Gordon's theorem for Gaussian matrices [11], we know its expected largest singular value is bounded by $\mathbb{E}[\lambda_W] \leq 1 + \sqrt{d_\ell/d_{\ell-1}}$, which is strictly greater than 1 and $\lambda_W$ usually increases during training. The above discussion also holds for other commonly used initialization methods [19, 22]. Furthermore, since most real world graphs are sparse with $\lambda_L$ close to 1, e.g., Cora has $\lambda_L = 0.9964$, Citeseer has $\lambda_L = 0.9987$, and Pubmed has $\lambda_L = 0.9905$, making assumption on $\lambda_W \lambda_L < 1$ is not realistic. As shown in Figure 3, when increasing the number of layers, we observe that the distance $d_{\mathcal{M}}(\mathbf{H}^{(\ell)})$ is decreasing on *untrained*-GCN models, however, the distance $d_{\mathcal{M}}(\mathbf{H}^{(\ell)})$ is increasing on *trained*-GCN models, which contradicts the conclusion in [41]. Due to the space limit, we defer the more empirical evidence to Appendix B. Through these findings, we cast doubt on the power of over-smoothing based analysis to provide a complete picture of why deep GCNs perform badly.

**Definition 1.** *Let $\mathcal{T}_i^L$ denote the $L$-layer computation tree of node $i$, which represents the structured $L$-hop neighbors of node $i$, where the children of any node $j$ in the tree are the nodes in $\mathcal{N}(i)$.*

**Exponentially growing expressiveness without strong assumptions.** Indeed, we argue that *deeper GCNs have stronger expressive power than the shallow GCNs.* To prove this, we employ the connection between WL test[3] [32] and GCNs. Recently, [39] shows that GCNs have the same expressiveness as the WL-test for graph isomorphism if they are appropriately trained, i.e., a properly trained $L$-layer GCN computes different node representations for two nodes if their $L$-layer *computation tree* (Definition 1) have different structure or different features on the corresponding nodes. Since $L$-GCN can encode any different computation tree into different representations, it is natural to characterize the expressiveness of $L$-GCN by the number of computation graph it can encode.

**Theorem 1.** *Suppose $\mathcal{T}^L$ is a computation tree with binary node features and node degree at least $d$. Then, by assuming the computation tree of two nodes are disjoint, the richness (i.e., the number of computation graphs a model can encode) of the output of $L$-GCN defined on $\mathcal{T}^L$ is at least $|L\text{-GCN}(\mathcal{T}^L)| \geq 2(d-1)^{L-1}$.*

The proof is deferred to Appendix C. The above theorem implies that the richness of $L$-GCN grows at least exponentially with respect to the number of layers.

**Comparison of expressiveness metrics.** Although distance-based expressiveness metric [41, 26] is strong than WL-based metric in the sense that node embeddings can be *distinct* but *close to each other*, the distance-based metric requires explicit assumptions on the GCN structures, weight matrices, and graph structures comparing to the WL-based metric, which has been shown that are not likely hold. On the other hand, WL-based metric has been widely used in characterizing the expressive power of GCNs in graph-level task [39, 37, 7, 5]. More details are deferred to related works (Section 2).

---

[3]WL test is a recursive algorithm where the label of a node depends on its own label and neighbors from the previous iterations, i.e., $c_i^{(\ell)} = \text{Hash}(c_i^{(\ell-1)}, \{c_j^{(\ell-1)} | j \in \mathcal{N}(i)\})$, where $\text{Hash}(\cdot)$ bijectively maps a set of values to a unique value that has not been used in the previous iterations. After $L$-iterations, the WL test will assign two nodes with a different label if the $L$-hop neighborhood of two nodes are non-isomorphic.

Although expressive, it is still unclear why the deeper GCN requires more training iterations to achieve small training error and reach the properly trained status. To understand this, we show in Theorem 2 that under assumptions on the width of the final layer, the deeper GCN can converge to its global optimal with linear convergence rate. More specifically, the theorem claims that if the dimension of the last layer of GCN $d_L$ is larger than the number of data $N$[4], then we can guarantee the loss $\mathcal{L}(\boldsymbol{\theta}_T) \leq \varepsilon$ after $T = \mathcal{O}(1/\varepsilon)$ iterations of the gradient updates. Besides, more training iterations is sufficient (but not necessary due to the assumptions made in our theoretical analysis) for a deeper model to achieve the same training error as the shallow ones.

**Theorem 2.** *Let $\boldsymbol{\theta}_t = \{\mathbf{W}_t^{(\ell)} \in \mathbb{R}^{d_{\ell-1} \times d_\ell}\}_{\ell=1}^{L+1}$ be the model parameter at the $t$-th iteration and using square loss $\mathcal{L}(\boldsymbol{\theta}) = \frac{1}{2}\|\mathbf{H}^{(L)}\mathbf{W}^{(L+1)} - \mathbf{Y}\|_{\mathrm{F}}^2$, $\mathbf{H}^{(\ell)} = \sigma(\mathbf{P}\mathbf{H}^{(\ell-1)}\mathbf{W}^{(\ell)})$ as objective function. Then, under the condition that $d_L \geq N$ we can obtain $\mathcal{L}(\boldsymbol{\theta}_T) \leq \epsilon$ if $T \geq C(L)\log(\mathcal{L}(\boldsymbol{\theta}_0)/\epsilon)$, where $\epsilon$ is the desired error and $C(L)$ is a function of GCN depth $L$ that grows as GCN becomes deeper.*

A formal statement of Theorem 2 and its proof are deferred to Appendix D. Besides, gradient stability also provides an alternative way of empirically understanding why deeper GCN requires more iterations: deeper neural networks are prone to exploding/vanishing gradient, which results in a very noisy gradient and requires small learning rate to stabilize the training. This issue can be significantly alleviated by adding skip-connections (Appendix E.5). When training with adaptive learning rate mechanisms, such as Adam [27][5], noisy gradient will result in a much smaller update on current model compared to a stabilized gradient, therefore more training iterations are required.

## 5 A different view from generalization

In the previous section, we provided evidence that a well-trained deep GCN is at least as powerful as a shallow one. However, it is still unclear why a deeper GCN has worse performance than a shallow GCN during the evaluation phase. To answer this question, we provide a different view by analyzing the impact of GCN structures on the generalization.

**Transductive uniform stability.** In the following, we study the generalization ability of GCNs via *transductive uniform stability* [17], where the generalization gap is defined as the difference between the training and testing errors for the random partition of a full dataset into training and testing sets. Transductive uniform stability is defined under the notation that the output of a classifier does not change much if the input is perturbed a bit, which is an extension of uniform stability [3] from the inductive to the transductive setting. The previous analysis on the uniform stability of GCNs [50] only shows the result of GCN with one graph convolutional layer under inductive learning setting, which cannot explain the effect of depth, model structure, and training data size on the generalization, and its extension to multi-layer GCNs and other GCN structures are non-trivial.

**Problem setup.** Let $m = |\mathcal{V}_{\text{train}}|$ and $u = |\mathcal{V}_{\text{test}}|$ denote the training and test dataset sizes, respectively. Under the transductive learning setting, we start with a fixed set of points $X_{m+u} = \{x_1, \ldots, x_{m+u}\}$. For notational convenience, we assume $X_m$ are the first $m$ data points and $X_u$ are the last $u$ data points of $X_{m+u}$. We randomly select a subset $X_m \subset X_{m+u}$ uniformly at random and reveal the labels $Y_m$ for the selected subset for training, but the labels for the remaining $u$ data points $Y_u = Y_{m+u} \setminus Y_m$ are not available during the training phase. Let $S_m = ((x_1, y_1), \ldots, (x_m, y_m))$ denotes the labeled set and $X_u = (x_{m+1}, \ldots, x_{m+u})$ denotes the unlabeled set. Our goal is to learn a model to label the remaining unlabeled set as accurately as possible.

For the analysis purpose, we assume a binary classifier is applied to the final layer node representation $f(\mathbf{h}_i^{(L)}) = \tilde{\sigma}(\mathbf{v}^\top \mathbf{h}_i^{(L)})$ with $\tilde{\sigma}(\cdot)$ denotes the sigmoid function. We predict $\hat{y}_i = 1$ if $f(\mathbf{h}_i^{(L)}) > 1/2$ and $\hat{y}_i = 0$ otherwise, with ground truth label $y_i \in \{0, 1\}$. Let denote the perturbed dataset as $S_m^{ij} \triangleq (S_m \setminus \{(x_i, y_i)\}) \cup \{(x_j, y_j)\}$ and $X_u^{ij} \triangleq (X_u \setminus \{x_j\}) \cup \{x_i\}$, which is obtained by replacing the $i$th example in training set $S_m$ with the $j$th example from the testing set $X_u$. Let $\boldsymbol{\theta}$ and $\boldsymbol{\theta}^{ij}$ denote the weight parameters trained on the original dataset $(S_m, X_u)$ and the perturbed dataset $(S_m^{ij}, X_u^{ij})$,

---

[4]This type of over-parameterization assumptions are required and commonly used in other neural network convergence analysis to guarantee that the model parameters do not change significantly during training.

[5]In the Adam optimizer, the contribution of bias correction of moments varies exponentially over epochs completed. Although the learning rate hyper-parameter is a constant, the contribution of gradients to updated weight varies over epochs, hence adaptive. Please refer to the Chapter 8.5 of [48] for more details.

respectively. Then, we say transductive learner $f$ is $\epsilon$-uniformly stable if the outputs change less than $\epsilon$ when we exchange two examples from the training set and testing set, i.e., for any $S_m \subset S_{m+u}$ and any $i, j \in [m+u]$ it holds that $\sup_{i \in [m+u]} |f(\mathbf{h}_i^{(L)}) - \tilde{f}(\tilde{\mathbf{h}}_i^{(L)}))| \leq \epsilon$, where $f(\mathbf{h}_i^{(L)})$ and $\tilde{f}(\tilde{\mathbf{h}}_i^{(L)})$ denote the prediction of node $i$ using parameters $\boldsymbol{\theta}$ and $\boldsymbol{\theta}^{ij}$ respectively.

To define testing error and training error in our setting, let us introduce the difference in probability between the correct and the incorrect label as $p(z, y) \triangleq y(2z-1) + (1-y)(1-2z)$ with $p(z, y) \leq 0$ if there exists a classification error. Let denote the $\gamma$-margin loss as $\Phi_\gamma(x) = \min(1, \max(0, 1 - x/\gamma))$. Then, the testing error is defined as $\mathcal{R}_u(f) = \frac{1}{u} \sum_{i=m+1}^{m+u} \mathbf{1}\{p(f(\mathbf{h}_i^{(L)}), y_i) \leq 0\}$ and the training loss is defined as $\mathcal{R}_m^\gamma(f) = \frac{1}{m} \sum_{i=1}^{m} \Phi_\gamma(-p(f(\mathbf{h}_i^{(L)}), y_i))$. [6]

**Theorem 3** (Transductive uniform stability bound [17]). *Let $f$ be a $\epsilon$-uniformly stable transductive learner and $\gamma, \delta > 0$, and define $Q = mu/(m+u)$. Then, with probability at least $1-\delta$ over all training and testing partitions, we have $\mathcal{R}_u(f) \leq \mathcal{R}_m^\gamma(f) + \frac{2}{\gamma}\mathcal{O}\left(\epsilon\sqrt{Q\ln(\delta^{-1})}\right) + \mathcal{O}\left(\frac{\ln(\delta^{-1})}{\sqrt{Q}}\right)$.*

Recall that as we discussed in Section 4, deeper GCN is provable more expressive and can achieve very low training error $\mathcal{R}_m^\gamma(f)$ if properly training. Then, if the dataset size is sufficiently large, the testing error $\mathcal{R}_u(f)$ will be dominated by the generalization gap, which is mainly controlled by uniformly stable constant $\epsilon$. In the following, we explore the impact of GCN structures on $\epsilon$. Our key idea is to decompose the $\epsilon$ into three terms: the Lipschitz continuous constant $\rho_f$, upper bound on gradient $G_f$, and the smoothness constant $L_f$ of GCNs. Please refer to Lemma 1 for details.

**Lemma 1.** *Suppose function $f(\mathbf{h}^{(L)})$ is $\rho_f$-Lipschitz continuous, $L_f$-smooth, and the gradient of loss w.r.t. the parameter is bounded by $G_f$. After $T$ steps of full-batch gradient descent, we have $\epsilon = \frac{2\eta\rho_f G_f}{m} \sum_{t=1}^{T}(1 + \eta L_f)^{t-1}$.*

The proof is deferred to Appendix K. By using Lemma 1, we can derive constant $\epsilon$ of different model structures by comparing the Lipschitz continuity, smoothness, and gradient scale.

Before proceeding to our result, we make the following standard assumption on the node feature vectors and weight matrices, which are previously used in generalization analysis of GCNs [18, 36].

**Assumption 1.** *We assume the norm of node feature vectors, weight parameters are bounded, i.e., $\|\mathbf{x}_i\|_2 \leq B_x$, $\|\mathbf{W}^{(\ell)}\|_2 \leq B_w$, and $\|\mathbf{v}\|_2 \leq 1$.*

In Theorem 4, we show that the generalization bounds of GCN and its variants are dominated by the following terms: maximum node degree $d$, model depth $L$, training/validation set size $(m, u)$, training iterations $T$, and spectral norm of the weight matrices $B_w$. The larger the aforementioned variables are, the larger the generalization gap is. We defer the formal statements and proofs to Appendices F, G, H, and I.

**Theorem 4** (Informal). *We say model is $\epsilon$-uniformly stable with $\epsilon = \frac{2\eta\rho_f G_f}{m} \sum_{t=1}^{T}(1 + \eta L_f)^{t-1}$ where the result of $\rho_f, G_f, L_f$ are summarized in Table 1, and other related constants as*

$$B_d^\alpha = (1-\alpha)\sum_{\ell=1}^{L}(\alpha\sqrt{d})^{\ell-1} + (\alpha\sqrt{d})^L, \ B_w^\beta = \beta B_w + (1-\beta),$$
$$B_{\ell,d}^{\alpha,\beta} = \max\left\{\beta\left((1-\alpha)L + \alpha\sqrt{d}\right), (1-\alpha)LB_w^\beta + 1\right\}. \tag{1}$$

In the following, we provide intuitions and discussions on the generalization bound of each algorithm:

- Deep **GCN** requires iterations $T$ to achieve small training error. Since the generalization bound increases with $T$, more iterations significantly hurt its generalization power. Note that our results considers both $B_w \leq 1$ and $B_w > 1$, where increasing model depth will not hurt the generalization if $B_w \leq 1$, and the generalization gap becomes sensitive to the model depth if $B_w > 1$. Notice that $B_w > 1$ is more likely to happen during training as we discussed in Section 4.
- **ResGCN** resolves the training difficulties by adding skip-connections between hidden layers. Although it requires less training iterations $T$, adding skip-connections enlarges the dependency

---

[6]Notice that a different loss function is used in convergence analysis (i.e., $\mathcal{L}(\boldsymbol{\theta})$ in Theorem 7) and generalization analysis ($\mathcal{R}_m^\gamma(f)$ in Theorem 4). We do this because current understanding on the convergence of deep neural networks is still mostly limited to the square loss, but the margin loss is a more suitable and widely accepted loss function for generalization analysis.

Table 1: Comparison of uniform stability constant $\epsilon$ of GCN variants, where $\mathcal{O}(\cdot)$ is used to hide constants that shared between all bounds.

| | $\rho_f$ and $G_f$ | $L_f$ | $C_1$ and $C_2$ |
|---|---|---|---|
| $\epsilon_{\text{GCN}}$ | $\mathcal{O}(C_1^L C_2)$ | $\mathcal{O}\big(C_1^L C_2\big((L+2)C_1^L C_2 + 2\big)\big)$ | $C_1 = \max\{1, \sqrt{d}B_w\},\ C_2 = \sqrt{d}(1+B_x)$ |
| $\epsilon_{\text{ResGCN}}$ | $\mathcal{O}(C_1^L C_2)$ | $\mathcal{O}\big(C_1^L C_2\big((L+2)C_1^L C_2 + 2\big)\big)$ | $C_1 = 1 + \sqrt{d}B_w,\ C_2 = \sqrt{d}(1+B_x)$ |
| $\epsilon_{\text{APPNP}}$ | $\mathcal{O}(C_1)$ | $\mathcal{O}\big(C_1(C_1 C_2) + 1\big)$ | $C_1 = B_d^\alpha B_x,\ C_2 = \max\{1, B_w\}$ |
| $\epsilon_{\text{GCNII}}$ | $\mathcal{O}(\beta C_1^L C_2)$ | $\mathcal{O}\big(\alpha\beta C_1^L C_2\big((\alpha\beta L + 2)C_1^L C_2 + 2\beta\big)\big)$ | $C_1 = \max\{1, \alpha\sqrt{d}B_w^\beta\},\ C_2 = \sqrt{d} + B_{\ell,d}^{\alpha,\beta}B_x$ |
| $\epsilon_{\text{DGCN}}$ | $\mathcal{O}(C_1)$ | $\mathcal{O}\big(C_1(C_1 C_2) + 1\big)$ | $C_1 = (\sqrt{d})^L B_x,\ C_2 = \max\{1, B_w\}$ |

on the number of layers $L$ and the spectral norm of weight matrices $B_w$, therefore results in a larger generalization gap and a poor generalization performance..

- **APPNP** alleviates the aforementioned dependency by decoupling the weight parameters and feature propagation. As a result, its generalization gap does not significantly change as $L$ and $B_w$ increase. The optimal $\alpha$ that minimizes the generalization gap can be obtained by finding the $\alpha$ that minimize the term $B_d^\alpha$. Although APPNP can significantly reduce the generalization gap, because a single weight matrix is shared between all layers, its expressive power is not enough for large-scale challenging graph datasets [24].

- To gain expressiveness, **GCNII** proposes to add the weight matrices back and add another hyper-parameter that explicitly controls the dependency on $B_w$. Although GCNII achieves the state-of-the-art performances on several graph datasets, the selection of hyper-parameters is non-trivial compared to APPNP because $\alpha, \beta$ are coupled with $L, B_w$, and $d$. In practice, [6] builds a very deep GCNII by choosing $\beta$ dynamically decreases as the number of layers and different $\alpha$ values for different datasets.

- By property chosen hyper-parameters, we have the following order on the generalization gap given the same training iteration $T$: APPNP $\leq$ GCNII $\leq$ GCN $\leq$ ResGCN, which exactly match our empirical evaluation on the generalization gap in Section 7 and Appendix E.

**Remark 1.** *It is worthy to provide an alternative view of DropEdge [47] and PairNorm [61] algorithms from a generalization perspective. To improve the generalization power of standard GCNs, DropEdge randomly drops edges in the training phase, which leads to a smaller maximum node degree $d_s < d$. PairNorm applies normalization on intermediate node embeddings to ensure that the total pairwise feature distances remain constant across layers, which leads to less dependency on $d$ and $B_w$. However, since deep GCN requires significantly more iterations to achieve low training error than shallow one, the performance of applying DropEdge and PairNorm on GCNs is still degrading as the number of layers increases. Most importantly, our empirical results in Appendix E.3 and E.4 suggest that applying Dropout and PairNorm is hurting the training accuracy (i.e., not alleviating over-smoothing) but reducing the generalization gap.*

## 6 Decoupled GCN

We propose to decouple the expressive power from generalization ability with **D**ecoupled **GCN** (DGCN). The DGCN model can be mathematically formulated as $\mathbf{Z} = \sum_{\ell=1}^{L} \alpha_\ell f^{(\ell)}(\mathbf{X})$ and $f^{(\ell)}(\mathbf{X}) = \mathbf{P}^\ell \mathbf{X}\big(\beta_\ell \mathbf{W}^{(\ell)} + (1-\beta_\ell)\mathbf{I}\big)$, where $\mathbf{W}^{(\ell)}$, $\alpha_\ell$ and $\beta_\ell$ are the learnable weights for $\ell$th layer function $f^{(\ell)}(\mathbf{X})$.[7] The design of DGCN has the following key ingredients:

- (Decoupling) The generalization gap in GCN grows exponentially with the number of layers. To overcome this issue, we propose to decouple the weight matrices from propagation by assigning weight $\mathbf{W}^{(\ell)}$ to each individual layerwise function $f^{(\ell)}(\mathbf{X})$. DGCN can be thought of as an ensemble of multiple SGCs [54] with depth from $1$ to $L$. By doing so, the generalization gap has less dependency on the number of weight matrices, and deep models with large receptive fields can incorporate information of the global graph structure. Please refer to Theorem 5 for the details.

- (Learnable $\alpha_\ell$) After decoupling the weight matrices from feature propagation, the layerwise function $f^{(\ell)}(\mathbf{X})$ with more propagation steps can suffer from less expressive power. Therefore,

---

[7]A similar architecture has been previously used in [8], where they empirically show that relaxing the layer weights to negative values can further boost the performance.

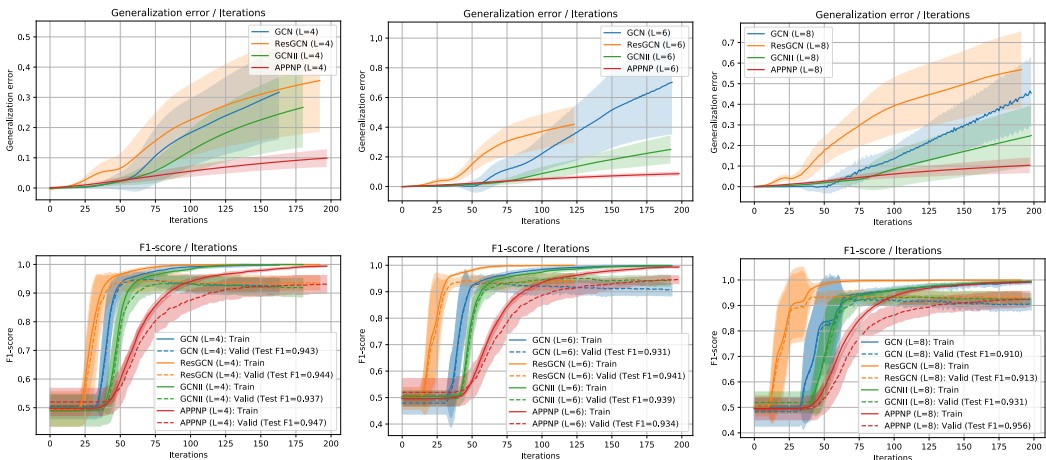

Figure 4: Comparison of generalization error on synthetic dataset. The curve early stopped at the largest training accuracy iteration.

we propose to assign a learnable weight $\alpha_\ell$ for each step of feature propagation. Intuitively, DGCN assigns smaller weight $\alpha_\ell$ to each layerwise function $f^{(\ell)}(\mathbf{X})$ with more propagation steps at the beginning of training. Throughout the training, DGCN gradually adjusts the weight to leverage more useful large receptive field information.

- (Learnable $\beta_\ell$) A learnable weight $\beta_\ell \in [0, 1]$ is assigned to each weight matrix to balance the expressiveness with model complexity, which guarantees a better generalization ability.

**Theorem 5.** *Let suppose $\alpha_\ell$ and $\beta_\ell$ are pre-selected and fixed during training. We say DGCN is $\epsilon_{DGCN}$-uniformly stable with $\epsilon_{DGCN} = \frac{2\eta\rho_f G_f}{m}\sum_{t=1}^{T}(1 + \eta L_f)^{t-1}$ where*

$$\rho_f = G_f = \mathcal{O}\big((\sqrt{d})^L B_x\big), L_f = \mathcal{O}\big((\sqrt{d})^L B_x\big((\sqrt{d})^L B_x \max\{1, B_w\} + 1)\big)\big). \tag{2}$$

The details are deferred to Appendix J, and comparison of bound to other GCN variants are summarized in Table 1. Depending on the automatic selection of $\alpha_\ell, \beta_\ell$, the generalization bound of DGCN is between APPNP and GCN. In the following, we make connection to many GCN structures:

- **Connections to APPNP:** APPNP can be thought of as a variant of DGCN. More specifically, the layerwise weight in APPNP is computed as $\alpha_\ell = \alpha(1 - \alpha)^\ell$ for $\ell < L$ and $\alpha_\ell = (1 - \alpha)^\ell$ for $\ell = L$ given some constant $\alpha \in (0, 1)$, and the weight matrix is shared between all layers. Although DGCN has $L$ weight matrices, its generalization is independent of the number of weight matrices, and thus enjoys a low generalization error with high expressiveness.

- **Connections to GCNII:** GCNII can be regarded as a variant of DGCN. Compared to GCNII, the decoupled propagation of DGCN significantly reduces the dependency of generalization error to the weight matrices. Besides, the learnable weights $\alpha_\ell$ and $\beta_\ell$ allow DGCN to automatically adapt to challenging large-scale datasets without time-consuming hyper-parameter selection.

- **Connections to ResGCN:** By expanding the forward computation of ResGCN, we know that ResGCN can be think of as training an ensemble of GCNs from 1 to $L$ layer, i.e., $\mathbf{H}^{(L)} = \sum_{\ell=1}^{L} \alpha_\ell \sigma(\mathbf{PH}^{(\ell-1)}\mathbf{W}^{(\ell)})$ with $\alpha_\ell = 1$. In other word, ResNet can be regarded as the "*summation* of the model complexity" of $L$-layer. However, DGCN is using $\sum_{\ell=1}^{L}\alpha_\ell = 1$, which can be thought of as a "*weighted average* of model complexity". Therefore, ResGCN is a special case of DGCN with equal weights $\alpha_\ell$ on each layerwise function. With just a simple change on the ResNet structure, our model DGCN is both easy to train and good to generalize.

## 7 Experiments

**Synthetic dataset.** We empirically compare the generalization error of different GCN structures on the synthetic dataset. In particular, we create the synthetic dataset by contextual stochastic block model (CSBM) [13] with two equal-size classes. CSBM is a graph generation algorithm that adds Gaussian random vectors as node features on top of classical SBM. CSBM allows for smooth control over the information ratio between node features and graph topology by a pair of hyper-parameter

$(\mu, \lambda)$, where $\mu$ controls the diversity of the Gaussian distribution and $\lambda$ controls the number of edges between intra- and inter-class nodes. We generate random graphs with 1000 nodes, average node degree as 5, and each node has a Gaussian random vector of dimension 1000 as node features. We chose 75% nodes as training set, 15% of nodes as validation set for hyper-parameter tuning, and the remaining nodes as testing set. We conduct an experiment 20 times by randomly selecting $(\mu, \lambda)$ such that both node feature and graph topology are equally informative.

As shown in Figure 4, we have the following order on the generalization gap given the same training iteration $T$: APPNP $\leq$ GCNII $\leq$ GCN $\leq$ ResGCN, which exactly match the theoretical result in Theorem 4. More specifically, ResGCN has the largest generalization gap due to the skip-connections, APPNP has the smallest generalization gap by removing the weight matrices in each individual layer. GCNII achieves a good balance between GCN and APPNP by balancing the expressive and generalization power. Finally, DGCN enjoys a small generalization error by using the decoupled GCN structure.

**Open graph benchmark dataset.** As pointed out by Hu et al. [24], the traditional commonly-used graph datasets are unable to provide a reliable evaluation due to various factors including dataset size, leakage of node features and no consensus on data splitting. To truly evaluate the expressive and the generalization power of existing methods, we evaluate on the open graph benchmark (OGB) dataset. Experiment setups are based on the default setting for GCN implementation on the leaderboard. We choose the hidden dimension as 128, learning rate as 0.01, dropout ratio as 0.5 for *Arxiv* dataset, and no dropout for *Products* and *Protein* datasets. We train $300/1000/500$ epochs for *Products*, *Proteins*, and *Arxiv* dataset respectively. Due to limited GPU memory, the number of layers is selected as the one with the best performance between 2 to 16 layers for *Arxiv* dataset, 2 to 8 layers for *Protein* dataset, and 2 to 4 for *Products* dataset. We choose $\alpha_\ell$ from $\{0.9, 0.8, 0.5\}$ for APPNP and GCNII, and use $\beta_\ell = 0.5/\ell$ for GCNII, and select the setup with the best validation result for comparison.

As shown in Table 2, DGCN achieves a compatible performance to GCNII[8] without the need of manually tuning the hyper-parameters for all settings, and it significantly outperform APPNP and ResGCN. Due to the space limit, the detailed setups and more results can be found in Appendix E. Notice that generalization bounds are more valuable when comparing two models with same training accuracy (therefore we first show in Section 4 that deeper model can also achieve low training error before our discussion on generalization in Section 5).

In Table 2, because ResGCN has no restriction on the weight matrices, it can achieve lower training error and its test performance is mainly restricted by its generalization error. However, because GCNII and APPNP have restrictions on the weight matrices, their performance is mainly restricted by their training error. A model with small generalization error and no restriction on the weight (e.g., DGCN) is preferred as it has higher potential to reach a better test accuracy by reducing its training error.

Table 2: Comparison of F1-score on OGB dataset.

| % | Products | Proteins | Arvix |
|---|---|---|---|
| **GCN** | $75.39 \pm 0.21$ | $71.66 \pm 0.48$ | $71.56 \pm 0.19$ |
| **ResGCN** | $75.53 \pm 0.12$ | $74.50 \pm 0.41$ | $72.56 \pm 0.31$ |
| **APPNP** | $66.35 \pm 0.10$ | $71.78 \pm 0.29$ | $68.02 \pm 0.55$ |
| **GCNII** | $71.93 \pm 0.35^\dagger$ | $75.60 \pm 0.47$ | $72.57 \pm 0.23^\ddagger$ |
| **DGCN** | $76.09 \pm 0.29$ | $75.45 \pm 0.24$ | $72.63 \pm 0.12$ |

# 8 Conclusion

In this work, we show that there exists a discrepancy between over-smoothing based theoretical results and the practical behavior of deep GCNs. Our theoretical result shows that a deeper GCN can be as expressive as a shallow GCN, if it is properly trained. To truly understand the performance decay issue of deep GCNs, we provide the first transductive uniform stability-based generalization analysis of GCNs and other GCN structures. To improve the optimization issue and benefit from depth, we propose DGCN that enjoys a provable high expressive power and generalization power. We conduct empirical evaluations on various synthetic and real-world datasets to validate the correctness of our theory and advantages over the baselines.

---

[8]$\dagger$ GCNII underfits with the default hyper-parameters. $\ddagger$ GCNII achieves $72.74 \pm 0.16$ by using hidden dimension as 256 and a different design of graph convolution layer. Refer here for details.

## Acknowledgements

This work was supported in part by NSF grant 2008398.

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
