# Supplementary Material
# On Provable Benefits of Depth in Training
# Graph Convolutional Networks

**Organization.** In Section A, we provide additional empirical evaluations on the change of pairwise distances for node embeddings as the number of layers increases. In Section B, we summarize the existing theoretical results on over-smoothing and provide empirical validation on whether over-smoothing happens in practice. In Sections C and D, we provide the proof of Theorem 1 (expressive power) and Theorem 2 (convergence to global optimal and characterization of number of iterations), respectively. In Section E, we provide more empirical results on the effectiveness of the proposed algorithm. In Sections F, G, H, I, and J we provide the generalization analysis of GCN, ResGCN, APPNP, GCNII, and DGCN, respectively. Code can be found at the following repository:

## A   Empirical results on the pairwise distance of node embeddings

In this section, we provide additional empirical evaluations on the change of pairwise distance for node embeddings. First we introduce a concrete definition of the intra- and inter-class pairwise distances, then provide the experimental setups, and finally illustrate the results.

**Definition of pairwise distance.** To verify whether over-smoothing exists in GCNs, we define the intra-class pairwise distance of $\mathbf{H}^{(\ell)}$ as the average pairwise Euclidean distance of two-node embeddings if they have the same ground truth label. Similarly, we define the inter-class pairwise distance of $\mathbf{H}^{(\ell)}$ as the average pairwise distance of two node embeddings if they have different ground truth labels.

$$\text{intra}(\mathbf{H}^{(\ell)}) \triangleq \frac{\sum_{i,j=1}^{N} \mathbf{1}\{y_i = y_j\} \cdot \|\boldsymbol{h}_i^{(\ell)} - \boldsymbol{h}_j^{(\ell)}\|_2^2}{\|\mathbf{H}^{(\ell)}\|_F \cdot \sum_{i,j=1}^{N} \mathbf{1}\{y_i = y_j\}},$$

$$\text{inter}(\mathbf{H}^{(\ell)}) \triangleq \frac{\sum_{i,j=1}^{N} \mathbf{1}\{y_i \neq y_j\} \cdot \|\boldsymbol{h}_i^{(\ell)} - \boldsymbol{h}_j^{(\ell)}\|_2^2}{\|\mathbf{H}^{(\ell)}\|_F \cdot \sum_{i,j=1}^{N} \mathbf{1}\{y_i \neq y_j\}}.$$

Both intra- and inter-class distances are normalized using the Frobenius norm to eliminate the difference caused by the scale of the node embedding matrix. By the definition of over-smoothing in [35], the distance between intra- and inter-class pairwise distance should be decreasing sharply as the number of GCN layers increases.

**Setup.** In Figure 5 and Figure 6, we plot the training error in a 10-layer GCN [28], SGC [54], and APPNP [29] models until convergence, and choose the model with the best validation score for pairwise distance computation. The pairwise distance at the $\ell$th layer is computed by the node embedding matrix $\mathbf{H}^{(\ell)}$ generated by the selected model. We repeat each experiment 10 times and plot the mean and standard deviation of the results.

**Results.** As shown in Figure 5 and Figure 6, when ignoring the weight matrices, i.e., sub-figures in box (b), the intra- and inter-class pairwise distance are decreasing but the gap between intra- and inter-class pairwise distance is increasing as the model goes deeper. That is, the difference between intra- and inter-class nodes' embeddings is increasing and becoming more discriminative. On the other hand, when considering the weight matrices, i.e., sub-figures in box (a), the difference between intra- and inter-class node embeddings is large and increasing as the model goes deeper. In other words, weight matrices learn to make node embeddings discriminative and generate expressive node embeddings during training.

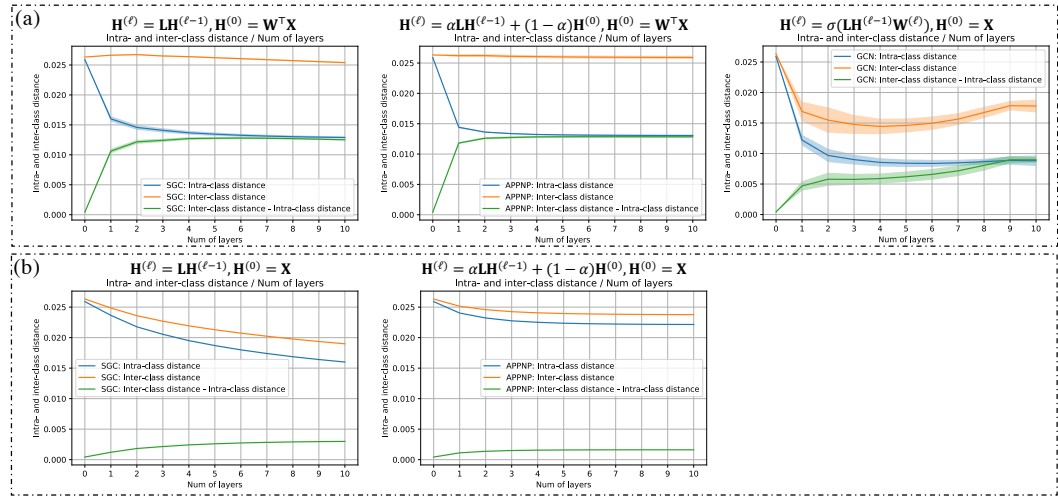

Figure 5: Comparison of the pairwise distance for intra- and inter-class node embeddings on *Cora* dataset for different models by increasing the number of layers in the proposed architecture.

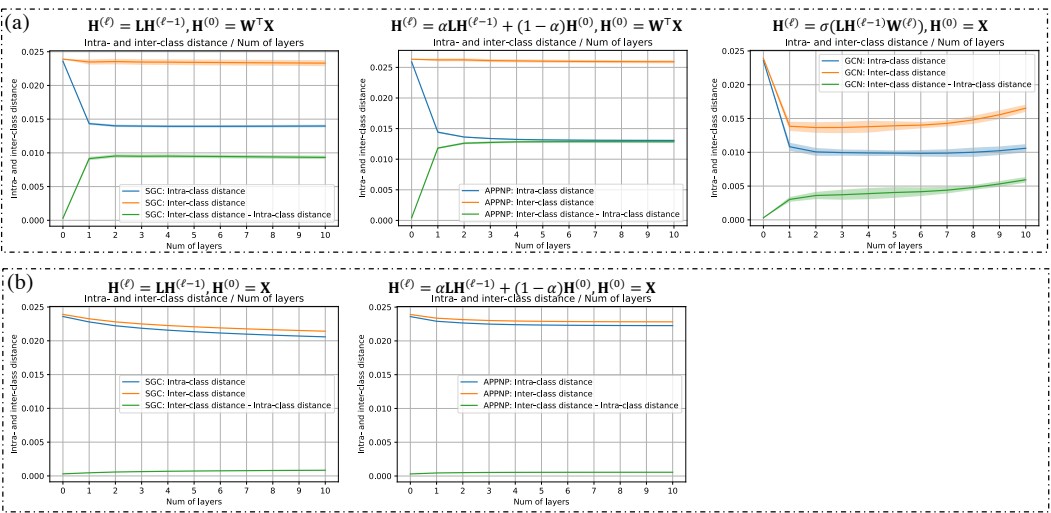

Figure 6: Comparison of the pairwise distance for intra- and inter-class node embeddings on *Citeseer* dataset for different models by increasing the number of layers in the proposed architecture.

## B   A summary of theoretical results on over-smoothing

In this section, we survey the existing theories on understanding over-smoothing in GCNs from [26, 41, 4] and illustrate why the underlying assumptions may not hold in practice. Finally, we discuss conditions where over-smoothing and over-fitting might happen simultaneously.

Before processing, let first recall the notation we defined in Section 3. Recall that $\mathbf{A} \in \mathbb{R}^{N \times N}$ denotes the adjacency matrix of the self-connected graph, i.e., $A_{i,j} = 1$ if edge $(i, j) \in \mathcal{E}$ and $A_{i,j} = 0$ otherwise, $\mathbf{D} \in \mathbb{R}^{N \times N}$ denotes the corresponding degree matrix, i.e., $D_{i,i} = \deg(i)$ and $D_{i,j} = 0$ if $i \neq j$, and the symmetric Laplacian matrix is computed as $\mathbf{L} = \mathbf{D}^{-1/2}\mathbf{A}\mathbf{D}^{-1/2}$.

### B.1   Expressive power based analysis [42, 26]

**Proposition 1** (Proposition 1 in [41], Theorem 1 of [26]). *Let $\lambda_1 \geq \ldots \geq \lambda_N$ denote the eigenvalues of the Laplacian matrix $\mathbf{L}$ in descending order, and let $\mathbf{e}_i$ denote the eigenvector associated with the eigenvalue $\lambda_i$. Suppose graph has $M \leq N$ connected components, then we have $\lambda_1 = \ldots = \lambda_M = 1$*

and $1 > \lambda_{M+1} > \ldots > \lambda_N$. *Let* $\mathbf{E} = \{\mathbf{e}_i\}_{i=1}^M \in \mathbb{R}^{M \times N}$ *denote the stack of eigenvectors that associated to eigenvalues* $\lambda_1, \ldots, \lambda_M$. *Then for all* $i \in [M]$, *the eigenvector* $\mathbf{e}_i \in \mathbb{R}^N$ *is defined as*

$$\mathbf{e}_i = \{e_i(j)\}_{j=1}^N, \ e_i(j) \propto \begin{cases} \sqrt{\deg(j)} & \text{if } j\text{th node is in the } i\text{th connected component} \\ 0 & \text{otherwise} \end{cases}. \quad (3)$$

*Proof.* The proof can be found in Section B of [41]. □

[41] proposes to measure the expressive power of node embeddings using its distance to a subspace $\mathcal{M}$ that only has node degree information, i.e., regardless of the node feature information.

**Definition 2** (Subspace and distance to subspace)**.** *We define subspace* $\mathcal{M}$ *as*

$$\mathcal{M} := \{\mathbf{E}^\top \mathbf{R} \in \mathbb{R}^{N \times d} \mid \mathbf{R} \in \mathbb{R}^{M \times d}\}, \quad (4)$$

*where* $\mathbf{E} = \{\mathbf{e}_i\}_{i=1}^M \in \mathbb{R}^{M \times N}$ *is the orthogonal basis of Laplacian matrix* $\mathbf{L}$, *and* $\mathbf{R}$ *is any random matrix. The distance of a node embedding matrix* $\mathbf{H}^{(\ell)} \in \mathbb{R}^{N \times d}$ *to the subspace can be computed as* $d_{\mathcal{M}}(\mathbf{H}^{(\ell)}) = \inf_{\mathbf{Y} \in \mathcal{M}} \|\mathbf{H}^{(\ell)} - \mathbf{Y}\|_{\mathrm{F}}$, *where* $\|\cdot\|_{\mathrm{F}}$ *denotes the Frobenius norm of the matrix.*

Subspace $\mathcal{M}$ is a $d$-dimensional subspace defined by orthogonal vectors $\{\mathbf{e}_i\}_{i=1}^M$ that only captures node degree information. Intuitively, a smaller distance $d_{\mathcal{M}}(\mathbf{H}^{(\ell)})$ means that $\mathbf{H}^{(\ell)}$ is losing its feature information but only carries the degree information.

In the following, we summarize the theoretical results of over-smoothing in [26] and extend their results to SGC and GCNII, which will be used in our discussion below.

**Theorem 6** (Theorem 2 of [26])**.** *Let* $\lambda = \max_{i \in [M+1, N]} |\lambda_i|$ *denote the largest absolute eigenvalue of Laplacian matrix* $\mathbf{L}$ *which is bounded by* 1, *let* $s$ *denote the largest singular-value of weight parameter* $\mathbf{W}^{(\ell)}$, $\ell \in [L]$, *then we have*

$$d_{\mathcal{M}}(\mathbf{H}^{(\ell)}) - \varepsilon \le \gamma \Big( d_{\mathcal{M}}(\mathbf{H}^{(\ell-1)}) - \varepsilon \Big), \quad (5)$$

*where* $\gamma$ *and* $\varepsilon$ *are functions of* $\lambda$ *and* $s$ *that depend on the model structure.*

**GCN.** For vanilla *GCN* model with the graph convolutional operation defined as

$$\mathbf{H}^{(\ell)} = \sigma(\mathbf{L}\mathbf{H}^{(\ell-1)}\mathbf{W}^{(\ell)}), \ \mathbf{H}^{(0)} = \mathbf{X}, \quad (6)$$

we have $\gamma_{\mathsf{GCN}} = \lambda s$ and $\epsilon_{\mathsf{GCN}} = 0$.

Under the assumption that $\gamma_{\mathsf{GCN}} \le 1$, i.e., the graph is densely connected and the largest singular value of weight matrices is small, we have $d_{\mathcal{M}}(\mathbf{H}^{(\ell+1)}) \le \gamma_{\mathsf{GCN}} \cdot d_{\mathcal{M}}(\mathbf{H}^{(\ell)})$. In other words, the vanilla *GCN* model is losing expressive power as the number of layers increases.

However, if we suppose there exist a trainable bias parameter $\mathbf{B}^{(\ell)} = \{\mathbf{b}^{(\ell)}\}_{i=1}^N \in \mathbb{R}^{N \times d_\ell}$ in each graph convolutional layer

$$\mathbf{H}^{(\ell)} = \sigma(\mathbf{L}\mathbf{H}^{(\ell-1)}\mathbf{W}^{(\ell)} + \mathbf{B}^{(\ell)}), \ \mathbf{H}^{(0)} = \mathbf{X}, \quad (7)$$

we have $\gamma_{\mathsf{GCN}} = \lambda s$ and $\epsilon_{\mathsf{GCN}} = d_{\mathcal{M}}(\mathbf{B}^{(\ell)})$. When $d_{\mathcal{M}}(\mathbf{B}^{(\ell)})$ is considerably large, there is no guarantee that *GCN-bias* suffers from losing expressive power issue.

**ResGCN.** For GCN with residual connection, the graph convolution operation is defined as

$$\mathbf{H}^{(\ell)} = \sigma(\mathbf{L}\mathbf{H}^{(\ell)}\mathbf{W}^{(\ell)}) + \mathbf{H}^{(\ell-1)}, \ \mathbf{H}^{(0)} = \mathbf{X}\mathbf{W}^{(0)}, \quad (8)$$

where we have $\gamma_{\mathsf{ResGCN}} = 1 + \lambda s$ and $\epsilon_{\mathsf{ResGCN}} = 0$. Since $\gamma_{\mathsf{ResGCN}} \ge 1$, there is no guarantee that *ResGCN* suffers from losing expressive power.

**APPNP.** For *APPNP* with graph convolution operation defined as

$$\mathbf{H}^{(\ell)} = \alpha\mathbf{L}\mathbf{H}^{(\ell)} + (1-\alpha)\mathbf{H}^{(0)}, \ \mathbf{H}^{(0)} = \mathbf{X}\mathbf{W}^{(0)}, \quad (9)$$

we have $\gamma_{\mathsf{APPNP}} = \alpha\lambda$ and $\epsilon_{\mathsf{APPNP}} = \frac{(1-\alpha)d_{\mathcal{M}}(\mathbf{H}^{(0)})}{1-\gamma_{\mathsf{APPNP}}}$. Although $\gamma_{\mathsf{APPNP}} < 1$, because $\epsilon_{\mathsf{APPNP}}$ can be large, there is no guarantee that *APPNP* suffers from losing expressive power.

**GCNII.** For *GCNII* with graph convolution operation defined as

$$\mathbf{H}^{(\ell)} = \sigma\Big(\big((\alpha\mathbf{L}\mathbf{H}^{(\ell)} + (1-\alpha)\mathbf{H}^{(0)})(\beta\mathbf{W}^{(\ell)} + (1-\beta)\mathbf{I}_N)\big)\Big), \ \mathbf{H}^{(0)} = \mathbf{X}\mathbf{W}^{(0)}, \qquad (10)$$

we have $\gamma_{\mathsf{GCNII}} = \Big(1 - (1-\beta)(1-s)\Big)\alpha\lambda$, and $\epsilon_{\mathsf{GCNII}} = \frac{(1-\alpha)d_{\mathcal{M}}(\mathbf{H}^{(0)})}{1-\gamma_{\mathsf{GCNII}}}$. Although we have $\gamma_{\mathsf{GCNII}} < 1$, because $\epsilon_{\mathsf{GCNII}}$ can be considerably large, there is no guarantee that *GCNII* suffers from losing expressive power.

**SGC.** The result can be also extended to the linear model *SGC* [54], where the graph convolution is defined as

$$\mathbf{H}^{(\ell)} = \mathbf{L}\mathbf{H}^{(\ell-1)}, \ \mathbf{H}^{(0)} = \mathbf{X}\mathbf{W}^{(0)}, \qquad (11)$$

we have $\gamma_{\mathsf{SGC}} = \lambda$, and $\epsilon_{\mathsf{SGC}} = 0$, which guarantees losing expressive power as the number of layers increases.

**Discussion on the result of Theorem 6.** In summary, the linear model *SGC* always suffers from losing expressive power without the assumption on the trainable parameters. Under the assumption that the multiplication of the *largest singular-value of trainable parameter* and *the largest absolute eigenvalue of Laplacian matrix smaller than* 1, i.e., $\lambda s \leq 1$, we can only guarantee that *GCN* suffers from losing expressive power issue, but cannot have the same guarantee on *GCN-bias*, *ResGCN*, *APPNP*, and *GCNII*. However, as we show in Figure 3, the assumption is not going to hold in practice, and the distances are not decreasing for most cases.

**Remark 2.** *[41] conducts experiments on Erdős-Rényi graph to show that when the graph is sufficiently dense and large, vanilla GCN suffers from expressive lower loss. Erdős-Rényi graph is constructed by connecting nodes randomly. Each edge is included in the graph with probability $p$ independent from every other edge. To guarantee a small $\lambda$, a dense graph with larger $p$ is required. For example, in the Section 6.2 of [41], they choose $p = 0.5$ (one node is connected to $50\%$ of other nodes) such that $\lambda = 0.063$ and choose $p = 0.1$ (one node is connected to $10\%$ of other nodes) such that $\lambda = 0.195$. However, real world datasets are sparse and have a $\lambda$ that is closer to $1$. For example, Cora has $\lambda = 0.9964$, Citeseer has $\lambda = 0.9987$, and Pubmed has $\lambda = 0.9905$.*

In Figure 7 and Figure 8, we compare the testing set F1-score, expressive power metric $d_{\mathcal{M}}(\mathbf{H}^{(\ell)})$ before and after training. $d_{\mathcal{M}}(\mathbf{H}^{(\ell)})$ is computed on *the final output of a $\ell$-layer GCN model*. The "after training results" are computed on the model with the best testing score. We repeat each experiment 10 times and report the average values. As shown in Figure 7 and Figure 8, the expressive power metric $d_{\mathcal{M}}(\mathbf{H}^{(\ell)})$ of the *untrained* vanilla GCN is decreasing as the number of layers increases. However, the expressive power metric $d_{\mathcal{M}}(\mathbf{H}^{(\ell)})$ of the *trained* vanilla GCN increases with more layers. Besides, we observe that there is no obvious connection between the testing F1-score to $d_{\mathcal{M}}(\mathbf{H}^{(\ell)})$ for other GCN variant. For example, the testing F1-score of APPNP is increasing while its $d_{\mathcal{M}}(\mathbf{H}^{(\ell)})$ is not changing much.

In Figure 9 and Figure 10, we compare the testing set F1-score, expressive power metric $d_{\mathcal{M}}(\mathbf{H}^{(\ell)})$ before and after training. $d_{\mathcal{M}}(\mathbf{H}^{(\ell)})$ is computed on *the $\ell$th layer intermediate node embeddings of a* 10-*layer GCN model*. The "after training results" are computed on the model with the best testing score. We repeat experiment 10 times and report the average values. As shown in Figure 9 and Figure 10, the expressive power metric $d_{\mathcal{M}}(\mathbf{H}^{(\ell)})$ of the *untrained* vanilla GCN is decreasing as the number of layers increases. However, the expressive power metric $d_{\mathcal{M}}(\mathbf{H}^{(\ell)})$ of the *trained* vanilla GCN increases as the number of layers increases.

## B.2 Dirichlet energy based analysis [4]

**Definition 3** (Dirichlet energy). *The Dirichlet energy of node embedding matrix $\mathbf{H}^{(\ell)}$ is defined as*

$$DE(\mathbf{H}^{(\ell)}) = \frac{1}{2}\sum_{i,j=1}^{N} L_{i,j}\Big\|\frac{\mathbf{h}_i^{(\ell)}}{\sqrt{\deg(i)}} - \frac{\mathbf{h}_j^{(\ell)}}{\sqrt{\deg(j)}}\Big\|_2^2. \qquad (12)$$

Intuitively, the Dirichlet energy measures the smoothness of node embeddings. Based on the definition of Dirichlet energy, they obtain a similar result as shown in [42] that

$$DE(\mathbf{H}^{(\ell+1)}) \leq \lambda s \cdot DE(\mathbf{H}^{(\ell)}), \qquad (13)$$

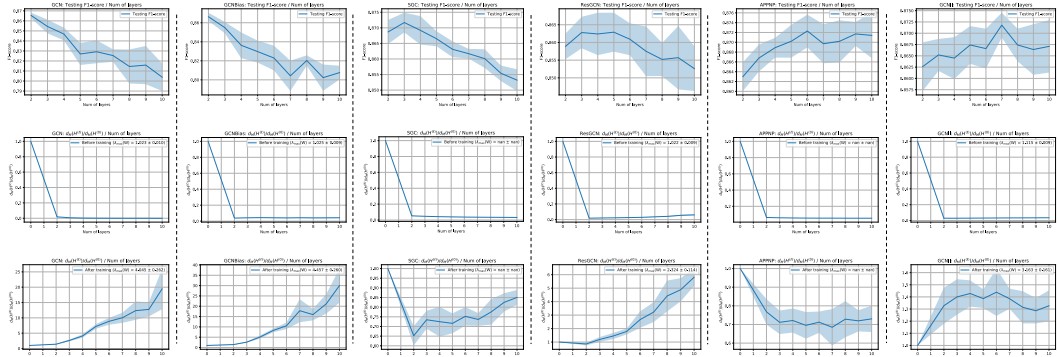

Figure 7: Comparison of F1-score, expressive power metric $d_{\mathcal{M}}(\mathbf{H}^{(\ell)})$ of *GCN*, *GCN-bias*, *SGC*, *ResGCN*, *APPNP*, and *GCNII* before and after training on *Cora* dataset. The average "largest sigular value" of weight matrices in graph convolutional layers are reported, where "nan" stands for no weight matrices inside graph convolution layers. We only consider GCN model with depth from 2-layers to 10-layers. (Better viewed in PDF version)

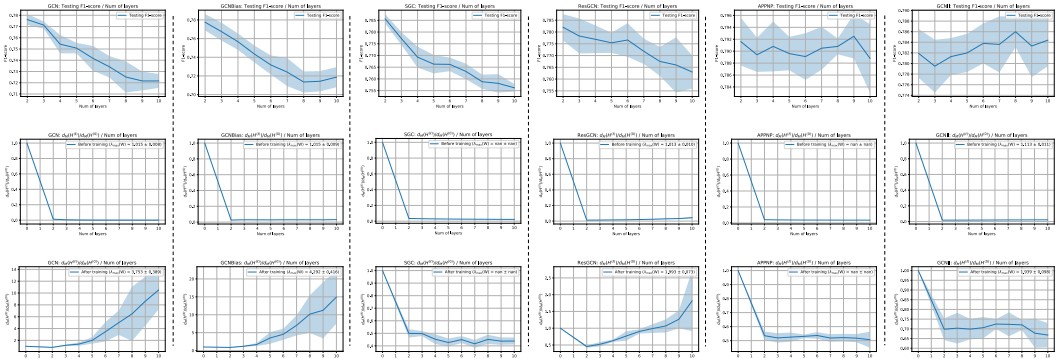

Figure 8: Comparison of F1-score, expressive power metric $d_{\mathcal{M}}(\mathbf{H}^{(\ell)})$ of *GCN*, *GCN-bias*, *SGC*, *ResGCN*, *APPNP*, and *GCNII* before and after training on *Citeseer* dataset. The average "largest singular value" of weight matrices in graph convolutional layers are reported, where "nan" stands for no weight matrices inside graph convolution layers. We only consider GCN model with depth from 2-layers to 10-layers. (Better viewed in PDF version)

where $\lambda$ is *largest absolute eigenvalue of Laplacian matrix* $\mathbf{L}$ *that is less than* 1, and $s$ is *the largest singular-value of trainable parameter* $\mathbf{W}^{(\ell)}$.

**Remark 3.** *The Dirichlet energy used in [4] is closely related to the pairwise distance as shown in Figure 2. In Figure 2, we measure the "smoothness" or "expressive power" of node embeddings using a normalized Dirichlet energy function, i.e., $\frac{DE(\mathbf{H}^{(\ell)})}{\|\mathbf{H}^{(\ell)}\|_F^2}$ for inner- and cross-class nodes respectively, which refers to the real smoothness metric of graph signal as pointed out in the footnote 1 of [4].*

Intuitively, under the assumption that the largest singular-value of weight matrices less than 1, as the number of layers increase, i.e., $\ell \to \infty$, the Dirichlet energy of node embeddings $\text{DE}(\mathbf{H}^{(\ell)})$ is decreasing. However, recall that this assumption is not likely to hold in practice because the real-world graphs are usually sparse and the largest singular value of weight matrices is usually larger than 1.

Besides, generalizing the result to other GCN variants (e.g., ResGCN, GCNII) is non-trivial. For example in ResGCN, we have

$$
\begin{aligned}
\text{DE}(\mathbf{H}^{(\ell)}) &= \text{DE}(\sigma(\mathbf{L}\mathbf{H}^{(\ell-1)}\mathbf{W}^{(\ell)}) + \mathbf{H}^{(\ell-1)}) \\
&\leq 2\text{DE}(\sigma(\mathbf{L}\mathbf{H}^{(\ell-1)}\mathbf{W}^{(\ell)})) + 2\text{DE}(\mathbf{H}^{(\ell-1)}) \\
&\leq 2(\lambda s + 1)\text{DE}(\mathbf{H}^{(\ell-1)}).
\end{aligned}
\tag{14}
$$

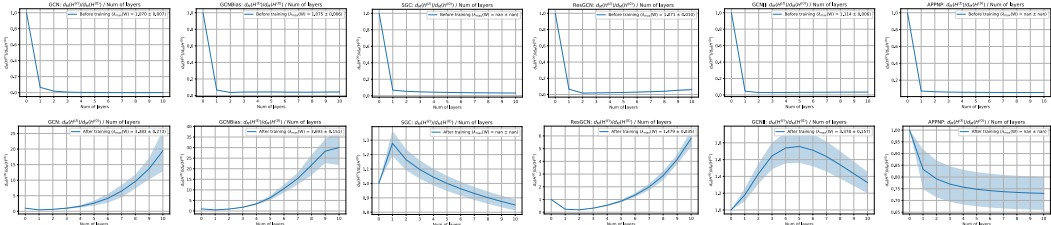

Figure 9: Comparison of F1-score, expressive power metric $d_{\mathcal{M}}(\mathbf{H}^{(\ell)})$ of 10-layer *GCN*, *GCN-bias*, *SGC*, *ResGCN*, *APPNP*, and *GCNII* before and after training on *Cora* dataset. The average "largest singular value" of weight matrices in graph convolutional layers are reported, where "nan" stands for no weight matrices inside graph convolution layers. (Better viewed in PDF version)

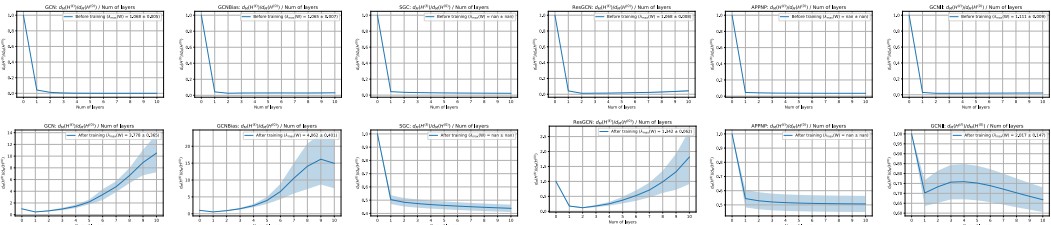

Figure 10: Comparison of F1-score, expressive power metric $d_{\mathcal{M}}(\mathbf{H}^{(\ell)})$ of 10-layer *GCN*, *GCN-bias*, *SGC*, *ResGCN*, *APPNP*, and *GCNII* before and after training on *Citeseer* dataset. The average "largest singular value" of weight matrices in graph convolutional layers are reported, where "nan" stands for no weight matrices inside graph convolution layers. (Better viewed in PDF version)

The coefficient 2 on the right hand side of the inequality makes the result less meaningful.

### B.3 A condition to have over-smoothing and over-fitting happen simultaneously

As it has alluded to before, in this paper, we argue that the performance degradation issue is mainly due to over-fitting, and over-smoothing is not likely to happen in practice. One might doubt whether over-smoothing can happen simultaneously with over-fitting. In the following, we show that over-smoothing and over-fitting might happen at the same time if a model can distinguish all nodes by only using graph structure information (e.g., node degree) and ignoring node feature information. However, such a condition is not likely in the practice.

For simplicity, let us assume that the graph contains only a single connected component. Let first consider the over-smoothing notation as defined in [41]. We can write the node embedding of node $i$ at the $\ell$th layer as $\mathbf{h}_i^{(\ell)} = \mathbf{e}\mathbf{R}_{i,\ell} + \boldsymbol{\varepsilon}_{i,\ell} \in \mathbb{R}^d$, where $\mathbf{e}$ is the eigen vector associated with the largest eigenvalue, $\mathbf{R}_{i,\ell} \in \mathbb{R}^{N \times d}$ is the random projection matrix, and $\boldsymbol{\varepsilon}_{i,\ell} \in \mathbb{R}^d$ is a $d$-dimensional random vector such that $d_{\mathcal{M}}(\mathbf{h}_i^{(\ell)}) = \|\boldsymbol{\varepsilon}_{i,\ell}\|_2$ corresponds to its distance to subspace $\mathcal{M}$. According to the notation in [41], we have $\|\boldsymbol{\varepsilon}_{i,\ell}\|_2 \to 0$ as $\ell \to +\infty$ for any $i \in \mathcal{V}$. The minimum training loss can be achieved by choosing the final layer weight vector $\mathbf{w} \in \mathbb{R}^d$ such that $\sum_{i \in \mathcal{V}} \text{Loss}\big(\mathbf{w}^\top (\mathbf{e}\mathbf{R}_{i,\ell} + \boldsymbol{\varepsilon}_{i,\ell}), y_i\big)$ is small. A small training error is achieved when $\mathbf{e}\mathbf{R}_{i,\ell}$ is discriminative. In other words, a model is under both over-smoothing and over-fitting conditions only if it can make predictions based on the graph structure, without leveraging node feature information. On the other hand, considering the over-smoothing notation as defined in [4], we know that over-smoothing happens if $\frac{\mathbf{h}_i^{(\ell)}}{\sqrt{\deg(i)}} \approx \frac{\mathbf{h}_j^{(\ell)}}{\sqrt{\deg(j)}}$ for any $i, j \in \mathcal{V}$, i.e., a classifier can distinguish any nodes based on its node degree (only based on graph information without leveraging its feature information).

# C  Proof of Theorem 1

Given a $L$-layer computation tree $\mathcal{T}^L$, let $v_i^\ell$ denote the $i$th node in the $\ell$th layer of a $L$-layer computation tree $\mathcal{T}^L$. Let suppose each node has at least $d$ neighbors and each node has binary feature.

Let $P(v_i^\ell)$ denote the parent of node $v_i^\ell$ and $C(v_i^\ell)$ denote the set of children of node $v_i^\ell$ with $|C(v_i^\ell)| \geq d$. By the definition of computation tree, we know that $P(v_i^\ell) \in C(v_i^\ell)$ for any $\ell > 1$. As a result, we know that $C(v_i^\ell)$ has at least $(d-1)$ different choices since one of the node in $C(v_i^\ell)$ must be the same as $P(v_i^\ell)$.

Therefore, we know that a $L$-layer computation tree $\mathcal{T}^L$ has at least $2(d-1)^{L-1}$ different choices, where the constant 2 is because each root node has at least two different choices.

# D  Proof of Theorem 2

In this section, we study the convergence of $L$-layer deep GCN model, and show that a deeper model requires more iterations $T$ to achieve $\epsilon$ training error. Before preceding, let first introduce the notations used for the convergence analysis of deep GCN. Let $\mathcal{G}(\mathcal{V}, \mathcal{E})$ as the graph of $N = |\mathcal{V}|$ nodes and each node in the graph is associated with a node feature and target label pair $(\mathbf{x}_i, \mathbf{y}_i)$, where $\mathbf{x}_i \in \mathbb{R}^{d_0}$ and $\mathbf{y}_i \in \mathbb{R}^{d_{L+1}}$. Let $\mathbf{X} = \{\mathbf{x}_i\}_{i=1}^N \in \mathbb{R}^{N \times d_0}$ and $\mathbf{Y} = \{\mathbf{y}_i\}_{i=1}^N \in \mathbb{R}^{N \times d_{L+1}}$ be the stack of node vector and target vector of the training data set of size $N$. Given a set of $L$ matrices $\boldsymbol{\theta} = \{\mathbf{W}^{(\ell)}\}_{\ell=1}^L$, we consider the following training objective with squared loss function to learn the parameters of the model:

$$\underset{\boldsymbol{\theta}}{\text{minimize}} \ \mathcal{L}(\boldsymbol{\theta}) = \frac{1}{2}\|\widehat{\mathbf{Y}}(\boldsymbol{\theta}) - \mathbf{Y}\|_{\mathrm{F}}^2, \ \widehat{\mathbf{Y}}(\boldsymbol{\theta}) = \mathbf{H}^{(L)}\mathbf{W}^{(L+1)}, \ \mathbf{H}^{(\ell)} = \sigma(\mathbf{L}\mathbf{H}^{(\ell-1)}\mathbf{W}^{(\ell)}), \quad (15)$$

where $\sigma(\cdot)$ is ReLU activation.

## D.1  Useful lemmas

The following lemma analyzes the Lipschitz continuity of deep GCNs, which characterizes the change of inner layer representations by slight change of the weight parameters.

**Lemma 2.** *Let* $\boldsymbol{\theta}_1 = \{\mathbf{W}_1^{(\ell)}\}_{\ell \in [L+1]}$ *and* $\boldsymbol{\theta}_2 = \{\mathbf{W}_2^{(\ell)}\}_{\ell \in [L+1]}$ *be two set of parameters. Let* $\lambda_\ell \geq \max\{\|\mathbf{W}_1^{(\ell)}\|_2, \|\mathbf{W}_2^{(\ell)}\|_2\}$, $s \geq \|\mathbf{L}\|_2$, *and* $\lambda_{i \to j} = \prod_{\ell=i}^j \lambda_\ell$, *we have*

$$\|\mathbf{H}_1^{(\ell)} - \mathbf{H}_2^{(\ell)}\|_{\mathrm{F}} \leq s^\ell \lambda_{1 \to \ell}\|\mathbf{X}\|_{\mathrm{F}} \sum_{j=1}^\ell \lambda_j^{-1}\|\mathbf{W}_1^{(j)} - \mathbf{W}_2^{(j)}\|_2. \quad (16)$$

*Proof of Lemma 2.* Our proof relies on the following inequality

$$\forall \mathbf{A} \in \mathbb{R}^{a \times b}, \mathbf{B} \in \mathbb{R}^{b \times c}, \ \|\mathbf{A}\mathbf{B}\|_{\mathrm{F}} \leq \|\mathbf{A}\|_2\|\mathbf{B}\|_{\mathrm{F}} \text{ and } \|\mathbf{A}\mathbf{B}\|_{\mathrm{F}} \leq \|\mathbf{A}\|_{\mathrm{F}}\|\mathbf{B}\|_2. \quad (17)$$

By definition of $\mathbf{H}^{(\ell)}$, we have for $\ell \in [L]$

$$\begin{aligned}
\|\mathbf{H}^{(\ell)}\|_{\mathrm{F}} &= \|\sigma(\mathbf{L}\mathbf{H}^{(\ell-1)}\mathbf{W}^{(\ell)})\|_{\mathrm{F}} \\
&\leq \|\mathbf{L}\mathbf{H}^{(\ell-1)}\mathbf{W}^{(\ell)}\|_{\mathrm{F}} \\
&\leq \|\mathbf{L}\|_2\|\mathbf{H}^{(\ell-1)}\mathbf{W}^{(\ell)}\|_{\mathrm{F}} \\
&\leq \|\mathbf{L}\|_2\|\mathbf{W}^{(\ell)}\|_2\|\mathbf{H}^{(\ell-1)}\|_{\mathrm{F}} \\
&\leq \|\mathbf{X}\|_{\mathrm{F}} \prod_{j=1}^\ell \left(\|\mathbf{L}\|_2\|\mathbf{W}^{(\ell)}\|_2\right).
\end{aligned} \quad (18)$$

Let $\boldsymbol{\theta}_1 = \{\mathbf{W}_1^{(\ell)}\}_{\ell \in [L+1]}$ and $\boldsymbol{\theta}_2 = \{\mathbf{W}_2^{(\ell)}\}_{\ell \in [L+1]}$ be two set of parameters, we have

$$
\begin{aligned}
\|\mathbf{H}_1^{(\ell)} - \mathbf{H}_2^{(\ell)}\|_{\mathrm{F}} &= \|\sigma(\mathbf{L}\mathbf{H}_1^{(\ell-1)}\mathbf{W}_1^{(\ell)}) - \sigma(\mathbf{L}\mathbf{H}_2^{(\ell-1)}\mathbf{W}_2^{(\ell)})\|_{\mathrm{F}} \\
&\underset{(a)}{\leq} \|\mathbf{L}\mathbf{H}_1^{(\ell-1)}\mathbf{W}_1^{(\ell)} - \mathbf{L}\mathbf{H}_2^{(\ell-1)}\mathbf{W}_2^{(\ell)}\|_{\mathrm{F}} \\
&\leq \|\mathbf{L}\|_2 \|\mathbf{H}_1^{(\ell-1)}\mathbf{W}_1^{(\ell)} - \mathbf{H}_2^{(\ell-1)}\mathbf{W}_2^{(\ell)}\|_{\mathrm{F}} \\
&= \|\mathbf{L}\|_2 \left( \|\mathbf{H}_1^{(\ell-1)}(\mathbf{W}_1^{(\ell)} - \mathbf{W}_2^{(\ell)}) + (\mathbf{H}_1^{(\ell-1)} - \mathbf{H}_2^{(\ell-1)})\mathbf{W}_2^{(\ell)}\|_{\mathrm{F}} \right) \\
&\leq \|\mathbf{L}\|_2 \left( \|\mathbf{H}_1^{(\ell-1)}\|_{\mathrm{F}} \|\mathbf{W}_1^{(\ell)} - \mathbf{W}_2^{(\ell)}\|_2 + \|\mathbf{H}_1^{(\ell-1)} - \mathbf{H}_2^{(\ell-1)}\|_{\mathrm{F}} \|\mathbf{W}_2^{(\ell)}\|_2 \right) \\
&\underset{(b)}{\leq} \|\mathbf{L}\|_2 \|\mathbf{X}\|_{\mathrm{F}} \left[ \prod_{j=1}^{\ell-1} \left( \|\mathbf{L}\|_2 \|\mathbf{W}_1^{(\ell)}\|_2 \right) \right] \|\mathbf{W}_1^{(\ell)} - \mathbf{W}_2^{(\ell)}\|_2 \\
&\quad + \|\mathbf{L}\|_2 \|\mathbf{W}_2^{(\ell)}\|_2 \|\mathbf{H}_1^{(\ell-1)} - \mathbf{H}_2^{(\ell-1)}\|_{\mathrm{F}},
\end{aligned}
\tag{19}
$$

where $(a)$ is due to $\sigma(\cdot)$ is 1-Lipschitz continuous, $(b)$ is due to Eq. 18.

Let $\lambda_\ell \geq \max\{\|\mathbf{W}_1^{(\ell)}\|_2, \|\mathbf{W}_2^{(\ell)}\|_2\}$, $s \geq \|\mathbf{L}\|_2$, and $\lambda_{i \to j} = \prod_{\ell=i}^{j} \lambda_\ell,$, we have

$$
\begin{aligned}
\|\mathbf{H}_1^{(\ell)} - \mathbf{H}_2^{(\ell)}\|_{\mathrm{F}} &\leq s^\ell \lambda_{1 \to (\ell-1)} \|\mathbf{X}\|_{\mathrm{F}} \|\mathbf{W}_1^{(\ell)} - \mathbf{W}_2^{(\ell)}\|_2 + s\lambda_\ell \|\mathbf{H}_1^{(\ell-1)} - \mathbf{H}_2^{(\ell-1)}\|_{\mathrm{F}} \\
&\leq s^\ell \lambda_{1 \to \ell} \|\mathbf{X}\|_{\mathrm{F}} \left( \sum_{j=1}^{\ell} \lambda_j^{-1} \|\mathbf{W}_1^{(j)} - \mathbf{W}_2^{(j)}\|_2 \right).
\end{aligned}
\tag{20}
$$

$\square$

The following lemma derives the upper bound of the gradient with respect to the weight parameters, which plays an important role in the convergence analysis of deep GCN.

**Lemma 3.** *Let* $\boldsymbol{\theta} = \{\mathbf{W}^{(\ell)}\}_{\ell \in [L+1]}$ *be the set of parameters,* $\lambda_\ell \geq \max\{\|\mathbf{W}_1^{(\ell)}\|_2, \|\mathbf{W}_2^{(\ell)}\|_2\}$, $s \geq \|\mathbf{L}\|_2$, *and* $\lambda_{i \to j} = \prod_{\ell=i}^{j} \lambda_\ell$, *we have*

$$
\left\| \frac{\partial \mathcal{L}(\boldsymbol{\theta})}{\partial \mathbf{W}^{(\ell)}} \right\|_{\mathrm{F}} \leq \frac{s^L \lambda_{1 \to (L+1)}}{\lambda_\ell} \|\mathbf{X}\|_{\mathrm{F}} \|\widehat{\mathbf{Y}} - \mathbf{Y}\|_{\mathrm{F}}.
\tag{21}
$$

*Proof of Lemma 3.* By the definition of $\frac{\partial \mathcal{L}(\boldsymbol{\theta})}{\partial \mathbf{W}^{(\ell)}}$, we have

$$
\begin{aligned}
\left\| \frac{\partial \mathcal{L}(\boldsymbol{\theta})}{\partial \mathbf{W}^{(\ell)}} \right\|_{\mathrm{F}} &= \left\| \left( \frac{\partial \widehat{\mathbf{Y}}}{\partial \mathbf{W}^{(\ell)}} \right)^\top (\widehat{\mathbf{Y}} - \mathbf{Y}) \right\|_{\mathrm{F}} \\
&= \left\| \frac{\partial [\mathbf{I}_N \otimes (\mathbf{W}^{(L+1)})^\top] \mathrm{vec}(\mathbf{H}^{(L)})}{\partial \mathrm{vec}(\mathbf{W}^{(\ell)})} \mathrm{vec}(\widehat{\mathbf{Y}} - \mathbf{Y}) \right\|_2 \\
&= \left\| [\mathbf{I}_N \otimes (\mathbf{W}^{(L+1)})^\top] \frac{\partial \mathrm{vec}(\mathbf{H}^{(L)})}{\partial \mathrm{vec}(\mathbf{H}^{(\ell)})} \frac{\partial \mathrm{vec}(\mathbf{H}^{(\ell)})}{\partial \mathrm{vec}(\mathbf{W}^{(\ell)})} \mathrm{vec}(\widehat{\mathbf{Y}} - \mathbf{Y}) \right\|_2 \\
&\underset{(a)}{\leq} \left\| [\mathbf{I}_N \otimes (\mathbf{W}^{(L+1)})^\top] \frac{\partial \mathrm{vec}(\mathbf{L}\mathbf{H}^{(L-1)}\mathbf{W}^{(L)})}{\partial \mathrm{vec}(\mathbf{H}^{(\ell)})} \frac{\partial \mathrm{vec}(\mathbf{L}\mathbf{H}^{(\ell-1)}\mathbf{W}^{(\ell)})}{\partial \mathrm{vec}(\mathbf{W}^{(\ell)})} \mathrm{vec}(\widehat{\mathbf{Y}} - \mathbf{Y}) \right\|_2 \\
&= \left\| [\mathbf{I}_N \otimes (\mathbf{W}^{(L+1)})^\top] \frac{\partial \mathrm{vec}(\mathbf{L} \otimes (\mathbf{W}^{(L)})^\top) \mathrm{vec}(\mathbf{H}^{(L-1)})}{\partial \mathrm{vec}(\mathbf{H}^{(\ell)})} \frac{\partial [\mathbf{L}\mathbf{H}^{(\ell-1)} \otimes \mathbf{I}_{d_\ell}] \mathrm{vec}(\mathbf{W}^{(\ell)})}{\partial \mathrm{vec}(\mathbf{W}^{(\ell)})} \mathrm{vec}(\widehat{\mathbf{Y}} - \mathbf{Y}) \right\|_2 \\
&= \left\| [\mathbf{I}_N \otimes (\mathbf{W}^{(L+1)})^\top] \mathrm{vec} \left( \mathbf{L}^{L-\ell} \otimes (\mathbf{W}^{(\ell+1)} \dots \mathbf{W}^{(L)})^\top \right) [\mathbf{L}\mathbf{H}^{(\ell-1)} \otimes \mathbf{I}_{d_\ell}] \mathrm{vec}(\widehat{\mathbf{Y}} - \mathbf{Y}) \right\|_2 \\
&= \left\| \underbrace{(\mathbf{L}^{L-\ell+1}\mathbf{H}^{(\ell-1)}) \otimes (\mathbf{W}^{(\ell+1)} \dots \mathbf{W}^{(L+1)})^\top}_{\mathbb{R}^{Nd_{L+1} \times d_{\ell-1}d_\ell}} \mathrm{vec}(\widehat{\mathbf{Y}} - \mathbf{Y}) \right\|_{\mathrm{F}}
\end{aligned}
$$

$$= \left\| (\mathbf{W}^{(\ell+1)} \cdots \mathbf{W}^{(L+1)})(\widehat{\mathbf{Y}} - \mathbf{Y})^\top \mathbf{L}^{L-\ell+1} \mathbf{H}^{(\ell-1)} \right\|_2$$

$$\leq s^{L-\ell+1} \lambda_{(\ell+1)\to(L+1)} \| \widehat{\mathbf{Y}} - \mathbf{Y} \|_{\mathrm{F}} \| \mathbf{H}^{(\ell-1)} \|_{\mathrm{F}},$$

where $(a)$ is due to $\| \sigma(x) \|_2 \leq \| x \|_2$.

Using similar proof strategy, we can upper bound $\| \mathbf{H}^{(\ell-1)} \|_{\mathrm{F}}$ by

$$
\begin{aligned}
\| \mathbf{H}^{(\ell-1)} \|_{\mathrm{F}} &= \| \sigma(\mathbf{L}\mathbf{H}^{(\ell-2)}\mathbf{W}^{(\ell-1)}) \|_{\mathrm{F}} \\
&\leq \| \mathbf{L}\mathbf{H}^{(\ell-2)}\mathbf{W}^{(\ell-1)} \|_{\mathrm{F}} \\
&\leq s^{\ell-1} \lambda_{1\to(\ell-1)} \| \mathbf{X} \|_{\mathrm{F}}.
\end{aligned}
\tag{22}
$$

By combining the above two equations, we have

$$\left\| \frac{\partial \mathcal{L}(\boldsymbol{\theta})}{\partial \mathbf{W}^{(\ell)}} \right\|_{\mathrm{F}} \leq \frac{s^L \lambda_{1\to(L+1)}}{\lambda_\ell} \| \mathbf{X} \|_{\mathrm{F}} \| \widehat{\mathbf{Y}} - \mathbf{Y} \|_{\mathrm{F}}. \tag{23}$$

$\square$

## D.2 Main result

In the following, we provide the convergence analysis on deep GCNs. In particular, Theorem 7 shows that under assumption on the weight initialization (Eq. 24) and the width of the last layer's input dimension ($d_L \geq N$), the deep GCN enjoys linear convergence rate. Recall our discussion in Section 4 that $\lambda_\ell \geq 1$ (by Gordon's Theorem for Gaussian matrices) and $\| \mathbf{L} \|_2 = 1$ for $\mathbf{L} = \mathbf{D}^{-1/2}\mathbf{A}\mathbf{D}^{-1/2}$, we know that $s\lambda_\ell \geq 1$, thus deeper model requires a smaller learning rate $\eta$ according to Eq. 25. Since the number of training iteration $T$ is reverse proportional to learning rate $\eta$, which explains why a deeper model requires more training iterations than the shallow one.

**Theorem 7.** *Consider a deep GCN with ReLU activation (defined in Eq. 15) where the width of the last hidden layer satisfies $d_L \geq N$. Let $\{C_\ell\}_{\ell \in [L+1]}$ be any sequence of positive numbers and define $\alpha_0 = \lambda_{\min}(\mathbf{H}_0^{(L)})$, $\lambda_\ell = \| \mathbf{W}_0^{(\ell)} \|_2 + C_\ell$, $\lambda_{i\to j} = \prod_{\ell=i}^j \lambda_\ell$.*

*Assume the following conditions are satisfied at the initialization:*

$$
\begin{aligned}
\alpha_0^2 &\geq 16 s^{2L} \| \mathbf{X} \|_{\mathrm{F}} \max_{\ell \in [L+1]} \frac{\lambda_{1\to(L+1)}}{\lambda_\ell C_\ell} \sqrt{2\mathcal{L}(\boldsymbol{\theta}_0)}, \\
\alpha_0^3 &\geq 32 s^{2L} \| \mathbf{X} \|_{\mathrm{F}}^2 \lambda_{L+1} \sum_{\ell=1}^L \frac{\lambda_{1\to L}^2}{\lambda_\ell^2} \sqrt{2\mathcal{L}(\boldsymbol{\theta}_0)}, \\
\alpha_0^2 &\geq 16 s^{2L} \| \mathbf{X} \|_{\mathrm{F}}^2 \lambda_{L+1}^2 \sum_{\ell=1}^L \frac{\lambda_{1\to L}^2}{\lambda_\ell^2}.
\end{aligned}
\tag{24}
$$

*Let the learning rate satisfies*

$$\eta \leq \min \left( \frac{8}{\alpha_0^2}, \frac{\left( \sum_{\ell=1}^L \lambda_\ell^{-2} \right)}{s^{2L} \lambda_{1\to(L+1)}^2 \| \mathbf{X} \|_{\mathrm{F}}^2 \left( \sum_{\ell=1}^{L+1} \lambda_\ell^{-2} \right)^2} \right). \tag{25}$$

*Then deep GCN can achieve $\mathcal{L}(\boldsymbol{\theta}_T) \leq \epsilon$ for any $T$ satisfied*

$$T \geq \frac{8}{\eta \alpha_0^2} \log \left( \frac{\mathcal{L}(\boldsymbol{\theta}_0)}{\epsilon} \right). \tag{26}$$

*Proof of Theorem 7.* The proof follows the proof of Theorem 2.2 in [40] by extending result from MLP to GCN. The key idea of the proof is to show the followings hold by induction:

$$
\begin{cases}
\| \mathbf{W}_r^{(\ell)} \|_2 \leq \lambda_\ell & \text{for all } \ell \in [L], \ r \in [0, t], \\
\lambda_{\min}(\mathbf{H}_r^{(L)}) \geq \frac{\alpha_0}{2} & \text{for all } r \in [0, t], \\
\mathcal{L}(\boldsymbol{\theta}_r) \leq (1 - \eta \alpha_0^2/8) \mathcal{L}(\boldsymbol{\theta}_0) & \text{for all } r \in [0, t].
\end{cases}
\tag{27}
$$

Clearly, Eq. 27 holds for $t = 0$. Let assume Eq. 27 holds up to iteration $t$, and let show it also holds for $(t + 1)$th iteration.

Let first prove the first inequality holds in Eq. 27. By the triangle inequality, we can decompose the distance between $\mathbf{W}_{t+1}^{(\ell)}$ to $\mathbf{W}_0^{(\ell)}$ by

$$
\begin{aligned}
\|\mathbf{W}_{t+1}^{(\ell)} - \mathbf{W}_0^{(\ell)}\|_{\mathrm{F}} &\leq \sum_{k=0}^{t} \|\mathbf{W}_{k+1}^{(\ell)} - \mathbf{W}_k^{(\ell)}\|_{\mathrm{F}} \\
&= \eta \sum_{k=0}^{t} \left\| \frac{\partial \mathcal{L}(\boldsymbol{\theta})}{\partial \mathbf{W}_k^{(\ell)}} \right\|_{\mathrm{F}} \\
&\leq \eta s^L \lambda_{1 \to (L+1)} \lambda_\ell^{-1} \|\mathbf{X}\|_{\mathrm{F}} \sum_{k=0}^{t} \|\widehat{\mathbf{Y}}_k - \mathbf{Y}\|_{\mathrm{F}} \\
&\leq \eta s^L \lambda_{1 \to (L+1)} \lambda_\ell^{-1} \|\mathbf{X}\|_{\mathrm{F}} \sum_{k=0}^{t} \sqrt{2\mathcal{L}(\boldsymbol{\theta}_k)} \\
&\underset{(a)}{\leq} \eta s^L \lambda_{1 \to (L+1)} \lambda_\ell^{-1} \|\mathbf{X}\|_{\mathrm{F}} \sum_{k=0}^{t} \left(1 - \eta \frac{\alpha_0^2}{8}\right)^{k/2} \sqrt{2\mathcal{L}(\boldsymbol{\theta}_0)},
\end{aligned}
\tag{28}
$$

where inequality $(a)$ follows the induction condition.

Let $u = \sqrt{1 - \eta\alpha_0^2/8}$ we have $\eta = 8(1 - u^2)/\alpha_0^2$ and

$$
\sum_{k=0}^{t} \left(1 - \eta\frac{\alpha_0^2}{8}\right)^{k/2} = \sum_{k=0}^{t} u^k = \frac{1 - u^{t+1}}{1 - u}.
\tag{29}
$$

Then, plugging the result into Eq. 28, we have

$$
\begin{aligned}
\|\mathbf{W}_{t+1}^{(\ell)} - \mathbf{W}_0^{(\ell)}\|_{\mathrm{F}} &\leq s^L \lambda_{1 \to (L+1)} \lambda_\ell^{-1} \|\mathbf{X}\|_{\mathrm{F}} \frac{8(1 - u^2)}{\alpha_0^2} \frac{1 - u^{t+1}}{1 - u} \sqrt{2\mathcal{L}(\boldsymbol{\theta}_0)}, \\
&\underset{(a)}{\leq} \frac{16}{\alpha_0^2} s^L \lambda_{1 \to (L+1)} \lambda_\ell^{-1} \|\mathbf{X}\|_{\mathrm{F}} \sqrt{2\mathcal{L}(\boldsymbol{\theta}_0)},
\end{aligned}
\tag{30}
$$

where $(a)$ is due to $u \in (0, 1)$. Let $C_\ell$ as a positive constant that

$$
C_\ell \geq \frac{16}{\alpha_0^2} s^L \lambda_{1 \to (L+1)} \lambda_\ell^{-1} \|\mathbf{X}\|_{\mathrm{F}} \sqrt{2\mathcal{L}(\boldsymbol{\theta}_0)}.
\tag{31}
$$

By Wely's inequality, we have

$$
\|\mathbf{W}_{t+1}^{(\ell)}\|_2 \leq \|\mathbf{W}_0^{(\ell)}\|_2 + C_\ell = \lambda_\ell.
\tag{32}
$$

To prove the second inequality holds in Eq. 27, we first upper bound $\|\mathbf{H}_{t+1}^{(L)} - \mathbf{H}_0^{(L)}\|_{\mathrm{F}}$ by

$$
\begin{aligned}
\|\mathbf{H}_{t+1}^{(L)} - \mathbf{H}_0^{(L)}\|_{\mathrm{F}} &\underset{(a)}{\leq} s^L \lambda_{1 \to L} \|\mathbf{X}\|_{\mathrm{F}} \sum_{\ell=1}^{L} \lambda_\ell^{-1} \|\mathbf{W}_{t+1}^{(\ell)} - \mathbf{W}_0^{(\ell)}\|_2 \\
&\underset{(b)}{\leq} s^L \lambda_{1 \to L} \|\mathbf{X}\|_{\mathrm{F}} \sum_{\ell=1}^{L} \lambda_\ell^{-1} \left(\frac{16}{\alpha_0^2} s^L \lambda_{1 \to (L+1)} \lambda_\ell^{-1} \|\mathbf{X}\|_{\mathrm{F}} \sqrt{2\mathcal{L}(\boldsymbol{\theta}_0)}\right) \\
&= \frac{16}{\alpha_0^2} s^{2L} \lambda_{1 \to L} \lambda_{1 \to (L+1)} \|\mathbf{X}\|_{\mathrm{F}}^2 \sqrt{2\mathcal{L}(\boldsymbol{\theta}_0)} \left(\sum_{\ell=1}^{L} \lambda_\ell^{-2}\right) \\
&\underset{(c)}{\leq} \frac{\alpha_0}{2},
\end{aligned}
\tag{33}
$$

where $(a)$ is due to Lemma 2, $(b)$ is due to Eq. 30, and $(c)$ is due to the second condition in Eq. 24.

By Weyl's inequality, this implies $|\lambda_{\min}(\mathbf{H}_{t+1}^{(L)}) - \lambda_{\min}(\mathbf{H}_0^{(L)})| = |\lambda_{\min}(\mathbf{H}_{t+1}^{(L)}) - \alpha_0| \leq \frac{\alpha_0}{2}$ and

$$\lambda_{\min}(\mathbf{H}_{t+1}^{(L)}) \geq \frac{\alpha_0}{2}. \tag{34}$$

Let $\mathbf{G} = \mathbf{H}_t^{(L)}\mathbf{W}_{t+1}^{(L+1)}$, then we have

$$
\begin{aligned}
2\mathcal{L}(\boldsymbol{\theta}_{t+1}) &= \|\widehat{\mathbf{Y}}_{t+1} - \mathbf{Y}\|_{\mathrm{F}}^2 \\
&= \|\widehat{\mathbf{Y}}_{t+1} - \widehat{\mathbf{Y}}_t + \widehat{\mathbf{Y}}_t - \mathbf{Y}\|_{\mathrm{F}}^2 \\
&= \|\widehat{\mathbf{Y}}_t - \mathbf{Y}\|_{\mathrm{F}}^2 + \|\widehat{\mathbf{Y}}_{t+1} - \widehat{\mathbf{Y}}_t\|_{\mathrm{F}}^2 + 2\langle \widehat{\mathbf{Y}}_t - \mathbf{Y}, \widehat{\mathbf{Y}}_{t+1} - \widehat{\mathbf{Y}}_t \rangle_{\mathrm{F}} \\
&= 2\mathcal{L}(\boldsymbol{\theta}_t) + \|\widehat{\mathbf{Y}}_{t+1} - \widehat{\mathbf{Y}}_t\|_{\mathrm{F}}^2 + 2\langle \widehat{\mathbf{Y}}_t - \mathbf{Y}, \widehat{\mathbf{Y}}_{t+1} - \mathbf{G} \rangle_{\mathrm{F}} + 2\langle \widehat{\mathbf{Y}}_t - \mathbf{Y}, \mathbf{G} - \widehat{\mathbf{Y}}_t \rangle_{\mathrm{F}}.
\end{aligned} \tag{35}
$$

We can upper bound $\|\widehat{\mathbf{Y}}_{t+1} - \widehat{\mathbf{Y}}_t\|_{\mathrm{F}}^2$ in Eq. 35 by

$$
\begin{aligned}
\|\widehat{\mathbf{Y}}_{t+1} - \widehat{\mathbf{Y}}_t\|_{\mathrm{F}} &= \|\mathbf{H}_{t+1}^{(L)}\mathbf{W}_{t+1}^{(L+1)} - \mathbf{H}_t^{(L)}\mathbf{W}_t^{(L+1)}\|_{\mathrm{F}} \\
&= \|\mathbf{H}_{t+1}^{(L)}(\mathbf{W}_{t+1}^{(L+1)} - \mathbf{W}_t^{(L+1)}) + (\mathbf{H}_{t+1}^{(L)} - \mathbf{H}_t^{(L)})\mathbf{W}_t^{(L+1)}\|_{\mathrm{F}} \\
&\leq \|\mathbf{H}_{t+1}^{(L)}\|_{\mathrm{F}}\|\mathbf{W}_{t+1}^{(L+1)} - \mathbf{W}_t^{(L+1)}\|_2 + \|\mathbf{H}_{t+1}^{(L)} - \mathbf{H}_t^{(L)}\|_{\mathrm{F}}\|\mathbf{W}_t^{(L+1)}\|_2 \\
&\underset{(a)}{\leq} s^L \lambda_{1\to(L+1)}\|\mathbf{X}\|_{\mathrm{F}} \sum_{\ell=1}^{L} \lambda_\ell^{-1}\|\mathbf{W}_{t+1}^{(\ell)} - \mathbf{W}_t^{(\ell)}\|_2 + s^L \lambda_{1\to L}\|\mathbf{X}\|_{\mathrm{F}}\|\mathbf{W}_{t+1}^{(L+1)} - \mathbf{W}_t^{(L+1)}\|_2 \\
&= s^L \lambda_{1\to(L+1)}\|\mathbf{X}\|_{\mathrm{F}} \sum_{\ell=1}^{L+1} \lambda_\ell^{-1}\|\mathbf{W}_{t+1}^{(\ell)} - \mathbf{W}_t^{(\ell)}\|_2 \\
&\underset{(b)}{\leq} \eta s^L \lambda_{1\to(L+1)}\|\mathbf{X}\|_{\mathrm{F}} \sum_{\ell=1}^{L+1} \lambda_\ell^{-1}\left(\frac{s^L \lambda_{1\to(L+1)}}{\lambda_\ell}\|\mathbf{X}\|_{\mathrm{F}}\|\widehat{\mathbf{Y}}_t - \mathbf{Y}\|_{\mathrm{F}}\right) \\
&= \eta s^{2L} \lambda_{1\to(L+1)}^2\|\mathbf{X}\|_{\mathrm{F}}^2 \left(\sum_{\ell=1}^{L+1} \lambda_\ell^{-2}\right)\|\widehat{\mathbf{Y}}_t - \mathbf{Y}\|_{\mathrm{F}},
\end{aligned} \tag{36}
$$

where $(a)$ is due to Eq. 18 and Lemma 2, and $(b)$ follows from Lemma 3.

We can upper bound $\langle \widehat{\mathbf{Y}}_t - \mathbf{Y}, \widehat{\mathbf{Y}}_{t+1} - \mathbf{G} \rangle_{\mathrm{F}}$ in Eq. 35 by

$$
\begin{aligned}
&\langle \widehat{\mathbf{Y}}_t - \mathbf{Y}, \widehat{\mathbf{Y}}_{t+1} - \mathbf{G} \rangle_{\mathrm{F}} \\
&\leq \|\widehat{\mathbf{Y}}_t - \mathbf{Y}\|_{\mathrm{F}}\|\widehat{\mathbf{Y}}_{t+1} - \mathbf{G}\|_{\mathrm{F}} \\
&= \|\widehat{\mathbf{Y}}_t - \mathbf{Y}\|_{\mathrm{F}}\|(\mathbf{H}_{t+1}^{(L)} - \mathbf{H}_t^{(L)})\mathbf{W}_{t+1}^{(L+1)}\|_{\mathrm{F}} \\
&\underset{(a)}{\leq} \lambda_{L+1}\|\widehat{\mathbf{Y}}_t - \mathbf{Y}\|_{\mathrm{F}}\left(s^L \lambda_{1\to L}\|\mathbf{X}\|_{\mathrm{F}} \sum_{\ell=1}^{L} \lambda_\ell^{-1}\|\mathbf{W}_{t+1}^{(\ell)} - \mathbf{W}_t^{(\ell)}\|_2\right) \\
&= s^L \lambda_{1\to(L+1)}\|\widehat{\mathbf{Y}}_t - \mathbf{Y}\|_{\mathrm{F}}\|\mathbf{X}\|_{\mathrm{F}}\left(\sum_{\ell=1}^{L} \lambda_\ell^{-1}\|\mathbf{W}_{t+1}^{(\ell)} - \mathbf{W}_t^{(\ell)}\|_2\right) \\
&\underset{(b)}{\leq} \eta s^L \lambda_{1\to(L+1)}\|\widehat{\mathbf{Y}}_t - \mathbf{Y}\|_{\mathrm{F}}\|\mathbf{X}\|_{\mathrm{F}}\left(\sum_{\ell=1}^{L} \lambda_\ell^{-1} \frac{s^L \lambda_{1\to(L+1)}}{\lambda_\ell}\|\mathbf{X}\|_{\mathrm{F}}\|\widehat{\mathbf{Y}} - \mathbf{Y}\|_{\mathrm{F}}\right) \\
&= \eta s^{2L} \lambda_{1\to(L+1)}^2\|\widehat{\mathbf{Y}}_t - \mathbf{Y}\|_{\mathrm{F}}^2\|\mathbf{X}\|_{\mathrm{F}}^2 \left(\sum_{\ell=1}^{L} \lambda_\ell^{-2}\right),
\end{aligned} \tag{37}
$$

where $(a)$ is due to Lemma 2, and $(b)$ follows from Lemma 3.

We can upper bound $\langle \widehat{\mathbf{Y}}_t - \mathbf{Y}, \mathbf{G} - \widehat{\mathbf{Y}}_t \rangle_{\mathrm{F}}$ by

$$
\begin{aligned}
\langle \widehat{\mathbf{Y}}_t - \mathbf{Y}, \mathbf{G} - \widehat{\mathbf{Y}}_t \rangle_{\mathrm{F}} &= \langle \widehat{\mathbf{Y}}_t - \mathbf{Y}, \mathbf{H}_t^{(L)}(\mathbf{W}_{t+1}^{(L+1)} - \mathbf{W}_t^{(L+1)}) \rangle_{\mathrm{F}} \\
&= \eta \left\langle \widehat{\mathbf{Y}}_t - \mathbf{Y}, \mathbf{H}_t^{(L)} \frac{\partial \mathcal{L}(\boldsymbol{\theta})}{\partial \mathbf{W}_t^{(L+1)}} \right\rangle_{\mathrm{F}} \\
&= -\eta \left\langle \widehat{\mathbf{Y}}_t - \mathbf{Y}, \mathbf{H}_t^{(L)} \left( (\mathbf{H}_t^{(L)})^\top (\widehat{\mathbf{Y}}_t - \mathbf{Y}) \right) \right\rangle_{\mathrm{F}} \\
&\underset{(a)}{=} -\eta \operatorname{tr}\left( (\widehat{\mathbf{Y}}_t - \mathbf{Y})^\top \mathbf{H}_t^{(L)} (\mathbf{H}_t^{(L)})^\top (\widehat{\mathbf{Y}}_t - \mathbf{Y}) \right) \\
&\underset{(b)}{=} -\eta \operatorname{tr}\left( \mathbf{H}_t^{(L)} (\mathbf{H}_t^{(L)})^\top (\widehat{\mathbf{Y}}_t - \mathbf{Y})(\widehat{\mathbf{Y}}_t - \mathbf{Y})^\top \right) \\
&\underset{(c)}{\leq} -\eta \lambda_{\min}\left( \mathbf{H}_t^{(L)} (\mathbf{H}_t^{(L)})^\top \right) \operatorname{tr}\left( (\widehat{\mathbf{Y}}_t - \mathbf{Y})(\widehat{\mathbf{Y}}_t - \mathbf{Y})^\top \right) \\
&\underset{(d)}{=} -\eta \lambda_{\min}\left( \mathbf{H}_t^{(L)} (\mathbf{H}_t^{(L)})^\top \right) \|\widehat{\mathbf{Y}}_t - \mathbf{Y}\|_{\mathrm{F}}^2 \\
&\underset{(e)}{=} -\eta \lambda_{\min}^2\left( \mathbf{H}_t^{(L)} \right) \|\widehat{\mathbf{Y}}_t - \mathbf{Y}\|_{\mathrm{F}}^2, \\
&\underset{(f)}{\leq} -\eta \frac{\alpha_0^2}{4} \|\widehat{\mathbf{Y}}_t - \mathbf{Y}\|_{\mathrm{F}}^2,
\end{aligned}
\tag{38}
$$

where $(a)$ is due to $\langle \mathbf{A}, \mathbf{B} \rangle = \operatorname{tr}(\mathbf{A}^\top \mathbf{B})$, $(b)$ is because of the cyclic property of the trace, $(c)$ is due to $\operatorname{tr}(\mathbf{A}\mathbf{B}) \geq \lambda_{\min}(\mathbf{A})\operatorname{tr}(\mathbf{B})$ for any symmetric matrices $\mathbf{A}, \mathbf{B} \in \mathbb{S}$ and $\mathbf{B}$ is positive semi-definite, $(d)$ is due to $\operatorname{tr}(\mathbf{A}\mathbf{A}^\top) = \|\mathbf{A}\|_{\mathrm{F}}^2$, $(e)$ requires the $d_L \geq N$ assumption[9], and $(f)$ is due to Eq. 34.

Let $A = s^{2L} \lambda_{1 \to (L+1)}^2 \|\mathbf{X}\|_{\mathrm{F}}^2 \left( \sum_{\ell=1}^{L+1} \lambda_\ell^{-2} \right)$ and $B = s^{2L} \lambda_{1 \to (L+1)}^2 \|\mathbf{X}\|_{\mathrm{F}}^2 \left( \sum_{\ell=1}^{L} \lambda_\ell^{-2} \right)$, we have

$$
\begin{aligned}
\mathcal{L}(\boldsymbol{\theta}_{t+1}) &\leq \left( 1 - \eta \frac{\alpha_0^2}{4} + \eta^2 A^2 + \eta B \right) \mathcal{L}(\boldsymbol{\theta}_t) \\
&\underset{(a)}{\leq} \left( 1 - \eta \left( \frac{\alpha_0^2}{4} - 2B \right) \right) \mathcal{L}(\boldsymbol{\theta}_t) \\
&\underset{(b)}{\leq} \left( 1 - \eta \frac{\alpha_0^2}{8} \right) \mathcal{L}(\boldsymbol{\theta}_t),
\end{aligned}
\tag{39}
$$

where $(a)$ requires choosing $\eta$ such that $\eta \leq B/A^2$, and $(b)$ holds due to the first condition in Eq. 24. Therefore, we have

$$
\mathcal{L}(\boldsymbol{\theta}_T) \leq \left( 1 - \eta \frac{\alpha_0^2}{8} \right)^T \mathcal{L}(\boldsymbol{\theta}_0) \leq \exp\left( -\eta T \frac{\alpha_0^2}{8} \right) \mathcal{L}(\boldsymbol{\theta}_0) = \epsilon.
\tag{40}
$$

By taking log on both side, we have

$$
T \geq \frac{8}{\eta \alpha_0^2} \log\left( \frac{\mathcal{L}(\boldsymbol{\theta}_0)}{\epsilon} \right).
\tag{41}
$$

$\square$

---

[9]The equality $(e)$ follows from the fact that given a fat matrix $\mathbf{A} \in \mathbb{R}^{N \times d}$ with $d > N$, all the eigenvalues of $\mathbf{A}\mathbf{A}^\top$ are just the square of the singular values of $\mathbf{A}$. However, note that this is no longer true when $d < N$ since in this case we have $\mathbf{A}\mathbf{A}^\top$ being a low rank matrix, hence its smallest eigenvalue is zero, whereas the smallest singular value of $\mathbf{A}$ (i.e., the $\min(N, d)$-th singular value of $\mathbf{A}$) can be strictly positive if $\mathbf{A}$ is full rank, hence (e) does not hold.

# E   Additional empirical results

In this section, we report additional empirical results to illustrate the correctness of our theoretical analysis and the benefits of decoupled GCN structure.

## E.1   Comparison of generalization error on the real-world datasets.

We empirically compare the generalization error of different GCN structures on the real-world datasets, where the generalization error is measured by the difference between validation loss and training loss.

**Setups.** We select hidden dimension as 64, $\alpha_\ell = 0.9$ for APPNP and GCNII, $\beta_\ell \approx \frac{0.5}{\ell}$ for GCNII, and without any weight decay ($\ell_2$ regularization on weight matrices) or dropout operators. Note that although in our theoretical analysis we suppose $\beta_\ell$ is a constant that does not change with respect to the layers, in this experiment we follow the configuration in [6] by selecting $\beta_\ell = \log(\frac{0.5}{\ell} + 1)$, which guarantees a small generalization error on most small scale datasets. The same setup is also used for the results on synthetic dataset in Figure 4.

**Results.** As shown in Figure 11 and Figure 12, ResGCN has the largest generalization gap due to the skip-connections, APPNP has the smallest generalization error by removing the weight matrices in each individual layers. On the other hand, GCNII achieves a good balance between GCN and APPNP by balancing the expressive and generalization power. Finally, DGCN enjoys a small generalization error by using the decoupled GCN structure.

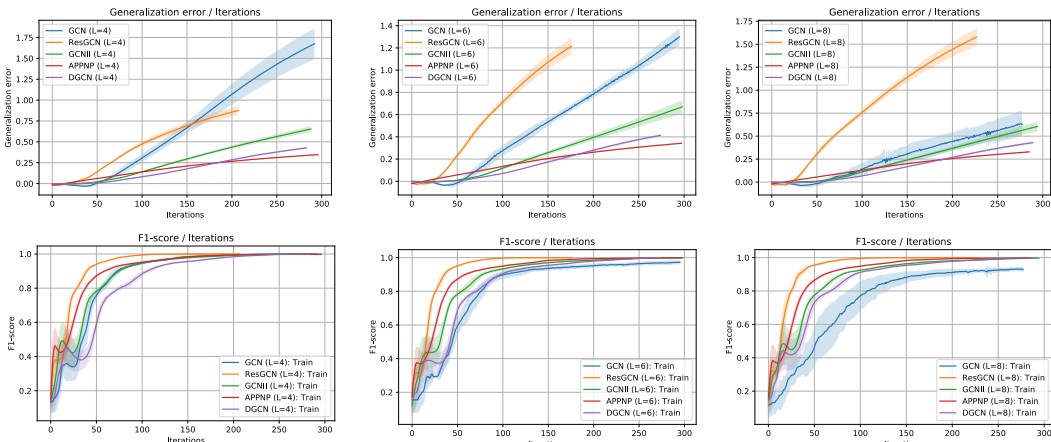

Figure 11: Comparison of generalization error and training F1-score on the *Cora* dataset. The curve stops early at the largest training accuracy iteration.

## E.2   Effect of hyper-parameters in GCNII.

In the following, we first compare the effect of $\alpha_\ell$ on the generalization error of GCNII. We select the hidden dimension as 64, and no dropout or weight decay is used for the training. As shown in Figure 13 and Figure 14, increasing $\alpha_\ell$ leads to a smaller generalization error but a slower convergence speed, i.e., GCNII trades off the expressiveness for generalization power by increasing $\alpha_\ell$ from 0 to 1. In practice, $\alpha_\ell = 0.9$ is utilized in GCNII [6] for empirical evaluations.[10]

Then, we compare the effect of $\beta_\ell$ on the generalization error of GCNII. Similar to the previous experiments, we choose hidden dimension as 64, without applying dropout or weight decay during training. As shown in Figure 15 and Figure 16, decreasing $\beta_\ell$ leads to a smaller generalization error. In practice, $\beta_\ell = \log(\frac{0.5}{\ell} + 1)$ is utilized in GCNII [6] for empirical evaluations, which guarantees a small generalization error.

---

[10]Note that the definition of $\alpha_\ell$ we use in this paper is different from the definition in [6], where $\alpha_\ell = 0.9$ in this paper stands for the selection of alpha as $1 - \alpha_\ell = 0.1$ in [6].

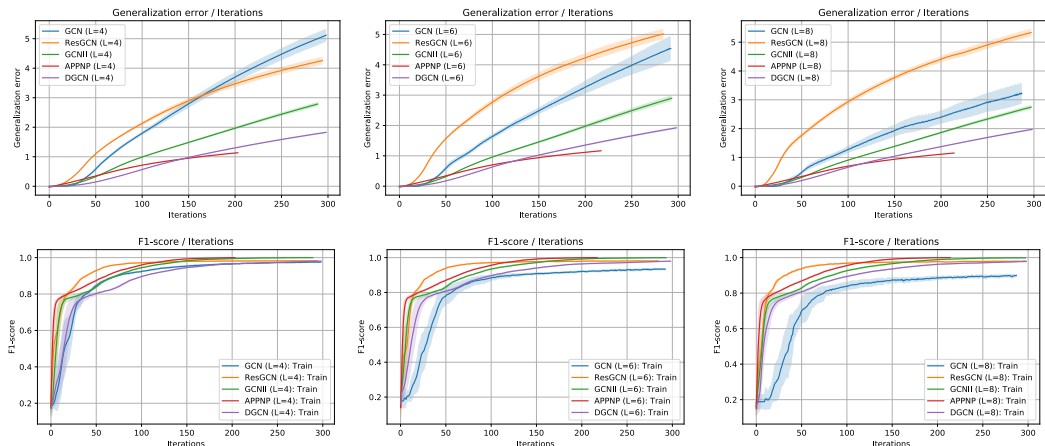

Figure 12: Comparison of generalization error and training F1-score on the *Citeseer* dataset. The curve stops early at the largest training accuracy iteration.

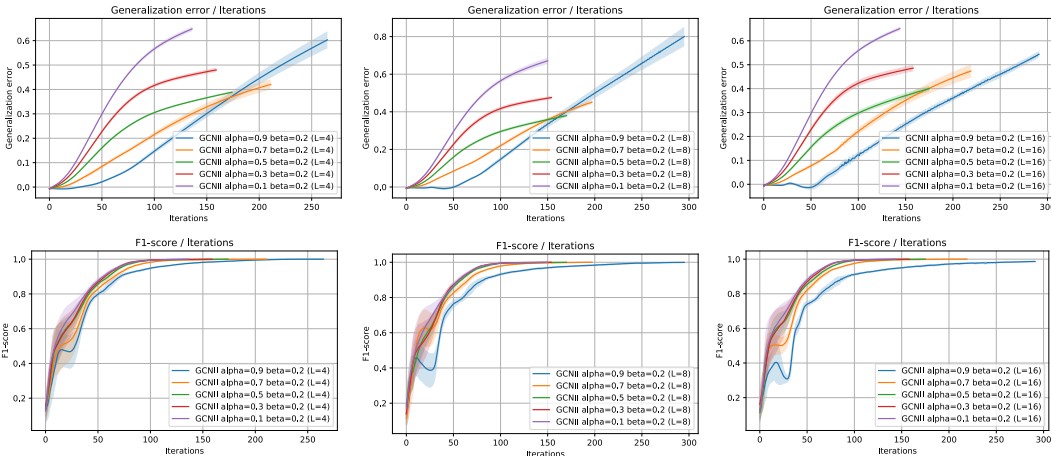

Figure 13: Comparison of $\alpha_\ell$ on the generalization error on *Cora* dataset. The curve stops early at the largest training accuracy iteration.

### E.3  Effect of DropEdge on generalization

In the following, we explore the effect of node embedding augmentation technique DropEdge [47] on the generalization of GCNs. Recall that DropEdge proposes to randomly drop a certain rate of edges in the input graph at each iteration and compute node embedding based on the sparser graph. The forward propagation rule of this technique can be formulated as $\mathbf{H}^{(\ell-1)} = \sigma(\widetilde{\mathbf{L}}\mathbf{H}^{(\ell-1)}\mathbf{W}^{(\ell)}))$ where $\widetilde{\mathbf{L}}$ is constructed by the adjacency matrix of the sparser graph with $\mathrm{supp}(\widetilde{\mathbf{L}}) \ll \mathrm{supp}(\mathbf{L})$.

Again, we choose hidden dimension as 64, without applying dropout or weight decay during training. As shown in Figure 17 and Figure 18, DropEdge reduces the generalization error by restricting the number of nodes used during training. Besides, we observe that both training accuracy and generalization error decrease when the fraction of remaining edges in the graph decreases, which implies that edge dropping is impacting the generalization of GCN rather than the discriminativeness of node representations.

### E.4  Effect of PairNorm on generalization

In the following, we explore the effect of node embedding augmentation technique PairNorm [61] on the generalization of GCNs. PairNorm proposes to normalized node embeddings by $\mathbf{H}^{(\ell)} =$

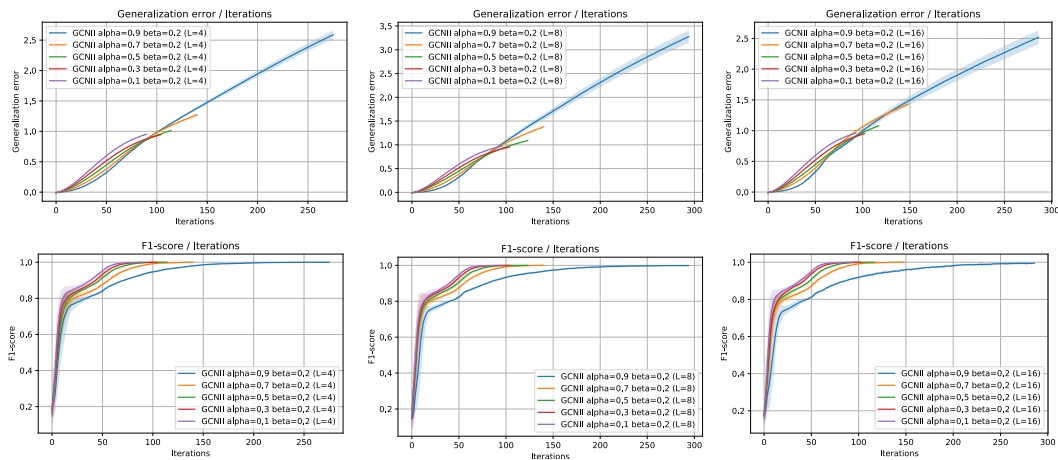

Figure 14: Comparison of $\alpha_\ell$ on the generalization error on *Citeseer* dataset. The curve stops early at the largest training accuracy iteration.

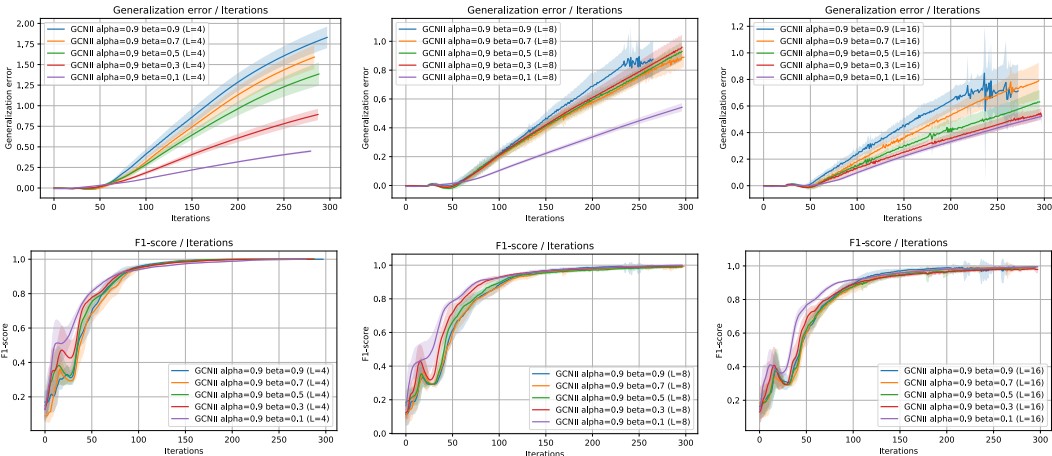

Figure 15: Comparison of $\beta_\ell$ on the generalization error on *Cora* dataset. The curve stops early at the largest training accuracy iteration.

$PN(\mathbf{H}^{(\ell)}) = \gamma \frac{\mathbf{H}^{(\ell)} - \boldsymbol{\mu}(\mathbf{H}^{(\ell)})}{\sigma(\mathbf{H}^{(\ell)})}$ where the average node embedding is computed as $\boldsymbol{\mu}(\mathbf{H}^{(\ell)}) = \frac{1}{N} \sum_{i=1}^{N} \mathbf{h}_i^{(\ell)}$, the variance of node embeddings is computed as $\sigma^2(\mathbf{H}^{(\ell)}) = \frac{1}{N} \sum_{i=1}^{N} \|\mathbf{h}_i^{(\ell)} - \boldsymbol{\mu}(\mathbf{H}^{(\ell)})\|_2^2$, and $\gamma \geq 0$ controls the scale of node embeddings.

We choose hidden dimension as 64, and no dropout or weight decay is used during training. As shown in Figure 19 and Figure 20, a larger scale ratio $\gamma$ can improve the discriminativeness of node embeddings, but will hurt the generalization error. A smaller scale ratio leads to a small generalization error, but it makes the node embeddings harder to discriminate, therefore over-smoothing happens (e.g., using $\gamma = 0.1$ can be think of as creating over-smoothing effect on node embeddings).

### E.5 Illustrating the gradient instability issue during GCN training

Gradient instability refers to a phenomenon that the gradient changes significantly at every iteration. The main reason for gradient instability is because the scale of weight matrices is large, which causes the calculated gradients become large. Notice that the gradient instability issue is more significant on vanilla GCN than other sequential GCN structures such as ResGCN and GCNII, which is one of the key factors that impacts the training phase of vanilla GCNs.

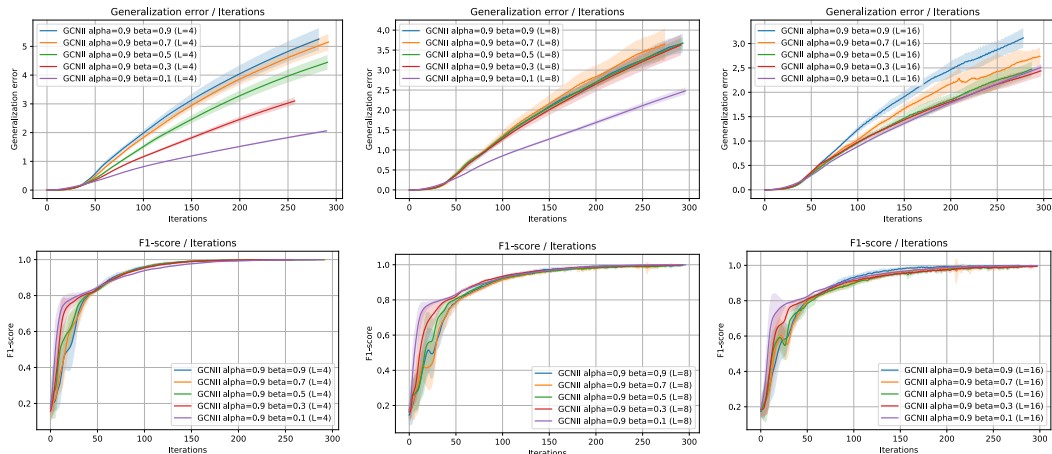

Figure 16: Comparison of $\beta_\ell$ on the generalization error on *Citeseer* dataset. The curve stops early at the largest training accuracy iteration.

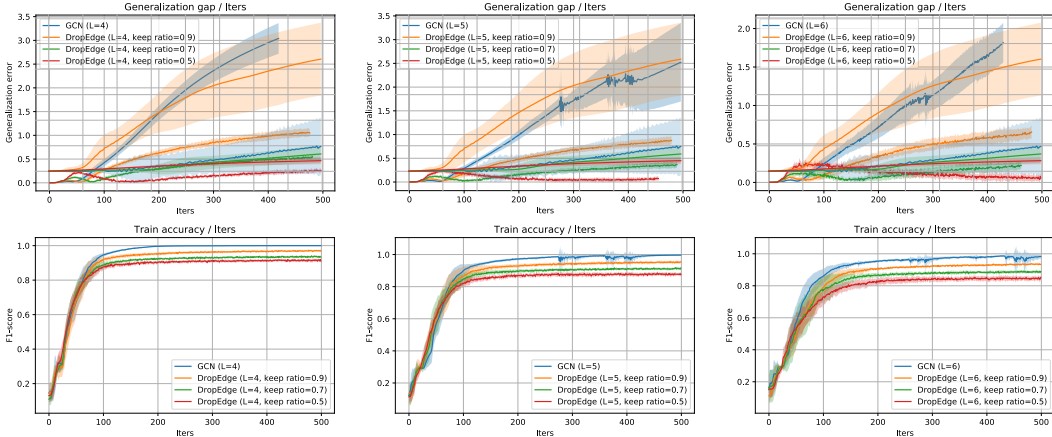

Figure 17: Comparison of generalization error of *DropEdge* on *Cora* dataset. The curve stops early at the largest training accuracy iteration.

To see this, let $\mathbf{W}^{(\ell)} \leftarrow \mathbf{W}^{(\ell)} - \eta \mathbf{G}^{(\ell)}$ denote the gradient descent update of the $\ell$th layer weight matrix, where $\mathbf{G}^{(\ell)}$ is the gradient with respect to the $\ell$th layer weight matrix $\mathbf{W}^{(\ell)}$. The upper bounds of the gradient for GCN, ResGCN, and GCNII are computed as

$$\|\mathbf{G}^{(\ell)}\|_2 = \mathcal{O}\Big((\max\{1, \sqrt{d}B_w\})^L\Big), \qquad \text{GCN (Eq. 71)}$$

$$\|\mathbf{G}^{(\ell)}\|_2 = \mathcal{O}\Big((1 + \sqrt{d}B_w)^L\Big), \qquad \text{ResGCN (Eq. 105)} \qquad (42)$$

$$\|\mathbf{G}^{(\ell)}\|_2 = \mathcal{O}\Big(\beta(\max\{1, \alpha\sqrt{d}B_w\})^L\Big), \quad \text{GCNII (Eq. 160)}$$

where $d$ is the largest number of node degree, $L$ is the number of layers, and $\|\mathbf{W}^{(\ell)}\|_2 \leq B_w$ is the largest singular value of weight matrix. Details please refer to the derivative of Eq. 71, Eq. 105, and Eq. 160.

From Eq. 42, we know that the largest singular value of weight matrices is the key factor that affects the scale of the gradient. However, upper bound can be vacuous if we simply ignore the impact of network structure on $B_w$.

From Figure 21 and Figure 22, we can observe that the residual connection has implicit regularization on the weight matrices, which makes the weight matrices in ResGCN has a smaller largest singular values than GCN. As a result, the ResGCN does not suffer from gradient instability even its gradient

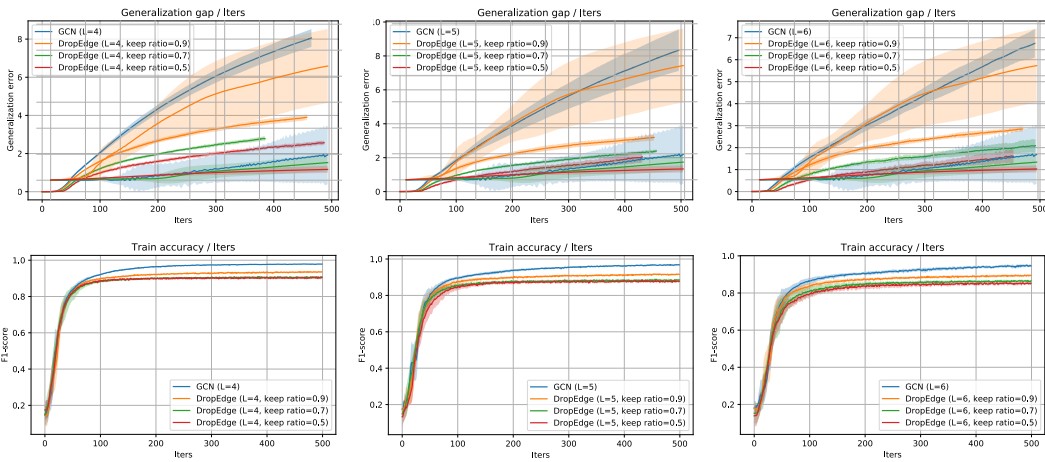

Figure 18: Comparison of generalization error of *DropEdge* on *Citeseer* dataset. The curve stops early at the largest training accuracy iteration.

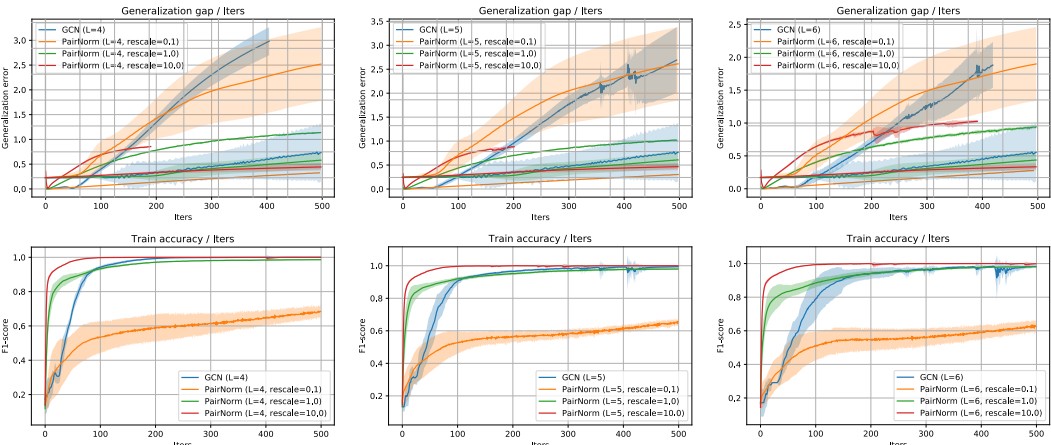

Figure 19: Comparison of generalization error of *PairNorm* on *Cora* dataset. The curve stops early at the largest training accuracy iteration.

norm upper bound in Eq. 42 is larger than GCN. Furthermore, although the largest singular value of the weight matrices for GCNII is larger than GCN, by selecting a small enough $\beta_\ell$, GCNII can be less impacted by gradient instability than vanilla GCN.

## E.6 Illustrating how more training leads to high training F1-score

As a compliment to Figure 1, we provide training and validation F1-score of the baseline models. During training, we chose hidden dimension as $64$, Adam optimizer with learning rate $0.001$, without any dropout or weight decay. Please note that removing dropout and weight decay is necessary because both operations are designed to prevent neural networks from overfitting, and will hurt the best training accuracy that a model can achieve. As shown in Figure 23 and Figure 24, all methods can achieve high training F1-score regardless the number of layers, which indicates node embeddings are distinguishable.

## E.7 Effect of number of layers on real-word datasets

In the following, we demonstrate the effect of the number of layers and hyper-parameters on the performance of the model on OGB Arxiv [24]. We follow the default hyper-parameter setup of GCN

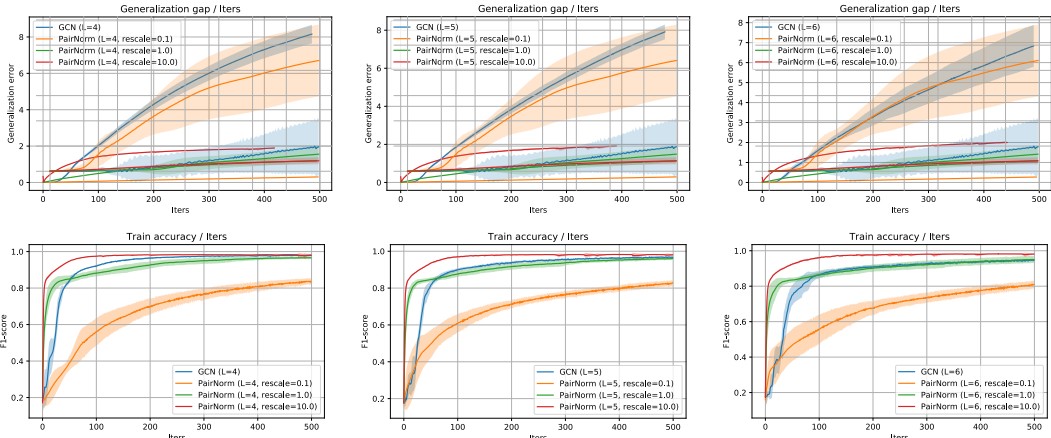

Figure 20: Comparison of generalization error of *PairNorm* on *Citeseer* dataset. The curve stops early at the largest training accuracy iteration.

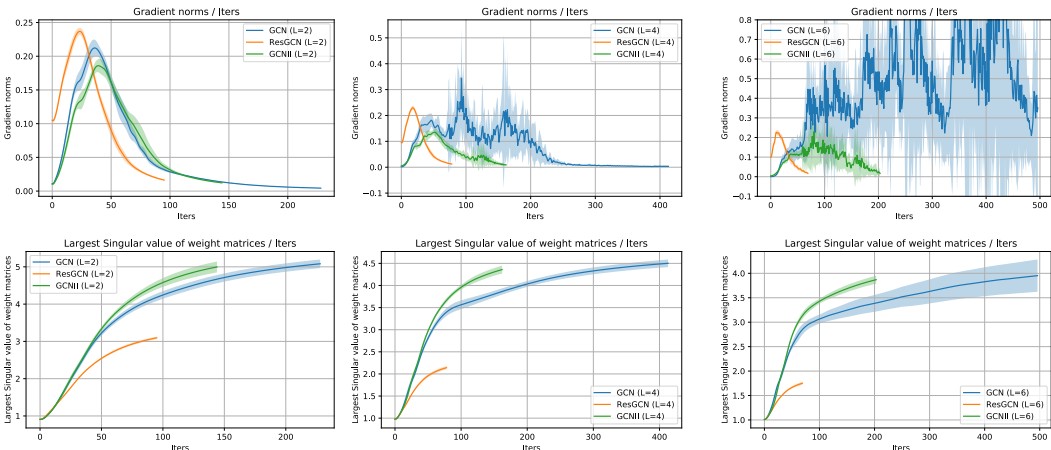

Figure 21: Comparison of gradient norm on *Cora* dataset. The curve stops early at the largest training accuracy iteration.

on the leaderboard,[11] i.e., we choose hidden dimension as $128$, dropout ratio as $0.5$, Adam optimizer with learning rate as $0.01$, and applying batch normalization after each graph convolutional layer. As shown in Table 3, the number of layers and the choice of hyper-parameters can largely impact the performance of the models. Since DGCN can automatically adjust the $\alpha_\ell$ and $\beta_\ell$ to better adapt to the change of model depth, it achieves a comparable and more stable performance than most baseline models.

## F   Generalization bound for GCN

In this section, we provide detailed proof on the generalization bound of GCN. Recall that the update rule of GCN is defined as

$$\mathbf{H}^{(\ell)} = \sigma(\mathbf{L}\mathbf{H}^{(\ell-1)}\mathbf{W}^{(\ell)}), \tag{43}$$

where $\sigma(\cdot)$ is the ReLU activation function. Note that although ReLU function $\sigma(x)$ is not differentiable when $x = 0$, for analysis purpose we suppose the $\sigma'(0) = 0$.[12]

---

[11]https://github.com/snap-stanford/ogb/blob/master/examples/nodeproppred/arxiv/gnn.py. Due to the memory limitation, we choose hidden dimension as $128$ instead of the default $256$ hidden dimension for all models.

[12]Widely used deep learning frameworks, including PyTorch and Tensorflow, also set the subgradient of ReLU as zero when its input is zero.

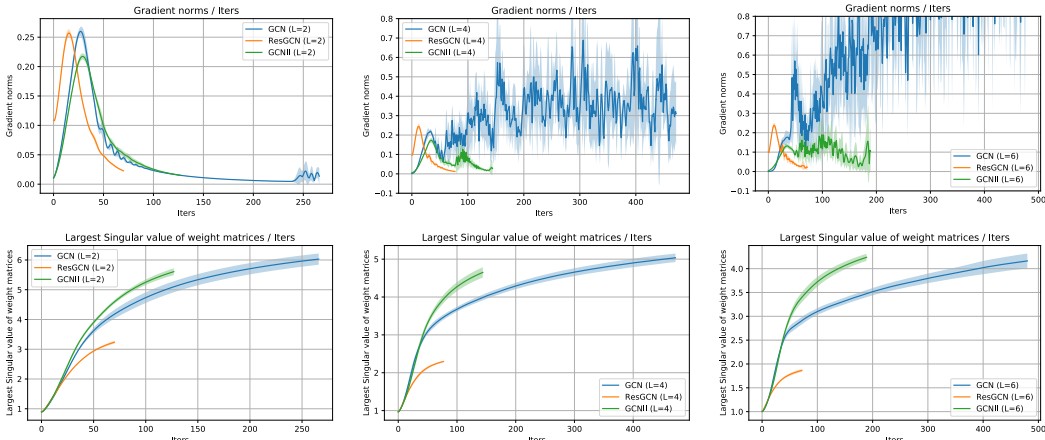

Figure 22: Comparison of gradient norm on *Citeseer* dataset. The curve stops early at the largest training accuracy iteration.

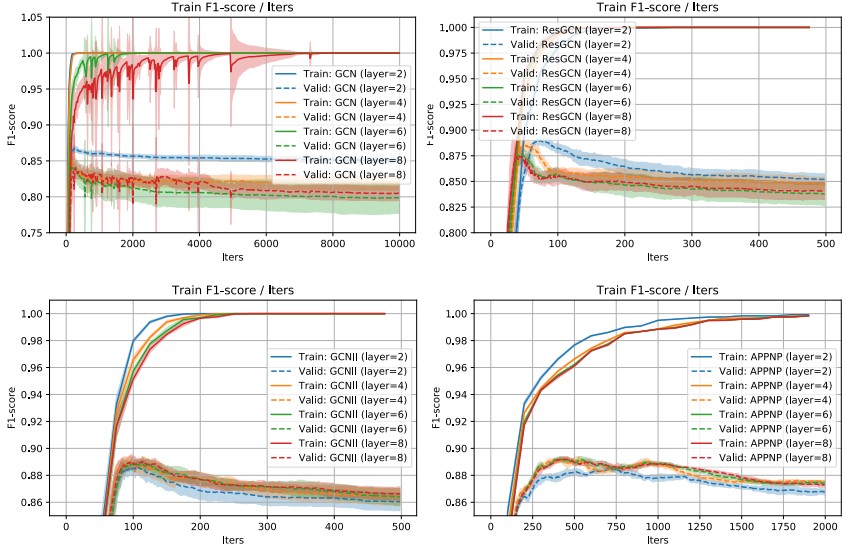

Figure 23: Comparison of training F1-score and number of iterations on *Cora* dataset.

The training of GCN is an empirical risk minimization with respect to a set of parameters $\boldsymbol{\theta} = \{\mathbf{W}^{(1)}, \ldots, \mathbf{W}^{(L)}, \mathbf{v}\}$, i.e.,

$$\mathcal{L}(\boldsymbol{\theta}) = \frac{1}{m} \sum_{i=1}^{m} \Phi_\gamma(-p(f(\mathbf{h}_i^{(L)}), y_i)), \ f(\mathbf{h}_i^{(L)}) = \tilde{\sigma}(\mathbf{v}^\top \mathbf{h}_i^{(L)}), \tag{44}$$

where $\mathbf{h}_i^{(L)}$ is the node representation of the $i$th node at the final layer, $f(\mathbf{h}_i^{(L)})$ is the predicted label for the $i$th node, $\tilde{\sigma}(x) = \frac{1}{\exp(-x)+1}$ is the sigmoid function, and loss function $\Phi_\gamma(-p(z, y))$ is $\frac{2}{\gamma}$-Lipschitz continuous with respect to its first input $z$ with $p(z, y)$ as defined in Section 5. For simplification, we will use $\text{Loss}(z, y)$ which represents $\Phi_\gamma(-p(z, y))$ in the proof.

To establish the generalization of GCN as stated in Theorem 3, we utilize the following result on transductive uniform stability from [17].

**Theorem 8** (Transductive uniform stability bound [17]). *Let $f$ be a $\epsilon$-uniformly stable transductive learner and $\gamma, \delta > 0$, and define $Q = mu/(m + u)$. Then, with probability at least $1 - \delta$ over all*

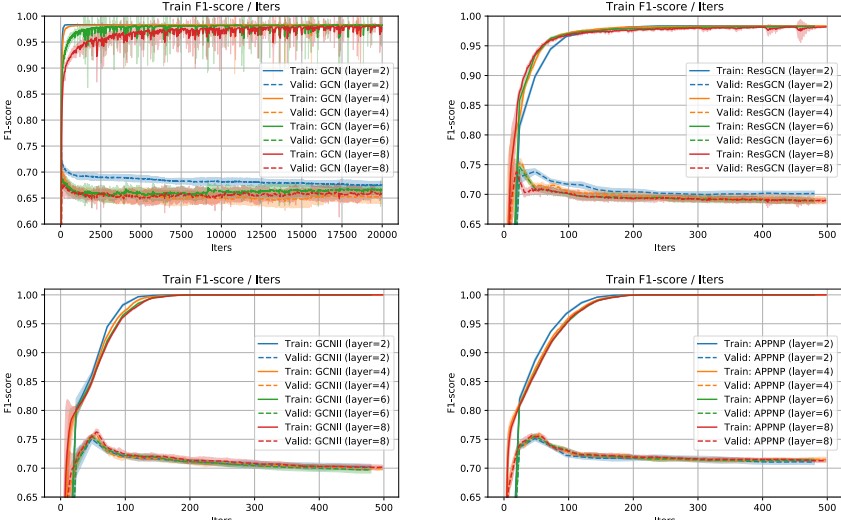

Figure 24: Comparison of training F1-score and number of iterations on *Citeseer* dataset.

Table 3: Comparison of F1-score on OGB-Arxiv dataset for different number of layers

| Model | $\alpha$ | 2 Layers | 4 Layers | 8 Layers | 12 Layers | 16 Layers |
|---|---|---|---|---|---|---|
| **GCN** | $-$ | $71.02\% \pm 0.14$ | $71.56\% \pm 0.19$ | $71.28\% \pm 0.33$ | $70.28\% \pm 0.23$ | $69.37\% \pm 0.46$ |
| **ResGCN** | $-$ | $70.66\% \pm 0.48$ | $72.41\% \pm 0.31$ | $72.56\% \pm 0.31$ | $72.46\% \pm 0.23$ | $72.11\% \pm 0.28$ |
| **GCNII** | $0.9$ | $71.35\% \pm 0.21$ | $72.57\% \pm 0.23$ | $72.06\% \pm 0.42$ | $71.31\% \pm 0.62$ | $69.99\% \pm 0.80$ |
| **GCNII** | $0.8$ | $71.14\% \pm 0.27$ | $72.32\% \pm 0.19$ | $71.90\% \pm 0.41$ | $71.21\% \pm 0.23$ | $70.56\% \pm 0.72$ |
| **GCNII** | $0.5$ | $70.54\% \pm 0.30$ | $72.09\% \pm 0.25$ | $71.92\% \pm 0.32$ | $71.24\% \pm 0.47$ | $71.02\% \pm 0.58$ |
| **APPNP** | $0.9$ | $67.38\% \pm 0.34$ | $68.02\% \pm 0.55$ | $66.62\% \pm 0.48$ | $67.43\% \pm 0.50$ | $67.42\% \pm 1.00$ |
| **APPNP** | $0.8$ | $66.71\% \pm 0.32$ | $68.25\% \pm 0.43$ | $66.40\% \pm 0.89$ | $66.51\% \pm 2.09$ | $66.56\% \pm 0.74$ |
| **DGCN** | $-$ | $71.21\% \pm 0.25$ | $72.29\% \pm 0.18$ | $72.39\% \pm 0.21$ | $\mathbf{72.63\% \pm 0.12}$ | $72.41\% \pm 0.07$ |

*training and testing partitions, we have*

$$\mathcal{R}_u(f) \le \mathcal{R}_m^\gamma(f) + \frac{2}{\gamma}\mathcal{O}\Big(\epsilon\sqrt{Q\ln(\delta^{-1})}\Big) + \mathcal{O}\Big(\frac{\ln(\delta^{-1})}{\sqrt{Q}}\Big).$$

Then, in Lemma 4, we derive the uniform stability constant for GCN, i.e., $\epsilon_{\text{GCN}}$.

**Lemma 4.** *The uniform stability constant for GCN is computed as* $\epsilon_{GCN} = \frac{2\eta\rho_f G_f}{m}\sum_{t=1}^{T}(1+\eta L_F)^{t-1}$
*where*

$$\rho_f = C_1^L C_2, \ G_f = \frac{2}{\gamma}(L+1)C_1^L C_2, \ L_f = \frac{2}{\gamma}(L+1)C_1^L C_2\Big((L+2)C_1^L C_2 + 2\Big),$$
$$C_1 = \max\{1, \sqrt{d}B_w\}, \ C_2 = \sqrt{d}(1+B_x). \tag{45}$$

By plugging the result in Lemma 4 back to Theorem 8, we establish the generalization bound for GCN.

The key idea of the proof is to decompose the change of the GCN output into two terms (in Lemma 5) which depend on

- (Lemma 6) The maximum change of node embeddings, i.e., $\Delta h_{\max}^{(\ell)} = \max_i \|[\mathbf{H}^{(\ell)} - \tilde{\mathbf{H}}^{(\ell)}]_{i,:}\|_2$,

- (Lemma 7) The maximum node embeddings, i.e., $h_{\max}^{(\ell)} = \max_i \|[\mathbf{H}^{(\ell)}]_{i,:}\|_2$.

**Lemma 5.** *Let* $f(\mathbf{h}_i^{(L)}) = \tilde{\sigma}(\mathbf{v}^\top \mathbf{h}_i^{(L)}), \tilde{f}(\tilde{\mathbf{h}}_i^{(L)}) = \tilde{\sigma}(\tilde{\mathbf{v}}^\top \tilde{\mathbf{h}}_i^{(L)})$ *denote the prediction of node* $i$ *using parameters* $\boldsymbol{\theta} = \{\mathbf{W}^{(1)}, \dots, \mathbf{W}^{(L)}, \mathbf{v}\}, \tilde{\boldsymbol{\theta}} = \{\tilde{\mathbf{W}}^{(1)}, \dots, \tilde{\mathbf{W}}^{(L)}, \tilde{\mathbf{v}}\}$ *(i.e., the two set of parameters*

*trained on the original and the perturbed dataset) respectively. Then we have*

$$\max_i |f(\mathbf{h}_i^{(L)}) - \tilde{f}(\tilde{\mathbf{h}}_i^{(L)})| \leq \Delta h_{\max}^{(L)} + h_{\max}^{(L)} \|\Delta \mathbf{v}\|_2,$$

$$\max_i \left\| \frac{\partial f(\mathbf{h}_i^{(L)})}{\partial \mathbf{h}_i^{(L)}} - \frac{\partial \tilde{f}(\tilde{\mathbf{h}}_i^{(L)})}{\partial \tilde{\mathbf{h}}_i^{(L)}} \right\|_2 \leq \Delta h_{\max}^{(L)} + (h_{\max}^{(L)} + 1)\|\Delta \mathbf{v}\|_2, \tag{46}$$

*where $\Delta \mathbf{v} = \mathbf{v} - \tilde{\mathbf{v}}$.*

**Lemma 6** (Upper bound of $h_{\max}^{(\ell)}$ for **GCN**). *Let suppose Assumption 1 hold. Then, the maximum node embeddings for any node at the $\ell$th layer is bounded by*

$$h_{\max}^{(\ell)} \leq B_x (\max\{1, \sqrt{d} B_w\})^\ell. \tag{47}$$

**Lemma 7** (Upper bound of $\Delta h_{\max}^{(\ell)}$ for **GCN**). *Let suppose Assumption 1 hold. Then, the maximum change between the node embeddings on two different set of weight parameters for any node at the $\ell$th layer is bounded by*

$$\Delta h_{\max}^{(\ell)} \leq \sqrt{d} B_x (\max\{1, \sqrt{d} B_w\})^{\ell-1} (\|\Delta \mathbf{W}^{(1)}\|_2 + \ldots + \|\Delta \mathbf{W}^{(\ell)}\|_2), \tag{48}$$

*where $\Delta \mathbf{W}^{(\ell)} = \mathbf{W}^{(\ell)} - \tilde{\mathbf{W}}^{(\ell)}$.*

Besides, in Lemma 8, we derive the upper bound on the maximum change of node embeddings before the activation function, i.e., $\Delta z_{\max}^{(\ell)} = \max_i \|[\mathbf{Z}^{(\ell)} - \tilde{\mathbf{Z}}^{(\ell)}]_{i,:}\|_2$. $\Delta z_{\max}^{(\ell)}$ will be used in the computation of the gradient related upper bounds.

**Lemma 8** (Upper bound of $\Delta z_{\max}^{(\ell)}$ for **GCN**). *Let suppose Assumption 1 hold. Then, the maximum change between the node embeddings before the activation function on two different set of weight parameters for any node at the $\ell$th layer is bounded by*

$$\Delta z_{\max}^{(\ell)} \leq \sqrt{d} B_x (\max\{1, \sqrt{d} B_w\})^{\ell-1} \Big( \|\Delta \mathbf{W}^{(1)}\|_2 + \ldots + \|\Delta \mathbf{W}^{(\ell)}\|_2 \Big), \tag{49}$$

*where $\Delta \mathbf{W}^{(\ell)} = \mathbf{W}^{(\ell)} - \tilde{\mathbf{W}}^{(\ell)}$.*

Then, in Lemma 9, we decompose the change of the model parameters into two terms which depend on

- The maximum change of gradient passing from the $(\ell+1)$th layer to the $\ell$th layer $\Delta d_{\max}^{(\ell)} = \max_i \left\| \left[ \frac{\partial \sigma(\mathbf{LH}^{(\ell-1)} \mathbf{W}^{(\ell)})}{\partial \mathbf{H}^{(\ell-1)}} - \frac{\partial \sigma(\mathbf{L}\tilde{\mathbf{H}}^{(\ell-1)} \tilde{\mathbf{W}}^{(\ell)})}{\partial \tilde{\mathbf{H}}^{(\ell-1)}} \right]_{i,:} \right\|_2$,

- The maximum gradient passing from the $(\ell+1)$th layer to the $\ell$th layer $d_{\max}^{(\ell)} = \max_i \left\| \left[ \frac{\partial \sigma(\mathbf{LH}^{(\ell-1)} \mathbf{W}^{(\ell)})}{\partial \mathbf{H}^{(\ell-1)}} \right]_{i,:} \right\|_2$.

**Lemma 9** (Upper bound of $d_{\max}^{(\ell)}, \Delta d_{\max}^{(\ell)}$ for **GCN**). *Let suppose Assumption 1 hold. Then, the upper bound on the maximum gradient passing from layer $\ell + 1$ to layer $\ell$ for any node is*

$$d_{\max}^{(\ell)} \leq \frac{2}{\gamma} (\max\{1, \sqrt{d} B_w\})^{L-\ell+1}, \tag{50}$$

*and the upper bound on the maximum change between the gradient passing from layer $\ell + 1$ to layer $\ell$ on two different set of weight parameters for any node is*

$$\Delta d_{\max}^{(\ell)} \leq (\max\{1, \sqrt{d} B_w\})^{L-\ell} \frac{2}{\gamma} \Big( (L+1)\sqrt{d}(B_x+1)(\max\{1, \sqrt{d} B_w\})^L + 1 \Big) \|\Delta \boldsymbol{\theta}\|_2, \tag{51}$$

*where $\|\Delta \boldsymbol{\theta}\|_2 = \|\mathbf{v} - \tilde{\mathbf{v}}\|_2 + \sum_{\ell-1}^L \|\mathbf{W}^{(\ell)} - \tilde{\mathbf{W}}^{(\ell)}\|_2$ denotes the change of two set of parameters.*

Finally, based on the previous result, in Lemma 10, we decompose the change of the model parameters into two terms which depend on

- The change of gradient with respect to the $\ell$th layer weight parameters $\|\Delta \mathbf{G}^{(\ell)}\|_2$,

- The gradient with respect to the $\ell$th layer weight parameters $\|\mathbf{G}^{(\ell)}\|_2$,

where $\mathbf{G}^{(L+1)}$ denotes the gradient with respect to the weight $\mathbf{v}$ of the binary classifier and $\mathbf{G}^{(\ell)}$ denotes the gradient with respect to the weight $\mathbf{W}^{(\ell)}$ of the $\ell$th graph convolutional layer. Notice that $\|\Delta\mathbf{G}^{(\ell)}\|_2$ reflects the smoothness of GCN model and $\|\mathbf{G}^{(\ell)}\|_2$ corresponds to the upper bound of gradient.

**Lemma 10** (Upper bound of $\|\mathbf{G}^{(\ell)}\|_2, \|\Delta\mathbf{G}^{(\ell)}\|_2$ for **GCN**). *Let suppose Assumption 1 hold and let $C_1 = \max\{1, \sqrt{d}B_w\}$ and $C_2 = \sqrt{d}(B_x + 1)$. Then, the gradient and the maximum change between gradients computed on two different set of weight parameters are bounded by*

$$
\begin{aligned}
\sum_{\ell=1}^{L+1} \|\mathbf{G}^{(\ell)}\|_2 &\le \frac{2}{\gamma}(L+1)C_1^L C_2 \|\Delta\boldsymbol{\theta}\|_2, \\
\sum_{\ell=1}^{L+1} \|\Delta\mathbf{G}^{(\ell)}\|_2 &\le \frac{2}{\gamma}(L+1)C_1^L C_2 \Big((L+2)C_1^L C_2 + 2\Big)\|\Delta\boldsymbol{\theta}\|_2,
\end{aligned}
\tag{52}
$$

*where $\|\Delta\boldsymbol{\theta}\|_2 = \|\mathbf{v} - \tilde{\mathbf{v}}\|_2 + \sum_{\ell-1}^{L} \|\mathbf{W}^{(\ell)} - \tilde{\mathbf{W}}^{(\ell)}\|_2$ denotes the change of two set of parameters.*

Equipped with above intermediate results, we now proceed to prove Lemma 4.

*Proof of Lemma 4.* Recall that our goal is to explore the impact of different GCN structures on the uniform stability constant $\epsilon_{\text{GCN}}$, which is a function of $\rho_f$, $G_f$, and $L_f$. Let $C_1 = \max\{1, \sqrt{d}B_w\}$ and $C_2 = \sqrt{d}(B_x + 1)$. Firstly, by plugging Lemma 7 and Lemma 6 into Lemma 5, we have

$$
\begin{aligned}
\max_i |f(\mathbf{h}_i^{(L)}) - \tilde{f}(\tilde{\mathbf{h}}_i^{(L)})| &\le \Delta h_{\max}^{(L)} + h_{\max}^{(L)}\|\Delta\mathbf{v}\|_2 \\
&\le \sqrt{d}B_x \cdot (\max\{1, \sqrt{d}B_w\})^L \|\Delta\boldsymbol{\theta}\|_2 \\
&\le C_1^L C_2 \|\Delta\boldsymbol{\theta}\|_2.
\end{aligned}
\tag{53}
$$

Therefore, we know that the function $f$ is $\rho_f$-Lipschitz continuous, with $\rho_f = C_1^L C_2$. Then, by Lemma 10, we know that the function $f$ is $L_f$-smoothness, and the gradient of each weight matrix is bounded by $G_f$, with

$$
G_f = \frac{2}{\gamma}(L+1)C_1^L C_2, \quad L_f = \frac{2}{\gamma}(L+1)C_1^L C_2\Big((L+2)C_1^L C_2 + 2\Big).
\tag{54}
$$

By plugging $\epsilon_{\text{GCN}}$ into Theorem 3, we obtain the generalization bound of GCN. $\qquad\square$

### F.1 Proof of Lemma 5

By the definition of $f(\mathbf{h}_i^{(L)})$ and $\tilde{f}(\tilde{\mathbf{h}}_i^{(L)})$, we have

$$
\begin{aligned}
\max_i |f(\mathbf{h}_i^{(L)}) - \tilde{f}(\tilde{\mathbf{h}}_i^{(L)})| &= \max_i |\tilde{\sigma}(\mathbf{v}^\top \mathbf{h}_i^{(L)}) - \tilde{\sigma}(\tilde{\mathbf{v}}^\top \tilde{\mathbf{h}}_i^{(L)})| \\
&\underset{(a)}{\le} \max_i \|\mathbf{v}^\top \mathbf{h}_i^{(L)} - \tilde{\mathbf{v}}^\top \tilde{\mathbf{h}}_i^{(L)}\|_2 \\
&\le \max_i \|\mathbf{v}^\top (\mathbf{h}_i^{(L)} - \tilde{\mathbf{h}}_i^{(L)})\|_2 + \max_i \|\tilde{\mathbf{h}}_i^{(L)}(\mathbf{v} - \tilde{\mathbf{v}})\|_2 \\
&\le \Delta h_{\max}^{(L)} + h_{\max}^{(L)}\Delta\mathbf{v},
\end{aligned}
\tag{55}
$$

where $(a)$ is due to the fact that sigmoid function is 1-Lipschitz continuous.

$$
\begin{aligned}
\max_i \left\| \frac{\partial f(\mathbf{h}_i^{(L)})}{\partial \mathbf{h}_i^{(L)}} - \frac{\partial \tilde{f}(\tilde{\mathbf{h}}_i^{(L)})}{\partial \tilde{\mathbf{h}}_i^{(L)}} \right\|_2 &= \max_i \|\tilde{\sigma}'(\mathbf{v}^\top \mathbf{h}_i^{(L)})\mathbf{v}^\top - \tilde{\sigma}'(\tilde{\mathbf{v}}^\top \tilde{\mathbf{h}}_i^{(L)})\tilde{\mathbf{v}}^\top\|_2 \\
&\leq \max_i \|\tilde{\sigma}'(\mathbf{v}^\top \mathbf{h}_i^{(L)})\mathbf{v}^\top - \tilde{\sigma}'(\mathbf{v}^\top \mathbf{h}_i^{(L)})\tilde{\mathbf{v}}^\top\|_2 \\
&\quad + \max_i \|\tilde{\sigma}'(\mathbf{v}^\top \mathbf{h}_i^{(L)})\tilde{\mathbf{v}}^\top - \tilde{\sigma}'(\tilde{\mathbf{v}}^\top \tilde{\mathbf{h}}_i^{(L)})\tilde{\mathbf{v}}^\top\|_2 \\
&\underset{(a)}{\leq} \Delta \mathbf{v} + \left( \Delta h_{\max}^{(L)} + h_{\max}^{(L)} \Delta \mathbf{v} \right) \\
&= \Delta h_{\max}^{(L)} + (h_{\max}^{(L)} + 1)\Delta \mathbf{v},
\end{aligned}
\tag{56}
$$

where $(a)$ is due to the fact that sigmoid function and its gradient are 1-Lipschitz continuous.

## F.2  Proof of Lemma 6

By the definition of $h_{\max}^{(\ell)}$, we have

$$
\begin{aligned}
h_{\max}^{(\ell)} &= \max_i \|[\sigma(\mathbf{L}\mathbf{H}^{(\ell-1)}\mathbf{W}^{(\ell)})]_{i,:}\|_2 \\
&\underset{(a)}{\leq} \max_i \|[\mathbf{L}\mathbf{H}^{(\ell-1)}\mathbf{W}^{(\ell)}]_{i,:}\|_2 \\
&\leq \max_i \|[\mathbf{L}\mathbf{H}^{(\ell-1)}]_{i,:}\|_2 \|\mathbf{W}^{(\ell)}\|_2 \\
&= \max_i \left\| \sum_{j=1}^{N} L_{i,j} \mathbf{h}_j^{(\ell-1)} \right\|_2 \|\mathbf{W}^{(\ell)}\|_2 \\
&\leq \max_i \| \sum_{j=1}^{N} L_{i,j}\|_2 \cdot \max_j \|\mathbf{h}_j^{(\ell-1)}\|_2 \cdot \|\mathbf{W}^{(\ell)}\|_2 \\
&\underset{(b)}{\leq} \sqrt{d}\|\mathbf{W}^{(\ell)}\|_2 \cdot h_{\max}^{(\ell-1)} \\
&\leq \sqrt{d}B_w \cdot h_{\max}^{(\ell-1)} \leq (\max\{1, \sqrt{d}B_w\})^\ell B_x,
\end{aligned}
\tag{57}
$$

where $(a)$ is due to $\|\sigma(\mathbf{x})\|_2 \leq \|\mathbf{x}\|_2$ and $(b)$ is due to Lemma 31.

## F.3  Proof of Lemma 7

By the definition of $\Delta h_{\max}^{(\ell)}$, we have

$$
\begin{aligned}
\Delta h_{\max}^{(\ell)} &= \max_i \|\mathbf{h}_i^{(\ell)} - \tilde{\mathbf{h}}_i^{(\ell)}\|_2 \\
&= \max_i \|[\sigma(\mathbf{L}\mathbf{H}^{(\ell-1)}\mathbf{W}^{(\ell)}) - \sigma(\mathbf{L}\tilde{\mathbf{H}}^{(\ell-1)}\tilde{\mathbf{W}}^{(\ell)})]_{i,:}\|_2 \\
&= \max_i \|[\mathbf{L}\mathbf{H}^{(\ell-1)}\mathbf{W}^{(\ell)} - \mathbf{L}\tilde{\mathbf{H}}^{(\ell-1)}\tilde{\mathbf{W}}^{(\ell)}]_{i,:}\|_2 \\
&\leq \max_i \|[\mathbf{L}\mathbf{H}^{(\ell-1)}(\mathbf{W}^{(\ell)} - \tilde{\mathbf{W}}^{(\ell)}) + \mathbf{L}(\mathbf{H}^{(\ell-1)} - \tilde{\mathbf{H}}^{(\ell-1)})\tilde{\mathbf{W}}^{(\ell)}]_{i,:}\|_2 \\
&\leq \max_i \left\| \sum_{j=1}^{N} L_{i,j} \mathbf{h}_j^{(\ell-1)} \right\|_2 \|\Delta\mathbf{W}^{(\ell)}\|_2 + \max_i \left\| \sum_{j=1}^{N} L_{i,j}\Delta\mathbf{h}_j^{(\ell-1)} \right\|_2 \|\tilde{\mathbf{W}}^{(\ell)}\|_2 \\
&\leq \sqrt{d}\Delta h_{\max}^{(\ell-1)}\|\tilde{\mathbf{W}}^{(\ell)}\|_2 + \sqrt{d}h_{\max}^{(\ell-1)}\|\Delta\mathbf{W}^{(\ell)}\|_2 \\
&\leq \sqrt{d}B_w \Delta h_{\max}^{(\ell-1)} + \sqrt{d}h_{\max}^{(\ell-1)}\|\Delta\mathbf{W}^{(\ell)}\|_2.
\end{aligned}
\tag{58}
$$

By induction, we have

$$
\begin{aligned}
\Delta h_{\max}^{(\ell)} &\le \sqrt{d}B_w \Delta h_{\max}^{(\ell-1)} + \sqrt{d}h_{\max}^{(\ell-1)}\|\Delta \mathbf{W}^{(\ell)}\|_2 \\
&\le (\sqrt{d}B_w)^2 \Delta h_{\max}^{(\ell-2)} + \sqrt{d}\Big(h_{\max}^{(\ell-1)}\|\Delta \mathbf{W}^{(\ell)}\|_2 + (\sqrt{d}B_w)h_{\max}^{(\ell-2)}\|\Delta \mathbf{W}^{(\ell-1)}\|_2\Big) \\
&\cdots \\
&\le (\sqrt{d}B_w)^{\ell}\Delta h_{\max}^{(0)} + \sqrt{d}\Big(h_{\max}^{(\ell-1)}\|\Delta \mathbf{W}^{(\ell)}\|_2 + (\sqrt{d}B_w)h_{\max}^{(\ell-2)}\|\Delta \mathbf{W}^{(\ell-1)}\|_2 + \ldots + (\sqrt{d}B_w)^{\ell-1}h_{\max}^{(0)}\|\Delta \mathbf{W}^{(1)}\|_2\Big) \\
&\underset{(a)}{=} \sqrt{d}\Big(h_{\max}^{(\ell-1)}\|\Delta \mathbf{W}^{(\ell)}\|_2 + (\sqrt{d}B_w)h_{\max}^{(\ell-2)}\|\Delta \mathbf{W}^{(\ell-1)}\|_2 + \ldots + (\sqrt{d}B_w)^{\ell-1}h_{\max}^{(0)}\|\Delta \mathbf{W}^{(1)}\|_2\Big),
\end{aligned}
\tag{59}
$$

where $(a)$ is due to $\Delta h_{\max}^{(0)} = 0$. Plugging in the upper bound of $h_{\max}^{(\ell)}$ in Lemma 6, yields

$$
\begin{aligned}
\Delta h_{\max}^{(\ell)} &\le \sqrt{d}B_x(\sqrt{d}B_w)^{\ell-1}\Big(\|\Delta \mathbf{W}^{(\ell)}\|_2 + \ldots + \|\Delta \mathbf{W}^{(1)}\|_2\Big) \\
&\le \sqrt{d}B_x(\max\{1, \sqrt{d}B_w\})^{\ell-1}\Big(\|\Delta \mathbf{W}^{(\ell)}\|_2 + \ldots + \|\Delta \mathbf{W}^{(1)}\|_2\Big).
\end{aligned}
\tag{60}
$$

### F.4 Proof of Lemma 8

By the definition of $\mathbf{Z}^{(\ell)}$, we have

$$
\begin{aligned}
\mathbf{Z}^{(\ell)} - \tilde{\mathbf{Z}}^{(\ell)} &= \mathbf{L}\mathbf{H}^{(\ell-1)}\mathbf{W}^{(\ell)} - \mathbf{L}\tilde{\mathbf{H}}^{(\ell-1)}\tilde{\mathbf{W}}^{(\ell)} \\
&= \mathbf{L}\Big(\mathbf{H}^{(\ell-1)}\mathbf{W}^{(\ell)} - \tilde{\mathbf{H}}^{(\ell-1)}\mathbf{W}^{(\ell)} + \tilde{\mathbf{H}}^{(\ell-1)}\mathbf{W}^{(\ell)} - \tilde{\mathbf{H}}^{(\ell-1)}\tilde{\mathbf{W}}^{(\ell)}\Big) \\
&= \mathbf{L}\big(\mathbf{H}^{(\ell-1)} - \tilde{\mathbf{H}}^{(\ell-1)}\big)\mathbf{W}^{(\ell)} + \mathbf{L}\tilde{\mathbf{H}}^{(\ell-1)}\big(\mathbf{W}^{(\ell)} - \tilde{\mathbf{W}}^{(\ell)}\big).
\end{aligned}
\tag{61}
$$

By taking norm on the both side of the equation, we have

$$
\begin{aligned}
\Delta z_{\max}^{(\ell)} &= \max_i \|[\mathbf{Z}^{(\ell)} - \tilde{\mathbf{Z}}^{(\ell)}]_{i,:}\|_2 \\
&\le \sqrt{d}B_w \cdot \max_i \|[\mathbf{Z}^{(\ell-1)} - \tilde{\mathbf{Z}}^{(\ell-1)}]_{i,:}\|_2 + \sqrt{d}h_{\max}^{(\ell-1)}\|\Delta \mathbf{W}^{(\ell)}\|_2 \\
&= \sqrt{d}B_w \cdot z_{\max}^{(\ell-1)} + \sqrt{d}h_{\max}^{(\ell-1)}\|\Delta \mathbf{W}^{(\ell)}\|_2.
\end{aligned}
\tag{62}
$$

By induction, we have

$$
\begin{aligned}
\Delta z_{\max}^{(\ell)} &\le \sqrt{d}B_w \cdot \Delta z_{\max}^{(\ell-1)} + \sqrt{d}h_{\max}^{(\ell-1)}\|\Delta \mathbf{W}^{(\ell)}\|_2 \\
&\underset{(a)}{\le} \sqrt{d}B_w \cdot \Delta z_{\max}^{(\ell-1)} + \sqrt{d}B_x(\sqrt{d}B_w)^{\ell-1}\|\Delta \mathbf{W}^{(\ell)}\|_2 \\
&\le \sqrt{d}B_w\Big(\sqrt{d}B_w \cdot \Delta z_{\max}^{(\ell-2)} + \sqrt{d}B_x(\sqrt{d}B_w)^{\ell-2}\|\Delta \mathbf{W}^{(\ell-1)}\|_2\Big) + \sqrt{d}B_x(\sqrt{d}B_w)^{\ell-1}\|\Delta \mathbf{W}^{(\ell)}\|_2 \\
&= (\sqrt{d}B_w)^2 \cdot \Delta z_{\max}^{(\ell-2)} + \sqrt{d}B_x(\sqrt{d}B_w)^{\ell-1}\Big(\|\Delta \mathbf{W}^{(\ell-1)}\|_2 + \|\Delta \mathbf{W}^{(\ell)}\|_2\Big) \\
&\le \ldots \\
&\le \sqrt{d}B_x(\sqrt{d}B_w)^{\ell-1}\Big(\|\Delta \mathbf{W}^{(1)}\|_2 + \ldots + \|\Delta \mathbf{W}^{(\ell)}\|_2\Big),
\end{aligned}
\tag{63}
$$

where (a) is due to $h_{\max}^{(\ell-1)} \le (\sqrt{d}B_w)^{\ell-1}B_x$.

## F.5 Proof of Lemma 9

For notation simplicity, let $\mathbf{D}^{(\ell)}$ denote the gradient passing from $\ell$th to $(\ell-1)$th layer. By the definition of $d_{\max}^{(\ell)}$, we have

$$
\begin{aligned}
d_{\max}^{(\ell)} &= \max_i \|\mathbf{d}_i^{(\ell)}\|_2 \\
&= \max_i \|[\mathbf{L}^\top \sigma'(\mathbf{Z}^{(\ell)}) \odot \mathbf{D}^{(\ell+1)} \mathbf{W}^{(\ell)}]_{i,:}\|_2 \\
&\underset{(a)}{\leq} \max_i \|[\mathbf{L}^\top \mathbf{D}^{(\ell+1)}]_{i,:}\|_2 \|\mathbf{W}^{(\ell)}\|_2 \\
&\leq \max_i \left\| \sum_{j=1}^N L_{i,j}^\top \mathbf{d}_j^{(\ell+1)} \right\|_2 \|\mathbf{W}^{(\ell)}\|_2 \\
&\leq \sqrt{d} \|\mathbf{W}^{(\ell)}\|_2 \cdot d_{\max}^{(\ell+1)} \\
&\leq \sqrt{d} B_w \cdot d_{\max}^{(\ell+1)} \\
&\underset{(b)}{\leq} \frac{2}{\gamma} (\sqrt{d} B_w)^{L-\ell+1} \leq \frac{2}{\gamma} (\max\{1, \sqrt{d} B_w\})^{L-\ell+1}
\end{aligned}
\tag{64}
$$

where $(a)$ is due to each element in $\sigma'(\mathbf{Z}^{(\ell)})$ is either 0 or 1 depending on $\mathbf{Z}^{(\ell)}$, and $(b)$ is due to $\|\frac{\partial \mathrm{Loss}(f(\mathbf{h}_i^{(L)}), y_i)}{\partial \mathbf{h}_i^{(L)}}\|_2 \leq 2/\gamma$. By taking norm on the both sides, we have

$$
\begin{aligned}
\Delta d_{\max}^{(\ell)} &= \max_i \|[\mathbf{D}^{(\ell)} - \tilde{\mathbf{D}}^{(\ell)}]_{i,:}\|_2 \\
&= \max_i \|[\mathbf{L}^\top (\mathbf{D}^{(\ell+1)} \odot \sigma'(\mathbf{Z}^{(\ell)}))[\mathbf{W}^{(\ell)}]^\top - \mathbf{L}^\top (\tilde{\mathbf{D}}^{(\ell+1)} \odot \sigma'(\tilde{\mathbf{Z}}^{(\ell)}))[\tilde{\mathbf{W}}^{(\ell)}]^\top]_{i,:}\|_2 \\
&\leq \max_i \|[\mathbf{L}^\top (\mathbf{D}^{(\ell+1)} \odot \sigma'(\mathbf{Z}^{(\ell)}))[\mathbf{W}^{(\ell)}]^\top - \mathbf{L}^\top (\tilde{\mathbf{D}}^{(\ell+1)} \odot \sigma'(\mathbf{Z}^{(\ell)}))[\mathbf{W}^{(\ell)}]^\top]_{i,:}\|_2 \\
&\quad + \max_i \|[\mathbf{L}^\top (\tilde{\mathbf{D}}^{(\ell+1)} \odot \sigma'(\mathbf{Z}^{(\ell)}))[\mathbf{W}^{(\ell)}]^\top - \mathbf{L}^\top (\tilde{\mathbf{D}}^{(\ell+1)} \odot \sigma'(\tilde{\mathbf{Z}}^{(\ell)}))[\mathbf{W}^{(\ell)}]^\top]_{i,:}\|_2 \\
&\quad + \max_i \|[\mathbf{L}^\top (\tilde{\mathbf{D}}^{(\ell+1)} \odot \sigma'(\tilde{\mathbf{Z}}^{(\ell)}))[\mathbf{W}^{(\ell)}]^\top - \mathbf{L}^\top (\tilde{\mathbf{D}}^{(\ell+1)} \odot \sigma'(\tilde{\mathbf{Z}}^{(\ell)}))[\tilde{\mathbf{W}}^{(\ell)}]^\top]_{i,:} \\
&\underset{(a)}{\leq} \max_i \|[\mathbf{L}^\top ((\mathbf{D}^{(\ell+1)} - \tilde{\mathbf{D}}^{(\ell+1)}))[\mathbf{W}^{(\ell)}]^\top]_{i,:}\|_2 \\
&\quad + \max_i \|[\mathbf{L}^\top (\tilde{\mathbf{D}}^{(\ell+1)} [\mathbf{W}^{(\ell)} - \tilde{\mathbf{W}}^{(\ell)}]^\top]_{i,:}\|_2 \\
&\quad + \max_i \|[\mathbf{L} \tilde{\mathbf{D}}^{(\ell+1)} (\mathbf{Z}^{(\ell)} - \tilde{\mathbf{Z}}^{(\ell)}) \mathbf{W}^{(\ell)}]_{i,:}\|_2 \\
&\leq \sqrt{d} B_w \Delta d_{\max}^{(\ell+1)} + \sqrt{d} d_{\max}^{(\ell+1)} \|\Delta \mathbf{W}^{(\ell)}\|_2 + \sqrt{d} B_w d_{\max}^{(\ell+1)} \Delta z_{\max}^{(\ell)} \\
&\leq \sqrt{d} B_w \Delta d_{\max}^{(\ell+1)} + \underbrace{d_{\max}^{(\ell+1)} \left( \sqrt{d} \|\Delta \mathbf{W}^{(\ell)}\|_2 + \sqrt{d} B_w \Delta z_{\max}^{(\ell)} \right)}_{(A)},
\end{aligned}
\tag{65}
$$

where inequality $(a)$ is due to the gradient of ReLU activation function is either 1 or 0.

Knowing that $d_{\max}^{(\ell+1)} \leq \frac{2}{\gamma} (\sqrt{d} B_w)^{L-\ell}$ and using the upper bound of $z_{\max}^{(\ell)}$ as derived in Lemma 8, we can upper bound $(A)$ as

$$
\begin{aligned}
d_{\max}^{(\ell+1)} &\left( \sqrt{d} \|\Delta \mathbf{W}^{(\ell)}\|_2 + \sqrt{d} B_w \Delta z_{\max}^{(\ell)} \right) \\
&\leq \frac{2}{\gamma} (\sqrt{d} B_w)^{L-\ell} \left( \sqrt{d} \|\Delta \mathbf{W}^{(\ell)}\|_2 + \sqrt{d} B_x (\sqrt{d} B_w)^\ell (\|\Delta \mathbf{W}^{(1)}\|_2 + \ldots + \|\Delta \mathbf{W}^{(\ell)}\|_2) \right) \\
&\leq \frac{2}{\gamma} (\sqrt{d} B_w)^{L-\ell} \left( \sqrt{d} + \sqrt{d} B_x (\sqrt{d} B_w)^\ell \right) \left( \|\Delta \mathbf{W}^{(1)}\|_2 + \ldots + \|\Delta \mathbf{W}^{(\ell)}\|_2 \right) \\
&\leq \frac{2}{\gamma} (\sqrt{d} B_w)^{L-\ell} \sqrt{d} (1 + B_x) (\max\{1, \sqrt{d} B_w\})^\ell \|\Delta \boldsymbol{\theta}\|_2 \\
&\leq \frac{2}{\gamma} \sqrt{d} (1 + B_x) (\max\{1, \sqrt{d} B_w\})^L \|\Delta \boldsymbol{\theta}\|_2.
\end{aligned}
\tag{66}
$$

Plugging it back, we have

$$
\begin{aligned}
\Delta d_{\max}^{(\ell)} &\leq \sqrt{d}B_w \Delta d_{\max}^{(\ell+1)} + \frac{2}{\gamma}\sqrt{d}(B_x+1)(\max\{1,\sqrt{d}B_w\})^L \|\Delta\boldsymbol{\theta}\|_2 \\
&\leq \sqrt{d}B_w\Big(\sqrt{d}B_w\Delta d_{\max}^{(\ell+2)} + \frac{2}{\gamma}\sqrt{d}(B_x+1)(\max\{1,\sqrt{d}B_w\})^L\|\Delta\boldsymbol{\theta}\|_2\Big) \\
&\quad + \frac{2}{\gamma}\sqrt{d}(B_x+1)(\max\{1,\sqrt{d}B_w\})^L\|\Delta\boldsymbol{\theta}\|_2 \\
&\leq \Big(1+\sqrt{d}B_w+\ldots+(\max\{1,\sqrt{d}B_w\})^{L-\ell-1}\Big)\cdot\frac{2}{\gamma}\sqrt{d}(B_x+1)(\max\{1,\sqrt{d}B_w\})^L\|\Delta\boldsymbol{\theta}\|_2 \\
&\quad + (\max\{1,\sqrt{d}B_w\})^{L-\ell+1}\Delta d_{\max}^{L+1} \\
&\leq (\max\{1,\sqrt{d}B_w\})^{L-\ell}\cdot\frac{2}{\gamma}L\sqrt{d}(B_x+1)(\max\{1,\sqrt{d}B_w\})^L\|\Delta\boldsymbol{\theta}\|_2 + (\max\{1,\sqrt{d}B_w\})^{L-\ell}\Delta d_{\max}^{L+1} \\
&= (\max\{1,\sqrt{d}B_w\})^{L-\ell}\Big(\frac{2}{\gamma}L\sqrt{d}(B_x+1)(\max\{1,\sqrt{d}B_w\})^L\|\Delta\boldsymbol{\theta}\|_2 + \Delta d_{\max}^{L+1}\Big).
\end{aligned}
$$
$$(67)$$

Then, our next step is to explicit upper bound $\Delta d_{\max}^{(L+1)}$. By definition, we can write $\Delta d_{\max}^{(L+1)}$ as

$$
\begin{aligned}
\Delta d_{\max}^{(L+1)} &= \max_i \|\mathbf{d}_i^{(L+1)} - \tilde{\mathbf{d}}_i^{(L+1)}\|_2 \\
&= \max_i \left\|\frac{\partial \mathrm{Loss}(f(\mathbf{h}_i^{(L)}),y_i)}{\partial \mathbf{h}_i^{(L)}} - \frac{\partial \mathrm{Loss}(\tilde{f}(\tilde{\mathbf{h}}_i^{(L)}),y_i)}{\partial \tilde{\mathbf{h}}_i^{(L)}}\right\|_2 \\
&\underset{(a)}{\leq} \frac{2}{\gamma}\max_i \left\|\frac{\partial f(\mathbf{h}_i^{(L)})}{\partial \mathbf{h}_i^{(L)}} - \frac{\partial \tilde{f}(\tilde{\mathbf{h}}_i^{(L)})}{\partial \tilde{\mathbf{h}}_i^{(L)}}\right\|_2 \\
&\underset{(b)}{\leq} \frac{2}{\gamma}\Big(\sqrt{d}B_x(\max\{1,\sqrt{d}B_w\})^{L-1}(\|\Delta\mathbf{W}^{(1)}\|_2+\ldots+\|\Delta\mathbf{W}^{(L)}\|_2) + \big(B_x(\max\{1,\sqrt{d}B_w\})^L+1\big)\Delta\mathbf{v}\Big) \\
&\leq \frac{2}{\gamma}\Big(\sqrt{d}B_x(\max\{1,\sqrt{d}B_w\})^L+1\Big)\Big(\|\Delta\mathbf{W}^{(1)}\|_2+\ldots+\|\Delta\mathbf{W}^{(L)}\|_2+\|\Delta\mathbf{v}\|_2\Big) \\
&= \frac{2}{\gamma}\Big(\sqrt{d}B_x(\max\{1,\sqrt{d}B_w\})^L+1\Big)\|\Delta\boldsymbol{\theta}\|_2,
\end{aligned}
$$
$$(68)$$

where $(a)$ is due to fact that $\nabla\mathrm{Loss}(z,y)$ is $2/\gamma$-Lipschitz continuous with respect to $z$, $(b)$ follows from Lemma 6 and Lemma 7.

Therefore, we have

$$
\begin{aligned}
\Delta d_{\max}^{(\ell)} &\leq (\max\{1,\sqrt{d}B_w\})^{L-\ell}\Big(\frac{2}{\gamma}L\sqrt{d}(B_x+1)(\max\{1,\sqrt{d}B_w\})^L\|\Delta\boldsymbol{\theta}\|_2 + \Delta d_{\max}^{L+1}\Big) \\
&\leq (\max\{1,\sqrt{d}B_w\})^{L-\ell}\frac{2}{\gamma}\Big(L\sqrt{d}(B_x+1)(\max\{1,\sqrt{d}B_w\})^L + \sqrt{d}B_x(\max\{1,\sqrt{d}B_w\})^L+1\Big)\|\Delta\boldsymbol{\theta}\|_2 \\
&\leq (\max\{1,\sqrt{d}B_w\})^{L-\ell}\frac{2}{\gamma}\Big((L+1)\sqrt{d}(B_x+1)(\max\{1,\sqrt{d}B_w\})^L+1\Big)\|\Delta\boldsymbol{\theta}\|_2.
\end{aligned}
$$
$$(69)$$

### F.6 Proof of Lemma 10

By the definition of $\mathbf{G}^{(\ell)}$, $\ell\in[L]$, we have

$$
\begin{aligned}
\|\mathbf{G}^{(\ell)}\|_2 &= \frac{1}{m}\|[\mathbf{L}\mathbf{H}^{(\ell-1)}]^\top\mathbf{D}^{(\ell)}\odot\sigma'(\mathbf{Z}^{(\ell)})\|_2 \\
&= \frac{1}{m}\|[\mathbf{L}\mathbf{H}^{(\ell-1)}]^\top\mathbf{D}^{(\ell)}\|_2 \\
&\leq \sqrt{d}h_{\max}^{(\ell-1)}d_{\max}^{(\ell)}.
\end{aligned}
$$
$$(70)$$

By plugging in the result from Lemma 6 and Lemma 9, we have

$$\|\mathbf{G}^{(\ell)}\|_2 \leq \frac{2}{\gamma}\sqrt{d}(B_x + 1)(\max\{1, \sqrt{d}B_w\})^L. \tag{71}$$

Besides, recall that $\mathbf{G}^{(L+1)}$ denotes the gradient with respect to the weight of binary classifier. Therefore, we have

$$\|\mathbf{G}^{(L+1)}\|_2 \leq \frac{2}{\gamma}h_{\max}^{(L)} \leq \frac{2}{\gamma}(\max\{1, \sqrt{d}B_w\})^L B_x, \tag{72}$$

which is smaller than the right hand side of the Eq. 71. Therefore, we have

$$\sum_{\ell=1}^{L+1}\|\mathbf{G}^{(\ell)}\|_2 \leq (L+1)\frac{2}{\gamma}\sqrt{d}(B_x + 1)(\max\{1, \sqrt{d}B_w\})^L. \tag{73}$$

Similarly, we can upper bound the difference between gradients computed on two different set of weight parameters for the $\ell$th layer as

$$\begin{aligned}
\|\mathbf{G}^{(\ell)} - \tilde{\mathbf{G}}^{(\ell)}\|_2 &= \frac{1}{m}\|[\mathbf{L}\mathbf{H}^{(\ell-1)}]^\top\mathbf{D}^{(\ell+1)}\odot\sigma'(\mathbf{Z}^{(\ell)}) - [\mathbf{L}\tilde{\mathbf{H}}^{(\ell-1)}]^\top\tilde{\mathbf{D}}^{(\ell+1)}\odot\sigma'(\tilde{\mathbf{Z}}^{(\ell)})\|_2 \\
&\leq \frac{1}{m}\|[\mathbf{L}\mathbf{H}^{(\ell-1)}]^\top\mathbf{D}^{(\ell+1)}\odot\sigma'(\mathbf{Z}^{(\ell)}) - [\mathbf{L}\tilde{\mathbf{H}}^{(\ell-1)}]^\top\mathbf{D}^{(\ell+1)}\odot\sigma'(\mathbf{Z}^{(\ell)})\|_2 \\
&\quad + \frac{1}{m}\|[\mathbf{L}\tilde{\mathbf{H}}^{(\ell-1)}]^\top\mathbf{D}^{(\ell+1)}\odot\sigma'(\mathbf{Z}^{(\ell)}) - [\mathbf{L}\tilde{\mathbf{H}}^{(\ell-1)}]^\top\tilde{\mathbf{D}}^{(\ell+1)}\odot\sigma'(\mathbf{Z}^{(\ell)})\|_2 \\
&\quad + \frac{1}{m}\|[\mathbf{L}\tilde{\mathbf{H}}^{(\ell-1)}]^\top\tilde{\mathbf{D}}^{(\ell+1)}\odot\sigma'(\mathbf{Z}^{(\ell)}) - [\mathbf{L}\tilde{\mathbf{H}}^{(\ell-1)}]^\top\tilde{\mathbf{D}}^{(\ell+1)}\odot\sigma'(\tilde{\mathbf{Z}}^{(\ell)})\|_2 \\
&\overset{\leq}{\scriptstyle(a)} \frac{1}{m}\|[\mathbf{L}(\mathbf{H}^{(\ell-1)} - \tilde{\mathbf{H}}^{(\ell-1)})]^\top\mathbf{D}^{(\ell+1)}\|_2 + \frac{1}{m}\|[\mathbf{L}\tilde{\mathbf{H}}^{(\ell-1)}]^\top(\mathbf{D}^{(\ell+1)} - \tilde{\mathbf{D}}^{(\ell+1)})\|_2 \\
&\quad + \frac{1}{m}\|[\mathbf{L}\tilde{\mathbf{H}}^{(\ell-1)}]^\top\tilde{\mathbf{D}}^{(\ell+1)}\odot\left(\sigma'(\mathbf{Z}^{(\ell)}) - \sigma'(\tilde{\mathbf{Z}}^{(\ell)})\right)\|_2 \\
&\overset{\leq}{\scriptstyle(b)} \max_i\|[[\mathbf{L}(\mathbf{H}^{(\ell-1)} - \tilde{\mathbf{H}}^{(\ell-1)})]^\top\mathbf{D}^{(\ell+1)}]_{i,:}\|_2 \\
&\quad + \max_i\|[[\mathbf{L}\tilde{\mathbf{H}}^{(\ell-1)}]^\top(\mathbf{D}^{(\ell+1)} - \tilde{\mathbf{D}}^{(\ell+1)})]_{i,:}\|_2 \\
&\quad + \max_i\|[\mathbf{L}\tilde{\mathbf{H}}^{(\ell-1)}]^\top\tilde{\mathbf{D}}^{(\ell+1)}\|_2\|\mathbf{Z}^{(\ell)} - \tilde{\mathbf{Z}}^{(\ell)}\|_2 \\
&\leq \sqrt{d}\Big(\underbrace{\Delta h_{\max}^{(\ell-1)}d_{\max}^{(\ell+1)}}_{(A)} + \underbrace{h_{\max}^{(\ell-1)}\Delta d_{\max}^{(\ell+1)}}_{(B)} + \underbrace{h_{\max}^{(\ell-1)}d_{\max}^{(\ell+1)}\Delta z_{\max}^{(\ell)}}_{(C)}\Big),
\end{aligned} \tag{74}$$

where inequalities $(a)$ and $(b)$ are due to the fact that the gradient of ReLU is either $0$ or $1$.

We now proceed to upper bound the three terms on the right hand side. By plugging the result from Lemma 6, Lemma 7, Lemma 9, and letting $C_1 = \max\{1, \sqrt{d}B_w\}$ and $C_2 = \sqrt{d}(B_x + 1)$, we can upper bound the term $(A)$ as

$$\begin{aligned}
\Delta h_{\max}^{(\ell-1)}d_{\max}^{(\ell+1)} &\leq \sqrt{d}B_x(\max\{1, \sqrt{d}B_w\})^{\ell-2}\cdot\frac{2}{\gamma}(\max\{1, \sqrt{d}B_w\})^{L-\ell} \\
&\leq \frac{2}{\gamma}\sqrt{d}B_x(\max\{1, \sqrt{d}B_w\})^L\|\Delta\boldsymbol{\theta}\|_2 \\
&\leq \frac{2}{\gamma}C_1^L C_2\|\Delta\boldsymbol{\theta}\|_2,
\end{aligned} \tag{75}$$

upper bound $(B)$ as

$$\begin{aligned}
h_{\max}^{(\ell-1)}\Delta d_{\max}^{(\ell+1)} &\leq B_x(\max\{1, \sqrt{d}B_w\})^L\frac{2}{\gamma}\Big((L+1)\sqrt{d}(B_x + 1)(\max\{1, \sqrt{d}B_w\})^L + 1\Big)\|\Delta\boldsymbol{\theta}\|_2 \\
&\leq \frac{2}{\gamma}C_1^L C_2\Big((L+1)C_1^L C_2 + 1\Big)\|\Delta\boldsymbol{\theta}\|_2,
\end{aligned} \tag{76}$$

and upper bound $(C)$ as

$$
\begin{aligned}
h_{\max}^{(\ell-1)} d_{\max}^{(\ell+1)} \Delta z_{\max}^{(\ell)} &\le B_x(\max\{1,\sqrt{d}B_w\})^{\ell-1} \cdot \frac{2}{\gamma}(\max\{1,\sqrt{d}B_w\})^{L-\ell} \cdot \sqrt{d}B_x(\max\{1,\sqrt{d}B_w\})^{\ell-1}\|\Delta\boldsymbol{\theta}\|_2 \\
&\le \frac{2}{\gamma}\sqrt{d}B_x^2(\max\{1,\sqrt{d}B_w\})^{2L}\|\Delta\boldsymbol{\theta}\|_2 \\
&\le \frac{2}{\gamma}(C_1^L C_2)^2\|\Delta\boldsymbol{\theta}\|_2.
\end{aligned}
\tag{77}
$$

By combining the results above, we can upper bound $\Delta\mathbf{G}^{(\ell)}$ as

$$
\begin{aligned}
\|\mathbf{G}^{(\ell)} - \tilde{\mathbf{G}}^{(\ell)}\|_2 &\le \sqrt{d}\Big(\Delta h_{\max}^{(\ell-1)} d_{\max}^{(\ell+1)} + h_{\max}^{(\ell-1)} \Delta d_{\max}^{(\ell+1)} + h_{\max}^{(\ell-1)} d_{\max}^{(\ell+1)} \cdot \Delta z_{\max}^{(\ell)}\Big) \\
&\le \frac{2}{\gamma}C_1^L C_2\Big((L+2)C_1^L C_2 + 2\Big)\|\Delta\boldsymbol{\theta}\|_2.
\end{aligned}
\tag{78}
$$

Similarly, we can upper bound the difference between gradient for the weight of binary classifier as

$$
\begin{aligned}
\|\mathbf{G}^{(L+1)} - \tilde{\mathbf{G}}^{(L+1)}\|_2 &\le \frac{2}{\gamma}\Delta h_{\max}^{(L)} \\
&\le \frac{2}{\gamma}\sqrt{d}B_x(\{1,\sqrt{d}B_w\})^{L-1}\Big(\|\mathbf{W}^{(1)}\|_2 + \ldots + \|\mathbf{W}^{(L)}\|_2\Big),
\end{aligned}
\tag{79}
$$

which is smaller than the right hand side of the previous equation. Therefore, we have

$$
\sum_{\ell=1}^{L+1}\|\mathbf{G}^{(\ell)} - \tilde{\mathbf{G}}^{(\ell)}\|_2 \le (L+1)\frac{2}{\gamma}C_1^L C_2\Big((L+2)C_1^L C_2 + 2\Big)\|\Delta\boldsymbol{\theta}\|_2.
\tag{80}
$$

# G    Generalization bound for ResGCN

In the following, we provide detailed proof on the generalization bound of ResGCN. Recall that the update rule of ResGCN is defined as

$$
\mathbf{H}^{(\ell)} = \sigma(\mathbf{L}\mathbf{H}^{(\ell-1)}\mathbf{W}^{(\ell)}) + \mathbf{H}^{(\ell-1)},
\tag{81}
$$

where $\sigma(\cdot)$ is ReLU activation function. Please notice that although ReLU function $\sigma(x)$ is not differentiable when $x = 0$, for analysis purpose, we suppose the $\sigma'(0) = 0$.

The training of ResGCN is an empirical risk minimization with respect to a set of parameters $\boldsymbol{\theta} = \{\mathbf{W}^{(1)}, \ldots, \mathbf{W}^{(L)}, \mathbf{v}\}$, i.e.,

$$
\mathcal{L}(\boldsymbol{\theta}) = \frac{1}{m}\sum_{i=1}^{m}\Phi_\gamma(-p(f(\mathbf{h}_i^{(L)}), y_i)), \; f(\mathbf{h}_i^{(L)}) = \tilde{\sigma}(\mathbf{v}^\top \mathbf{h}_i^{(L)}),
\tag{82}
$$

where $\mathbf{h}_i^{(L)}$ is the node representation of the $i$th node at the final layer, $f(\mathbf{h}_i^{(L)})$ is the prediction of the $i$th node, $\tilde{\sigma}(x) = \frac{1}{\exp(-x)+1}$ is the sigmoid function, and loss function $\Phi_\gamma(-p(z,y))$ is $\frac{2}{\gamma}$-Lipschitz continuous with respect to its first input $z$. For simplification, we will use $\mathrm{Loss}(z,y)$ denote $\Phi_\gamma(-p(z,y))$ in the proof.

To establish the generalization of ResGCN as stated in Theorem 3, we utilize the result on transductive uniform stability from [17] (Theorem 8 in Appendix F). Then, in Lemma 11, we derive the uniform stability constant for ResGCN, i.e., $\epsilon_{\text{ResGCN}}$.

**Lemma 11.** *The uniform stability constant for ResGCN is computed as* $\epsilon_{ResGCN} = \frac{2\eta\rho_f G_f}{m}\sum_{t=1}^{T}(1 + \eta L_F)^{t-1}$ *where*

$$
\begin{aligned}
\rho_f &= C_1^L C_2, \; G_f = \frac{2}{\gamma}(L+1)C_1^L C_2, \; L_f = \frac{2}{\gamma}(L+1)C_1^L C_2\Big((L+2)C_1^L C_2 + 2\Big), \\
C_1 &= 1 + \sqrt{d}B_w, \; C_2 = \sqrt{d}(B_x + 1).
\end{aligned}
\tag{83}
$$

By plugging the result in Lemma 11 back to Theorem 8, we establish the generalization bound for ResGCN.

Similar to the proof on the generalization bound of GCN in Section F, the key idea of the proof is to decompose the change if the ResGCN output into two terms (in Lemma 5) which depend on

- (Lemma 12) The maximum change of node representation, i.e., $\Delta h_{\max}^{(\ell)} = \max_i \|[\mathbf{H}^{(\ell)} - \tilde{\mathbf{H}}^{(\ell)}]_{i,:}\|_2$,

- (Lemma 13) The maximum node representation, i.e., $h_{\max}^{(\ell)} = \max_i \|[\mathbf{H}^{(\ell)}]_{i,:}\|_2$.

**Lemma 12** (Upper bound of $h_{\max}^{(\ell)}$ for **ResGCN**). *Let suppose Assumptoon 1 hold. Then, the maximum node embeddings for any node at the $\ell$th layer is bounded by*

$$h_{\max}^{(\ell)} \leq B_x(1 + \sqrt{d}B_w)^{\ell}. \tag{84}$$

**Lemma 13** (Upper bound of $\Delta h_{\max}^{(\ell)}$ for **ResGCN**). *Let suppose Assumption 1 hold. Then, the maximum change between the node embeddings on two different set of weight parameters for any node at the $\ell$th layer is bounded by*

$$\Delta h_{\max}^{(\ell)} \leq \sqrt{d}B_x(1 + \sqrt{d}B_w)^{\ell-1}(\|\Delta\mathbf{W}^{(1)}\|_2 + \ldots + \|\Delta\mathbf{W}^{(\ell)}\|_2), \tag{85}$$

*where $\Delta\mathbf{W}^{(\ell)} = \mathbf{W}^{(\ell)} - \tilde{\mathbf{W}}^{(\ell)}$.*

Besides, in Lemma 14, we derive the upper bound on the maximum change of node embeddings before the activation function, i.e., $\Delta z_{\max}^{(\ell)} = \max_i \|[\mathbf{Z}^{(\ell)} - \tilde{\mathbf{Z}}^{(\ell)}]_{i,:}\|_2$, which will be used for the proof of gradient related upper bounds.

**Lemma 14** (Upper bound of $\Delta z_{\max}^{(\ell)}$ for **ResGCN**). *Let suppose Assumption 1 hold. Then, the maximum change between the node embeddings before the activation function on two different set of weight parameters for any node at the $\ell$th layer is bounded by*

$$\Delta z_{\max}^{(\ell)} \leq \sqrt{d}B_x(1 + \sqrt{d}B_w)^{\ell-1}(\|\Delta\mathbf{W}^{(1)}\|_2 + \ldots + \|\Delta\mathbf{W}^{(\ell)}\|_2), \tag{86}$$

*where $\Delta\mathbf{W}^{(\ell)} = \mathbf{W}^{(\ell)} - \tilde{\mathbf{W}}^{(\ell)}$.*

Then, in Lemma 15, we decompose the change of the model parameters into two terms which depend on

- The maximum change of gradient passing from the $(\ell+1)$th layer to the $\ell$th layer

$$\Delta d_{\max}^{(\ell)} = \max_i \left\| \left[ \frac{\partial(\sigma(\mathbf{L}\mathbf{H}^{(\ell-1)}\mathbf{W}^{(\ell)}) + \mathbf{H}^{(\ell-1)})}{\partial\mathbf{H}^{(\ell-1)}} - \frac{\partial(\sigma(\mathbf{L}\tilde{\mathbf{H}}^{(\ell-1)}\tilde{\mathbf{W}}^{(\ell)}) + \tilde{\mathbf{H}}^{(\ell-1)})}{\partial\tilde{\mathbf{H}}^{(\ell-1)}} \right]_{i,:} \right\|_2.$$

- The maximum gradient passing from the $(\ell+1)$th layer to the $\ell$th layer

$$d_{\max}^{(\ell)} = \max_i \left\| \left[ \frac{\partial(\sigma(\mathbf{L}\mathbf{H}^{(\ell-1)}\mathbf{W}^{(\ell)}) + \mathbf{H}^{(\ell-1)})}{\partial\mathbf{H}^{(\ell-1)}} \right]_{i,:} \right\|_2.$$

**Lemma 15** (Upper bound of $d_{\max}^{(\ell)}, \Delta d_{\max}^{(\ell)}$ for **ResGCN**). *Let suppose Assumption 1 hold. Then, the maximum gradient passing from layer $\ell+1$ to layer $\ell$ for any node is bounded by*

$$d_{\max}^{(\ell)} \leq \frac{2}{\gamma}(1 + \sqrt{d}B_w)^{L-\ell+1}, \tag{87}$$

*and the maximum change between the gradient passing from layer $\ell+1$ to layer $\ell$ on two different set of weight parameters for any node is bounded by*

$$\Delta d_{\max}^{(\ell)} \leq \frac{2}{\gamma}(1 + \sqrt{d}B_w)^{L-\ell}\Big((L+1)\sqrt{d}(B_x+1)(1 + \sqrt{d}B_w)^L + 1\Big)\|\Delta\boldsymbol{\theta}\|_2, \tag{88}$$

*where $\|\Delta\boldsymbol{\theta}\|_2 = \|\mathbf{v} - \tilde{\mathbf{v}}\|_2 + \sum_{\ell-1}^{L} \|\mathbf{W}^{(\ell)} - \tilde{\mathbf{W}}^{(\ell)}\|_2$ denotes the change of two set of parameters.*

Finally, based on the previous result, in Lemma 16, we decompose the change of the model parameters into two terms which depend on

- The change of gradient with respect to the $\ell$th layer weight matrix $\|\Delta \mathbf{G}^{(\ell)}\|_2$,

- The gradient with respect to the $\ell$th layer weight matrix $\|\mathbf{G}^{(\ell)}\|_2$,

where $\mathbf{G}^{(L+1)}$ denotes the gradient with respect to the weight $\mathbf{v}$ of the binary classifier and $\mathbf{G}^{(\ell)}$ denotes the gradient with respect to the weight $\mathbf{W}^{(\ell)}$ of the $\ell$th layer graph convolutional layer. Notice that $\|\Delta \mathbf{G}^{(\ell)}\|_2$ reflect the smoothness of ResGCN model and $\|\mathbf{G}^{(\ell)}\|_2$ correspond the upper bound of gradient.

**Lemma 16** (Upper bound of $\|\mathbf{G}^{(\ell)}\|_2, \|\Delta \mathbf{G}^{(\ell)}\|_2$ for **ResGCN**). *Let suppose Assumption 1 hold and let $C_1 = 1 + \sqrt{d}B_w$ and $C_2 = \sqrt{d}(B_x + 1)$. Then, the gradient and the maximum change between gradients on two different set of weight parameters are bounded by*

$$
\begin{aligned}
\sum_{\ell=1}^{L+1} \|\mathbf{G}^{(\ell)}\|_2 &\leq \frac{2}{\gamma}(L+1)C_1^L C_2 \|\Delta \boldsymbol{\theta}\|_2, \\
\sum_{\ell=1}^{L+1} \|\Delta \mathbf{G}^{(\ell)}\|_2, &\leq \frac{2}{\gamma}(L+1)C_1^L C_2 \Big((L+2)C_1^L C_2 + 2\Big) \|\Delta \boldsymbol{\theta}\|_2,
\end{aligned}
\tag{89}
$$

*where $\|\Delta \boldsymbol{\theta}\|_2 = \|\mathbf{v} - \tilde{\mathbf{v}}\|_2 + \sum_{\ell-1}^{L} \|\mathbf{W}^{(\ell)} - \tilde{\mathbf{W}}^{(\ell)}\|_2$ denotes the change of two set of parameters.*

Equipped with above intermediate results, we now proceed to prove Lemma 11.

*Proof.* Recall that our goal is to explore the impact of different GCN structures on the uniform stability constant $\epsilon_{\text{ResGCN}}$, which is a function of $\rho_f$, $G_f$, and $L_f$. Let $C_1 = 1 + \sqrt{d}B_w$, $C_2 = \sqrt{d}(B_x + 1)$. Firstly, by plugging Lemma 13 and Lemma 12 into Lemma 5, we have that

$$
\begin{aligned}
\max_i |f(\mathbf{h}_i^{(L)}) - \tilde{f}(\tilde{\mathbf{h}}_i^{(L)})| &\leq \Delta h_{\max}^{(L)} + h_{\max}^{(L)} \|\Delta \mathbf{v}\|_2 \\
&\leq \sqrt{d}B_x \cdot (\max\{1, \sqrt{d}B_w\})^L \|\Delta \boldsymbol{\theta}\|_2 \\
&\leq C_1^L C_2 \|\Delta \boldsymbol{\theta}\|_2.
\end{aligned}
\tag{90}
$$

Therefore, we know that the function $f$ is $\rho_f$-Lipschitz continuous, with $\rho_f = C_1^L C_2$. Then, by Lemma 16, we know that the function $f$ is $L_f$-smoothness, and the gradient of each weight matrices is bounded by $G_f$, with

$$
G_f = \frac{2}{\gamma}(L+1)C_1^L C_2, \quad L_f = \frac{2}{\gamma}(L+1)C_1^L C_2 \Big((L+2)C_1^L C_2 + 2\Big).
\tag{91}
$$

By plugging $\epsilon_{\text{ResGCN}}$ into Theorem 3, we obtain the generalization bound of ResGCN.

$\square$

## G.1  Proof of Lemma 12

By the definition of $h_{\max}^{(\ell)}$, we have

$$
\begin{aligned}
h_{\max}^{(\ell)} &= \max_i \|[\sigma(\mathbf{L}\mathbf{H}^{(\ell-1)}\mathbf{W}^{(\ell)}) + \mathbf{H}^{(\ell-1)}]_{i,:}\|_2 \\
&\leq \max_i \|[\mathbf{L}\mathbf{H}^{(\ell-1)}\mathbf{W}^{(\ell)}]_{i,:}\|_2 + \max_i \|\mathbf{h}_i^{(\ell-1)}\|_2 \\
&\leq \max_i \|[\mathbf{L}\mathbf{H}^{(\ell-1)}]_{i,:}\|_2 \|\mathbf{W}^{(\ell)}\|_2 + h_{\max}^{(\ell-1)} \\
&= \max_i \left\| \sum_{j=1}^N L_{i,j} \mathbf{h}_j^{(\ell-1)} \right\|_2 \|\mathbf{W}^{(\ell)}\|_2 + h_{\max}^{(\ell-1)} \\
&\leq \max_i \left\| \sum_{j=1}^N L_{i,j} \right\|_2 \cdot \max_j \|\mathbf{h}_j^{(\ell-1)}\|_2 \cdot \|\mathbf{W}^{(\ell)}\|_2 + h_{\max}^{(\ell-1)} \\
&\underset{(a)}{\leq} (1 + \sqrt{d}\|\mathbf{W}^{(\ell)}\|_2) \cdot h_{\max}^{(\ell-1)} \\
&\leq (1 + \sqrt{d}B_w) \cdot h_{\max}^{(\ell-1)} \leq B_x(1 + \sqrt{d}B_w)^\ell
\end{aligned}
\tag{92}
$$

where inequality $(a)$ follows from Lemma 31.

## G.2  Proof of Lemma 13

By the definition of $\Delta h_{\max}^{(\ell)}$, we have

$$
\begin{aligned}
\Delta h_{\max}^{(\ell)} &= \max_i \|\mathbf{h}_i^{(\ell)} - \tilde{\mathbf{h}}_i^{(\ell)}\|_2 \\
&= \max_i \|[\sigma(\mathbf{L}\mathbf{H}^{(\ell-1)}\mathbf{W}^{(\ell)}) - \sigma(\mathbf{L}\tilde{\mathbf{H}}^{(\ell-1)}\tilde{\mathbf{W}}^{(\ell)}) + \mathbf{H}^{(\ell)} - \tilde{\mathbf{H}}^{(\ell)}]_{i,:}\|_2 \\
&\leq \max_i \|[\mathbf{L}\mathbf{H}^{(\ell-1)}\mathbf{W}^{(\ell)} - \mathbf{L}\tilde{\mathbf{H}}^{(\ell-1)}\tilde{\mathbf{W}}^{(\ell)}]_{i,:}\|_2 + \max_i \|[\mathbf{H}^{(\ell)} - \tilde{\mathbf{H}}^{(\ell)}]_{i,:}\|_2 \\
&\leq \max_i \|[\mathbf{L}\mathbf{H}^{(\ell-1)}(\mathbf{W}^{(\ell)} - \tilde{\mathbf{W}}^{(\ell)}) - \mathbf{L}(\mathbf{H}^{(\ell-1)} - \tilde{\mathbf{H}}^{(\ell-1)})\tilde{\mathbf{W}}^{(\ell)}]_{i,:}\|_2 + \Delta h_{\max}^{(\ell-1)} \\
&\underset{(a)}{\leq} \max_i \left\| \sum_{j=1}^N L_{i,j} \mathbf{h}_j^{(\ell-1)} \right\|_2 \|\Delta\mathbf{W}^{(\ell)}\|_2 + \max_i \left\| \sum_{j=1}^N L_{i,j} \Delta\mathbf{h}_j^{(\ell-1)} \right\|_2 \|\tilde{\mathbf{W}}^{(\ell)}\|_2 + \Delta h_{\max}^{(\ell-1)} \\
&\leq (1 + \sqrt{d}B_w) \cdot \Delta h_{\max}^{(\ell-1)} + \sqrt{d}h_{\max}^{(\ell-1)}\|\Delta\mathbf{W}^{(\ell)}\|_2,
\end{aligned}
\tag{93}
$$

where inequality $(a)$ follows from Lemma 31.

By induction, we have

$$
\begin{aligned}
\Delta h_{\max}^{(\ell)} &\leq (1 + \sqrt{d}B_w) \cdot \Delta h_{\max}^{(\ell-1)} + \sqrt{d}h_{\max}^{(\ell-1)}\|\Delta\mathbf{W}^{(\ell)}\|_2 \\
&\leq (1 + \sqrt{d}B_w)^2 \cdot \Delta h_{\max}^{(\ell-2)} \\
&\quad + \sqrt{d}\Big(h_{\max}^{(\ell-1)}\|\Delta\mathbf{W}^{(\ell)}\|_2 + (1 + \sqrt{d}B_w)h_{\max}^{(\ell-2)}\|\Delta\mathbf{W}^{(\ell-1)}\|_2\Big) \\
&\cdots \\
&\leq (1 + \sqrt{d}B_w)^\ell \cdot \Delta h_{\max}^{(0)} \\
&\quad + \sqrt{d}\Big(h_{\max}^{(\ell-1)}\|\Delta\mathbf{W}^{(\ell)}\|_2 + (1 + \sqrt{d}B_w)h_{\max}^{(\ell-2)}\|\Delta\mathbf{W}^{(\ell-1)}\|_2 + \ldots + (1 + \sqrt{d}B_w)^{\ell-1}h_{\max}^{(0)}\|\Delta\mathbf{W}^{(1)}\|_2\Big) \\
&= \sqrt{d}\Big(h_{\max}^{(\ell-1)}\|\Delta\mathbf{W}^{(\ell)}\|_2 + (1 + \sqrt{d}B_w)h_{\max}^{(\ell-2)}\|\Delta\mathbf{W}^{(\ell-1)}\|_2 + \ldots + (1 + \sqrt{d}B_w)^{\ell-1}h_{\max}^{(0)}\|\Delta\mathbf{W}^{(1)}\|_2\Big),
\end{aligned}
\tag{94}
$$

where the last equality is due to $\Delta h_{\max}^{(0)} = 0$.

Plugging in the upper bound of $h_{\max}^{(\ell)}$ in Lemma 13, we have

$$
\Delta h_{\max}^{(\ell)} \leq \sqrt{d}B_x(1 + \sqrt{d}B_w)^{\ell-1}(\|\mathbf{W}^{(1)}\|_2 + \ldots + \|\mathbf{W}^{(\ell)}\|_2).
\tag{95}
$$

### G.3 Proof of Lemma 14

By the definition of $\mathbf{Z}^{(\ell)}$, we have

$$
\begin{aligned}
\mathbf{Z}^{(\ell)} - \tilde{\mathbf{Z}}^{(\ell)} &= \mathbf{L}\mathbf{H}^{(\ell-1)}\mathbf{W}^{(\ell)} - \mathbf{L}\tilde{\mathbf{H}}^{(\ell-1)}\tilde{\mathbf{W}}^{(\ell)} \\
&= \mathbf{L}\Big(\mathbf{H}^{(\ell-1)}\mathbf{W}^{(\ell)} - \tilde{\mathbf{H}}^{(\ell-1)}\mathbf{W}^{(\ell)} + \tilde{\mathbf{H}}^{(\ell-1)}\mathbf{W}^{(\ell)} - \tilde{\mathbf{H}}^{(\ell-1)}\tilde{\mathbf{W}}^{(\ell)}\Big) \quad (96) \\
&= \mathbf{L}\big(\mathbf{H}^{(\ell-1)} - \tilde{\mathbf{H}}^{(\ell-1)}\big)\mathbf{W}^{(\ell)} + \mathbf{L}\tilde{\mathbf{H}}^{(\ell-1)}\big(\mathbf{W}^{(\ell)} - \tilde{\mathbf{W}}^{(\ell)}\big).
\end{aligned}
$$

Then, by taking the norm of both sides, we have

$$
\begin{aligned}
\Delta z_{\max}^{(\ell)} &= \max_i \|[\mathbf{Z}^{(\ell)} - \tilde{\mathbf{Z}}^{(\ell)}]_{i,:}\|_2 \\
&\leq \sqrt{d}B_w \cdot \max_i \|[\mathbf{Z}^{(\ell-1)} - \tilde{\mathbf{Z}}^{(\ell-1)}]_{i,:}\|_2 + \sqrt{d}h_{\max}^{(\ell-1)}\|\Delta\mathbf{W}^{(\ell)}\|_2 \quad (97) \\
&= \sqrt{d}B_w \cdot z_{\max}^{(\ell-1)} + \sqrt{d}h_{\max}^{(\ell-1)}\|\Delta\mathbf{W}^{(\ell)}\|_2.
\end{aligned}
$$

By induction, we have

$$
\begin{aligned}
\Delta z_{\max}^{(\ell)} &\leq \sqrt{d}B_w \cdot \Delta z_{\max}^{(\ell-1)} + \sqrt{d}h_{\max}^{(\ell-1)}\|\Delta\mathbf{W}^{(\ell)}\|_2 \\
&\underset{(a)}{\leq} \sqrt{d}B_w \cdot \Delta z_{\max}^{(\ell-1)} + \sqrt{d}B_x(\sqrt{d}B_w)^{\ell-1}\|\Delta\mathbf{W}^{(\ell)}\|_2 \\
&\leq \sqrt{d}B_w\Big(\sqrt{d}B_w \cdot \Delta z_{\max}^{(\ell-2)} + \sqrt{d}B_x(\sqrt{d}B_w)^{\ell-2}\|\Delta\mathbf{W}^{(\ell-1)}\|_2\Big) + \sqrt{d}B_x(1 + \sqrt{d}B_w)^{\ell-1}\|\Delta\mathbf{W}^{(\ell)}\|_2 \\
&= (\sqrt{d}B_w)^2 \cdot \Delta z_{\max}^{(\ell-2)} + \sqrt{d}B_x(\sqrt{d}B_w)^{\ell-1}\Big(\|\Delta\mathbf{W}^{(\ell-1)}\|_2 + \|\Delta\mathbf{W}^{(\ell)}\|_2\Big) \\
&\leq \ldots \\
&\leq \sqrt{d}B_x(\sqrt{d}B_w)^{\ell-1}\Big(\|\Delta\mathbf{W}^{(1)}\|_2 + \ldots + \|\Delta\mathbf{W}^{(\ell)}\|_2\Big),
\end{aligned}
$$

$$(98)$$

where (a) is due to $h_{\max}^{(\ell-1)} \leq (\sqrt{d}B_w)^{\ell-1}B_x$.

### G.4 Proof of Lemma 15

For notation simplicity, let $\mathbf{D}^{(\ell)}$ denote the gradient passing from the $\ell$th to the $(\ell-1)$th layer. By the definition of $d_{\max}^{(\ell)}$, we have

$$
\begin{aligned}
d_{\max}^{(\ell)} &= \max_i \|\mathbf{d}_i^{(\ell)}\|_2 \\
&= \max_i [\|\mathbf{L}^\top\sigma'(\mathbf{Z}^{(\ell)}) \odot \mathbf{D}^{(\ell+1)}\mathbf{W}^{(\ell)} + \mathbf{D}^{(\ell+1)}\|]_{i,:} \\
&\leq \max_i [\|\mathbf{L}^\top\mathbf{D}^{(\ell+1)}\|_2]_{i,:}\|\mathbf{W}^{(\ell)}\|_2 + d_{\max}^{(\ell+1)} \\
&\leq \max_i \left\|\sum_{j=1}^N L_{i,j}^\top \mathbf{d}_j^{(\ell+1)}\right\|_2 \|\mathbf{W}^{(\ell)}\|_2 + d_{\max}^{(\ell+1)} \quad (99) \\
&\leq (1 + \sqrt{d}\|\mathbf{W}^{(\ell)}\|_2)d_{\max}^{(\ell+1)} \\
&\underset{(a)}{\leq} (1 + \sqrt{d}B_w) \cdot d_{\max}^{(\ell+1)} \leq \frac{2}{\gamma}(1 + \sqrt{d}B_w)^{L-\ell+1},
\end{aligned}
$$

where inequality $(a)$ follows from Lemma 31.

By the definition of $\Delta d_{\max}^{(\ell)}$, we have

$$
\begin{aligned}
\Delta d_{\max}^{(\ell)} &= \max_i \|[\mathbf{D}^{(\ell)} - \tilde{\mathbf{D}}^{(\ell)}]_{i,:}\|_2 \\
&= \max_i \|[\mathbf{L}^\top(\mathbf{D}^{(\ell+1)} \odot \sigma'(\mathbf{Z}^{(\ell)}))[\mathbf{W}^{(\ell)}]^\top + \mathbf{D}^{(\ell+1)} - \mathbf{L}^\top(\tilde{\mathbf{D}}^{(\ell+1)} \odot \sigma'(\tilde{\mathbf{Z}}^{(\ell)}))[\tilde{\mathbf{W}}^{(\ell)}]^\top - \tilde{\mathbf{D}}^{(\ell+1)}]_{i,:}\|_2 \\
&\leq \max_i \|[\mathbf{L}^\top(\mathbf{D}^{(\ell+1)} \odot \sigma'(\mathbf{Z}^{(\ell)}))[\mathbf{W}^{(\ell)}]^\top - \mathbf{L}^\top(\tilde{\mathbf{D}}^{(\ell+1)} \odot \sigma'(\mathbf{Z}^{(\ell)}))[\mathbf{W}^{(\ell)}]^\top]_{i,:}\|_2 \\
&\quad + \max_i \|[\mathbf{L}^\top(\tilde{\mathbf{D}}^{(\ell+1)} \odot \sigma'(\mathbf{Z}^{(\ell)}))[\mathbf{W}^{(\ell)}]^\top - \mathbf{L}^\top(\tilde{\mathbf{D}}^{(\ell+1)} \odot \sigma'(\mathbf{Z}^{(\ell)}))[\tilde{\mathbf{W}}^{(\ell)}]^\top]_{i,:}\|_2 \\
&\quad + \max_i \|[\mathbf{L}^\top(\tilde{\mathbf{D}}^{(\ell+1)} \odot \sigma'(\mathbf{Z}^{(\ell)}))[\tilde{\mathbf{W}}^{(\ell)}]^\top - \mathbf{L}^\top(\tilde{\mathbf{D}}^{(\ell+1)} \odot \sigma'(\tilde{\mathbf{Z}}^{(\ell)}))[\tilde{\mathbf{W}}^{(\ell)}]^\top]_{i,:}\|_2 + \max_i \|[\mathbf{D}^{(\ell+1)} - \tilde{\mathbf{D}}^{(\ell+1)}]_{i,:}\|_2 \\
&\underset{(a)}{\leq} \max_i \|[\mathbf{L}^\top((\mathbf{D}^{(\ell+1)} - \tilde{\mathbf{D}}^{(\ell+1)}))[\mathbf{W}^{(\ell)}]^\top]_{i,:}\|_2 + \max_i \|[\mathbf{L}^\top(\tilde{\mathbf{D}}^{(\ell+1)}[\mathbf{W}^{(\ell)} - \tilde{\mathbf{W}}^{(\ell)}]^\top]_{i,:}\|_2 \\
&\quad + \max_i \|[\mathbf{L}^\top\tilde{\mathbf{D}}^{(\ell+1)}[\tilde{\mathbf{W}}^{(\ell)}]^\top]_{i,:}\|_2 \max_i \|[\mathbf{Z}^{(\ell)} - \tilde{\mathbf{Z}}^{(\ell)}]_{i,:}\| + \max_i \|[\mathbf{D}^{(\ell+1)} - \tilde{\mathbf{D}}^{(\ell+1)}]_{i,:}\|_2 \\
&\leq (1 + \sqrt{d}B_w)\Delta d_{\max}^{(\ell+1)} + \sqrt{d}d_{\max}^{(\ell+1)}\|\Delta\mathbf{W}^{(\ell)}\|_2 + \sqrt{d}B_w d_{\max}^{(\ell+1)}\Delta z_{\max}^{(\ell)} \\
&= (1 + \sqrt{d}B_w)\Delta d_{\max}^{(\ell+1)} + \underbrace{d_{\max}^{(\ell+1)}\left(\sqrt{d}\|\Delta\mathbf{W}^{(\ell)}\|_2 + \sqrt{d}B_w\Delta z_{\max}^{(\ell)}\right)}_{(A)},
\end{aligned}
$$

(100)

where inequality $(a)$ is due to the fact that the gradient of ReLU is either $1$ or $0$.

Knowing that $d_{\max}^{(\ell+1)} \leq \frac{2}{\gamma}(1 + \sqrt{d}B_w)^{L-\ell}$ and $z_{\max}^{(\ell)} \leq \sqrt{d}B_x(\sqrt{d}B_w)^{\ell-1}\left(\|\mathbf{W}^{(1)}\|_2 + \ldots + \|\mathbf{W}^{(\ell)}\|_2\right)$, we can upper bound $(A)$ by

$$
\begin{aligned}
&d_{\max}^{(\ell+1)}\left(\sqrt{d}\|\Delta\mathbf{W}^{(\ell)}\|_2 + \sqrt{d}B_w\Delta z_{\max}^{(\ell)}\right) \\
&\leq \frac{2}{\gamma}(1 + \sqrt{d}B_w)^{L-\ell}\left(\sqrt{d}\|\Delta\mathbf{W}^{(\ell)}\|_2 + \sqrt{d}B_x(\sqrt{d}B_w)^\ell(\|\mathbf{W}^{(1)}\|_2 + \ldots + \|\mathbf{W}^{(\ell)}\|_2)\right) \\
&\leq \frac{2}{\gamma}(1 + \sqrt{d}B_w)^{L-\ell}\left(\sqrt{d} + \sqrt{d}B_x(\sqrt{d}B_w)^\ell\right)(\|\mathbf{W}^{(1)}\|_2 + \ldots + \|\mathbf{W}^{(\ell)}\|_2) \\
&\leq \frac{2}{\gamma}\sqrt{d}(1 + B_x)(1 + \sqrt{d}B_w)^L\|\Delta\boldsymbol{\theta}\|_2.
\end{aligned}
$$

(101)

By plugging it back, we have

$$
\begin{aligned}
\Delta d_{\max}^{(\ell)} &\leq \sqrt{d}B_w\Delta d_{\max}^{(\ell+1)} + \frac{2}{\gamma}\sqrt{d}(B_x + 1)(1 + \sqrt{d}B_w)^L\|\Delta\boldsymbol{\theta}\|_2 \\
&\leq \sqrt{d}B_w\left(\sqrt{d}B_w\Delta d_{\max}^{(\ell+2)} + \frac{2}{\gamma}\sqrt{d}(B_x + 1)(1 + \sqrt{d}B_w)^L\|\Delta\boldsymbol{\theta}\|_2\right) \\
&\quad + \frac{2}{\gamma}\sqrt{d}(B_x + 1)(1 + \sqrt{d}B_w)^L\|\Delta\boldsymbol{\theta}\|_2 \\
&\leq \left(1 + \sqrt{d}B_w + \ldots + (1 + \sqrt{d}B_w)^{L-\ell-1}\right) \cdot \frac{2}{\gamma}\sqrt{d}(B_x + 1)(1 + \sqrt{d}B_w)^L\|\Delta\boldsymbol{\theta}\|_2 \\
&\quad + (1 + \sqrt{d}B_w)^{L-\ell+1}\Delta d_{\max}^{L+1} \\
&\leq (1 + \sqrt{d}B_w)^{L-\ell} \cdot \frac{2}{\gamma}L\sqrt{d}(B_x + 1)(1 + \sqrt{d}B_w)^L\|\Delta\boldsymbol{\theta}\|_2 + (1 + \sqrt{d}B_w)^{L-\ell}\Delta d_{\max}^{L+1} \\
&= (1 + \sqrt{d}B_w)^{L-\ell}\left(\frac{2}{\gamma}L\sqrt{d}(B_x + 1)(1 + \sqrt{d}B_w)^L\|\Delta\boldsymbol{\theta}\|_2 + \Delta d_{\max}^{L+1}\right).
\end{aligned}
$$

(102)

Let first explicit upper bound $\Delta d_{\max}^{(L+1)}$. By definition, we can write $\Delta d_{\max}^{(L+1)}$ as

$$
\begin{aligned}
\Delta d_{\max}^{(L+1)} &= \max_i \|\mathbf{d}_i^{(L+1)} - \tilde{\mathbf{d}}_i^{(L+1)}\|_2 \\
&= \max_i \left\| \frac{\partial \mathrm{Loss}(f(\mathbf{h}_i^{(L)}), y_i)}{\partial \mathbf{h}_i^{(L)}} - \frac{\partial \mathrm{Loss}(\tilde{f}(\tilde{\mathbf{h}}_i^{(L)}), y_i)}{\partial \tilde{\mathbf{h}}_i^{(L)}} \right\|_2 \\
&\underset{(a)}{\leq} \frac{2}{\gamma} \max_i \left\| \frac{\partial f(\mathbf{h}_i^{(L)})}{\partial \mathbf{h}_i^{(L)}} - \frac{\partial f(\tilde{\mathbf{h}}_i^{(L)})}{\partial \tilde{\mathbf{h}}_i^{(L)}} \right\|_2 \\
&\underset{(b)}{\leq} \frac{2}{\gamma} \Big( \sqrt{d}B_x(1 + \sqrt{d}B_w)^{L-1}(\|\Delta\mathbf{W}^{(1)}\|_2 + \ldots + \|\Delta\mathbf{W}^{(L)}\|_2) + \big(B_x(1 + \sqrt{d}B_w)^L + 1\big)\Delta\mathbf{v} \Big) \\
&\leq \frac{2}{\gamma} \Big( \sqrt{d}B_x(1 + \sqrt{d}B_w)^L + 1 \Big) \Big( \|\Delta\mathbf{W}^{(1)}\|_2 + \ldots + \|\Delta\mathbf{W}^{(L)}\|_2 + \|\Delta\mathbf{v}\|_2 \Big) \\
&= \frac{2}{\gamma} \Big( \sqrt{d}B_x(1 + \sqrt{d}B_w)^L + 1 \Big) \|\Delta\boldsymbol{\theta}\|_2,
\end{aligned}
\tag{103}
$$

where inequality $(a)$ is due to the fact that $\nabla\mathrm{Loss}(z, y)$ is $\frac{2}{\gamma}$-Lipschitz continuous with respect to $z$, and inequality $(b)$ follows from Lemma 12 and Lemma 13. Therefore, we have

$$
\begin{aligned}
\Delta d_{\max}^{(\ell)} &\leq (1 + \sqrt{d}B_w)^{L-\ell} \Big( \frac{2}{\gamma}L\sqrt{d}(B_x + 1)(1 + \sqrt{d}B_w)^L\|\Delta\boldsymbol{\theta}\|_2 + \Delta d_{\max}^{L+1} \Big) \\
&\leq (1 + \sqrt{d}B_w)^{L-\ell}\frac{2}{\gamma} \Big( L\sqrt{d}(B_x + 1)(1 + \sqrt{d}B_w)^L + \sqrt{d}B_x(1 + \sqrt{d}B_w)^L + 1 \Big) \|\Delta\boldsymbol{\theta}\|_2 \\
&\leq (1 + \sqrt{d}B_w)^{L-\ell}\frac{2}{\gamma} \Big( (L+1)\sqrt{d}(B_x + 1)(1 + \sqrt{d}B_w)^L + 1 \Big) \|\Delta\boldsymbol{\theta}\|_2.
\end{aligned}
\tag{104}
$$

## G.5   Proof of Lemma 16

First By the definition of $\mathbf{G}^{(\ell)}$, $\ell \in [L]$, we have

$$
\begin{aligned}
\|\mathbf{G}^{(\ell)}\|_2 &= \frac{1}{m}\|[\mathbf{L}\mathbf{H}^{(\ell-1)}]^\top\mathbf{D}^{(\ell)} \odot \sigma'(\mathbf{Z}^{(\ell)})\|_2 \\
&\leq \frac{1}{m}\|[\mathbf{L}\mathbf{H}^{(\ell-1)}]^\top\mathbf{D}^{(\ell)}\|_2 \\
&\leq \sqrt{d}h_{\max}^{(\ell-1)}d_{\max}^{(\ell)} \\
&\underset{(a)}{\leq} \frac{2}{\gamma}\sqrt{d}(B_x + 1)(1 + \sqrt{d}B_w)^L,
\end{aligned}
\tag{105}
$$

where inequality $(a)$ follows from Lemma 12 and Lemma 15, and the fact that loss function is $\frac{2}{\gamma}$-Lipschitz continuous.

Similarly, by the definition of $\mathbf{G}^{(L+1)}$, we have

$$
\|\mathbf{G}^{(L+1)}\|_2 \underset{(a)}{\leq} \frac{2}{\gamma}h_{\max}^{(L)} \underset{(b)}{\leq} \frac{2}{\gamma}(B_x + 1)(1 + \sqrt{d}B_w)^L,
\tag{106}
$$

which is smaller then the right hind side of Eq. 105, and inequality $(a)$ is due to the fact that loss function is $\frac{2}{\gamma}$-Lipschitz continuous and inequality $(b)$ follows from Lemma 12.

By combining the above two inequalities together, we have

$$
\sum_{\ell=1}^{L+1} \|\mathbf{G}^{(\ell)}\|_2 \leq \frac{2}{\gamma}(L+1)\sqrt{d}(B_x + 1)(1 + \sqrt{d}B_w)^L.
\tag{107}
$$

Furthermore, we can upper bound the difference between gradients computed on two different set of weight parameters for the $\ell$th layer as

$$
\begin{aligned}
\|\mathbf{G}^{(\ell)} - \tilde{\mathbf{G}}^{(\ell)}\|_2 &= \frac{1}{m}\|[\mathbf{L}\mathbf{H}^{(\ell-1)}]^\top \mathbf{D}^{(\ell+1)} \odot \sigma'(\mathbf{Z}^{(\ell)}) - [\mathbf{L}\tilde{\mathbf{H}}^{(\ell-1)}]^\top \tilde{\mathbf{D}}^{(\ell+1)} \odot \sigma'(\tilde{\mathbf{Z}}^{(\ell)})\|_2 \\
&\leq \frac{1}{m}\|[\mathbf{L}\mathbf{H}^{(\ell-1)}]^\top \mathbf{D}^{(\ell+1)} \odot \sigma'(\mathbf{Z}^{(\ell)}) - [\mathbf{L}\tilde{\mathbf{H}}^{(\ell-1)}]^\top \mathbf{D}^{(\ell+1)} \odot \sigma'(\mathbf{Z}^{(\ell)})\|_2 \\
&\quad + \frac{1}{m}\|[\mathbf{L}\tilde{\mathbf{H}}^{(\ell-1)}]^\top \mathbf{D}^{(\ell+1)} \odot \sigma'(\mathbf{Z}^{(\ell)}) - [\mathbf{L}\tilde{\mathbf{H}}^{(\ell-1)}]^\top \tilde{\mathbf{D}}^{(\ell+1)} \odot \sigma'(\mathbf{Z}^{(\ell)})\|_2 \\
&\quad + \frac{1}{m}\|[\mathbf{L}\tilde{\mathbf{H}}^{(\ell-1)}]^\top \tilde{\mathbf{D}}^{(\ell+1)} \odot \sigma'(\mathbf{Z}^{(\ell)}) - [\mathbf{L}\tilde{\mathbf{H}}^{(\ell-1)}]^\top \tilde{\mathbf{D}}^{(\ell+1)} \odot \sigma'(\tilde{\mathbf{Z}}^{(\ell)})\|_2 \\
&\overset{(a)}{\leq} \frac{1}{m}\|[\mathbf{L}(\mathbf{H}^{(\ell-1)} - \tilde{\mathbf{H}}^{(\ell-1)})]^\top \mathbf{D}^{(\ell+1)}\|_2 + \frac{1}{m}\|[\mathbf{L}\tilde{\mathbf{H}}^{(\ell-1)}]^\top (\mathbf{D}^{(\ell+1)} - \tilde{\mathbf{D}}^{(\ell+1)})\|_2 \\
&\quad + \frac{1}{m}\|[\mathbf{L}\tilde{\mathbf{H}}^{(\ell-1)}]^\top \tilde{\mathbf{D}}^{(\ell+1)} \odot \big(\sigma'(\mathbf{Z}^{(\ell)}) - \sigma'(\tilde{\mathbf{Z}}^{(\ell)})\big)\|_2 \\
&\overset{(b)}{\leq} \max_i \|[[\mathbf{L}(\mathbf{H}^{(\ell-1)} - \tilde{\mathbf{H}}^{(\ell-1)})]^\top \mathbf{D}^{(\ell+1)}]_{i,:}\|_2 \\
&\quad + \max_i \|[[\mathbf{L}\tilde{\mathbf{H}}^{(\ell-1)}]^\top (\mathbf{D}^{(\ell+1)} - \tilde{\mathbf{D}}^{(\ell+1)})]_{i,:}\|_2 \\
&\quad + \max_i \|[\mathbf{L}\tilde{\mathbf{H}}^{(\ell-1)}]^\top \tilde{\mathbf{D}}^{(\ell+1)}\|_2 \|\mathbf{Z}^{(\ell)} - \tilde{\mathbf{Z}}^{(\ell)}\|_2 \\
&\leq \sqrt{d}\Big( \underbrace{\Delta h_{\max}^{(\ell-1)} d_{\max}^{(\ell+1)}}_{(A)} + \underbrace{h_{\max}^{(\ell-1)} \Delta d_{\max}^{(\ell+1)}}_{(B)} + \underbrace{h_{\max}^{(\ell-1)} d_{\max}^{(\ell+1)} \Delta z_{\max}^{(\ell)}}_{(C)} \Big),
\end{aligned}
$$
(108)

where inequality $(a)$ and $(b)$ is due to the fact that the gradient of ReLU activation function is element-wise either 0 or 1.

By plugging the result from Lemma 12, Lemma 13, Lemma 15, and letting $C_1 = 1 + \sqrt{d}B_w$ and $C_2 = \sqrt{d}(B_x + 1)$ we can upper bound $(A)$ as

$$
\begin{aligned}
\Delta h_{\max}^{(\ell-1)} d_{\max}^{(\ell+1)} &\leq \sqrt{d}B_x(1 + \sqrt{d}B_w)^{\ell-2} \cdot \frac{2}{\gamma}(1 + \sqrt{d}B_w)^{L-\ell} \\
&\leq \frac{2}{\gamma}\sqrt{d}B_x(1 + \sqrt{d}B_w)^L \|\Delta\boldsymbol{\theta}\|_2 \\
&\leq \frac{2}{\gamma}C_1^L C_2 \|\Delta\boldsymbol{\theta}\|_2,
\end{aligned}
$$
(109)

upper bound $(B)$ as

$$
\begin{aligned}
h_{\max}^{(\ell-1)} \Delta d_{\max}^{(\ell+1)} &\leq B_x(\max\{1, \sqrt{d}B_w\})^L \frac{2}{\gamma}\Big((L+1)\sqrt{d}(B_x+1)(\max\{1, \sqrt{d}B_w\})^L + 1\Big)\|\Delta\boldsymbol{\theta}\|_2 \\
&\leq \frac{2}{\gamma}C_1^L C_2\Big((L+1)C_1^L C_2 + 1\Big)\|\Delta\boldsymbol{\theta}\|_2,
\end{aligned}
$$
(110)

and upper bound $(C)$ as

$$
\begin{aligned}
h_{\max}^{(\ell-1)} d_{\max}^{(\ell+1)} \Delta z_{\max}^{(\ell)} &\leq B_x(1 + \sqrt{d}B_w)^{\ell-1} \cdot \frac{2}{\gamma}(1 + \sqrt{d}B_w)^{L-\ell} \cdot \sqrt{d}B_x(1 + \sqrt{d}B_w)^{\ell-1}\|\Delta\boldsymbol{\theta}\|_2 \\
&\leq \frac{2}{\gamma}\sqrt{d}B_x^2(1 + \sqrt{d}B_w)^{2L}\|\Delta\boldsymbol{\theta}\|_2 \\
&\leq \frac{2}{\gamma}(C_1^L C_2)^2 \|\Delta\boldsymbol{\theta}\|_2.
\end{aligned}
$$
(111)

By combining the results above, we can upper bound $\Delta\mathbf{G}^{(\ell)}$ as

$$
\begin{aligned}
\|\mathbf{G}^{(\ell)} - \tilde{\mathbf{G}}^{(\ell)}\|_2 &\leq \sqrt{d}\Big(\Delta h_{\max}^{(\ell-1)} d_{\max}^{(\ell+1)} + h_{\max}^{(\ell-1)} \Delta d_{\max}^{(\ell+1)} + h_{\max}^{(\ell-1)} d_{\max}^{(\ell+1)} \cdot \Delta z_{\max}^{(\ell)}\Big) \\
&\leq \frac{2}{\gamma}C_1^L C_2\Big((L+2)C_1^L C_2 + 2\Big)\|\Delta\boldsymbol{\theta}\|_2.
\end{aligned}
$$
(112)

By plugging the result from Lemma 12, Lemma 13, Lemma 15, we can upper bound $\Delta\mathbf{G}^{(\ell)}$ as

$$\|\Delta\mathbf{G}^{(\ell)}\|_2 \leq \sqrt{d}\Big(\Delta h_{\max}^{(\ell-1)} d_{\max}^{(\ell+1)} + h_{\max}^{(\ell-1)}\Delta d_{\max}^{(\ell+1)}\Big)$$

$$\leq \frac{2}{\gamma}\Big(\sqrt{d}B_x(1+\sqrt{d}B_w)^L + (1+\sqrt{d}B_w) + \sqrt{d}\Big)\|\Delta\boldsymbol{\theta}\|_2 \cdot \sqrt{d}B_x(1+\sqrt{d}B_w)^L. \tag{113}$$

Similarly, we can upper bound the difference between gradient for the weight parameters of the binary classifier as

$$\|\mathbf{G}^{(L+1)} - \tilde{\mathbf{G}}^{(L+1)}\|_2 \underset{(a)}{\leq} \frac{2}{\gamma}\Delta h_{\max}^{(L)}$$

$$\leq \frac{2}{\gamma}\sqrt{d}B_x(1+\sqrt{d}B_w)^{L-1}(\|\mathbf{W}^{(1)}\|_2 + \ldots + \|\mathbf{W}^{(L)}\|_2), \tag{114}$$

where $(a)$ is due to the fact that loss function is $\frac{2}{\gamma}$-Lipschitz continuous.

Therefore, by combining the above inequalites, we have

$$\sum_{\ell=1}^{L}\|\Delta\mathbf{G}^{(\ell)}\|_2 \leq \frac{2}{\gamma}(L+1)\Big(\sqrt{d}B_x(1+\sqrt{d}B_w)^L + (1+\sqrt{d}B_w) + \sqrt{d}\Big)\|\Delta\boldsymbol{\theta}\|_2 \cdot \sqrt{d}B_x(1+\sqrt{d}B_w)^L. \tag{115}$$

# H Generalization bound for APPNP

In the following, we provide detailed proof on the generalization bound of APPNP. Recall that the update rule of APPNP is defined as

$$\mathbf{H}^{(\ell)} = \alpha\mathbf{L}\mathbf{H}^{(\ell-1)} + (1-\alpha)\mathbf{H}^{(0)}, \ \mathbf{H}^{(0)} = \mathbf{W}\mathbf{X}. \tag{116}$$

The training of APPNP is an empirical risk minimization with respect to weight parameters $\boldsymbol{\theta} = \{\mathbf{W}^{(1)}, \ldots, \mathbf{W}^{(L)}, \boldsymbol{v}\}$, i.e.,

$$\mathcal{L}(\boldsymbol{\theta}) = \frac{1}{m}\sum_{i=1}^{m}\Phi_\gamma(-p(f(\mathbf{h}_i^{(L)}), y_i)), \ f(\mathbf{h}_i^{(L)}) = \tilde{\sigma}(\boldsymbol{v}^\top\mathbf{h}_i^{(L)}), \tag{117}$$

where $\mathbf{h}_i^{(L)}$ is the node representation of the $i$th node at the final layer, $f(\mathbf{h}_i^{(L)})$ is the predicted label for the $i$th node, $\tilde{\sigma}(x) = \frac{1}{\exp(-x)+1}$ is the sigmoid function, and loss function $\Phi_\gamma(-p(z,y))$ is $\frac{2}{\gamma}$-Lipschitz continuous with respect to its first input $z$. For simplification, we will use $\text{Loss}(z,y)$ denote $\Phi_\gamma(-p(z,y))$ in the proof.

To establish the generalization of APPNP as stated in Theorem 3, we utilize the result on transductive uniform stability from [17] (Theorem 8 in Appendix F). Then, in Lemma 17, we derive the uniform stability constant for APPNP, i.e., $\epsilon_{\text{APPNP}}$.

**Lemma 17.** *The uniform stability constant for APPNP is computed as* $\epsilon_{APPNP} = \frac{2\eta\rho_f G_f}{m}\sum_{t=1}^{T}(1+\eta L_F)^{t-1}$ *where*

$$\rho_f = C_1 B_w, \ G_f = \frac{4}{\gamma}C_1, L_f = \frac{4}{\gamma}C_1(C_1 C_2 + 1),$$

$$C_1 = B_d^\alpha B_x, \ C_2 = \max\{1, B_w\}, \ B_d^\alpha = (1-\alpha)\sum_{\ell=1}^{L}(\alpha\sqrt{d})^{\ell-1} + (\alpha\sqrt{d})^L. \tag{118}$$

By plugging the result in Lemma 17 back to Theorem 8, we establish the generalization bound for APPNP.

The key idea of the proof is to decompose the change of the APPNP output into two terms (in Lemma 5) which depend on

- (Lemma 18) The maximum change of node representation $\Delta h_{\max}^{(L)} = \max_i \|[\mathbf{H}^{(L)} - \tilde{\mathbf{H}}^{(L)}]_{i,:}\|_2$,

- (Lemma 19) The maximum node representation $h_{\max}^{(L)} = \max_i \|[\mathbf{H}^{(L)}]_{i,:}\|_2$.

**Lemma 18** (Upper bound of $h_{\max}^{(L)}$ for **APPNP**). *Let suppose Assumption 1 hold. Then, the maximum node embeddings for any node at the $\ell$th layer is bounded by*

$$h_{\max}^{(L)} \leq B_d^\alpha B_x B_w, \tag{119}$$

*where $B_d^\alpha = (1 - \alpha) \sum_{\ell=1}^{L} (\alpha\sqrt{d})^{\ell-1} + (\alpha\sqrt{d})^L$.*

**Lemma 19** (Upper bound of $\Delta h_{\max}^{(L)}$ for **APPNP**). *Let suppose Assumption 1 hold. Then, the maximum change between the node embeddings on two different set of weight parameters for any node at the $\ell$th layer is bounded by*

$$\Delta h_{\max}^{(L)} \leq B_d^\alpha B_x \|\Delta \mathbf{W}\|_2, \tag{120}$$

*where $B_d^\alpha = (1 - \alpha) \sum_{\ell=1}^{L} (\alpha\sqrt{d})^{\ell-1} + (\alpha\sqrt{d})^L$ and $\Delta \mathbf{W}^{(\ell)} = \mathbf{W}^{(\ell)} - \tilde{\mathbf{W}}^{(\ell)}$.*

Then, in Lemma 20, we decompose the change of the model parameters into two terms which depend on

- The change of gradient with respect to the $\ell$th layer weight matrix $\|\Delta \mathbf{G}^{(\ell)}\|_2$,

- The gradient with respect to the $\ell$th layer weight matrix $\|\mathbf{G}^{(\ell)}\|_2$,

where $\|\Delta \mathbf{G}^{(\ell)}\|_2$ reflects the smoothness of APPNP model and $\|\mathbf{G}^{(\ell)}\|_2$ corresponds the upper bound of gradient.

**Lemma 20** (Upper bound of $\|\mathbf{G}^{(\ell)}\|_2, \|\Delta \mathbf{G}^{(\ell)}\|_2$ for **APPNP**). *Let suppose Assumption 1 hold. Then, the gradient and the maximum change between gradients on two different set of weight parameters are bounded by*

$$\sum_{\ell=1}^{2} \|\mathbf{G}^{(\ell)}\|_2 \leq \frac{4}{\gamma} B_d^\alpha B_x,$$
$$\sum_{\ell=1}^{2} \|\Delta \mathbf{G}^{(1)}\|_2 \leq \frac{4}{\gamma} B_d^\alpha B_x \Big( B_d^\alpha B_x \max\{1, B_w\} + 1 \Big) \|\Delta \boldsymbol{\theta}\|_2, \tag{121}$$

*where $\|\Delta \boldsymbol{\theta}\|_2 = \|\boldsymbol{v} - \tilde{\boldsymbol{v}}\|_2 + \|\mathbf{W} - \tilde{\mathbf{W}}\|_2$ denotes the change of two set of parameters.*

Equipped with above intermediate results, we now proceed to prove Lemma 17.

*Proof.* Recall that our goal is to explore the impact of different GCN structures on the uniform stability constant $\epsilon_{\text{APPNP}}$, which is a function of $\rho_f$, $G_f$, and $L_f$. By Lemma 19, we know that the function $f$ is $\rho_f$-Lipschitz continuous, with $\rho_f = B_d^\alpha B_x$.

By Lemma 20, we know that the function $f$ is $L_f$-smoothness, and the gradient of each weight matrices is bounded by $G_f$, with

$$G_f \leq \frac{4}{\gamma} B_d^\alpha B_x, \ L_f \leq \frac{4}{\gamma} B_d^\alpha B_x \Big( B_d^\alpha B_x \max\{1, B_w\} + 1 \Big).$$

By plugging $\epsilon_{\text{APPNP}}$ into Theorem 3, we obtain the generalization bound of APPNP. $\qquad \square$

## H.1 Proof of Lemma 18

By the definition of $h_{\max}^{(L)}$, we have

$$
\begin{aligned}
h_{\max}^{(L)} &= \max_i \|[\alpha \mathbf{L}\mathbf{H}^{(L-1)} + (1-\alpha)\mathbf{H}^{(0)}]_{i,:}\|_2 \\
&\leq \alpha \max_i \|[\mathbf{L}\mathbf{H}^{(L-1)}]_{i,:}\|_2 + (1-\alpha) \max_i \|[\mathbf{H}^{(0)}]_{i,:}\|_2 \\
&= \alpha \max_i \left\| \sum_{j=1}^N L_{i,j} \mathbf{h}_j^{(L-1)} \right\|_2 + (1-\alpha)B_w B_x \\
&\underset{(a)}{\leq} \alpha \sqrt{d} h_{\max}^{(L-1)} + (1-\alpha)B_w B_x \\
&\underset{(b)}{\leq} \left( (1-\alpha) \sum_{\ell=1}^L (\alpha\sqrt{d})^{\ell-1} + (\alpha\sqrt{d})^L \right) B_x B_w,
\end{aligned}
\tag{122}
$$

where inequality $(a)$ is follows from Lemma 31 and inequality $(b)$ can be obtained by recursively applying $(a)$.

Let denote $B_d^\alpha = (1-\alpha)\sum_{\ell=1}^L (\alpha\sqrt{d})^{\ell-1} + (\alpha\sqrt{d})^L$, then we have

$$
h_{\max}^{(L)} \leq B_d^\alpha B_x B_w.
\tag{123}
$$

## H.2 Proof of Lemma 19

By the definition of $\Delta h_{\max}^{(L)}$, we have

$$
\begin{aligned}
\Delta h_{\max}^{(L)} &= \max_i \|\mathbf{h}_i^{(L)} - \tilde{\mathbf{h}}_i^{(L)}\|_2 \\
&= \max_i \|[(\alpha \mathbf{L}\mathbf{H}^{(L-1)} + (1-\alpha)\mathbf{H}^{(0)}) - (\alpha \mathbf{L}\tilde{\mathbf{H}}^{(L-1)} + (1-\alpha)\tilde{\mathbf{H}}^{(0)})]_{i,:}\|_2 \\
&\leq \alpha \max_i \|[\mathbf{L}(\mathbf{H}^{(L-1)} - \tilde{\mathbf{H}}^{(L-1)})]_{i,:}\|_2 + (1-\alpha) \max_i \|[\mathbf{H}^{(0)} - \tilde{\mathbf{H}}^{(0)}]_{i,:}\|_2 \\
&\underset{(a)}{\leq} \alpha \sqrt{d} \Delta h_{\max}^{(L-1)} + (1-\alpha)B_x \|\Delta\mathbf{W}\|_2 \\
&\underset{(b)}{\leq} \left( (1-\alpha) \sum_{L=1}^L (\alpha\sqrt{d})^{\ell-1} + (\alpha\sqrt{d})^\ell \right) B_x \|\Delta\mathbf{W}\|_2,
\end{aligned}
\tag{124}
$$

where inequality $(a)$ is follows from Lemma 31 and inequality $(b)$ can be obtained by recursively applying $(a)$.

Let denote $B_d^\alpha = (1-\alpha)\sum_{\ell=1}^L (\alpha\sqrt{d})^{\ell-1} + (\alpha\sqrt{d})^L$, then we have

$$
\Delta h_{\max}^{(L)} \leq B_d^\alpha B_x \|\Delta\mathbf{W}\|_2.
\tag{125}
$$

## H.3 Proof of Lemma 20

By definition, we know that $\mathbf{H}^{(L)}$ is computed as

$$
\mathbf{H}^{(L)} = \left( (\alpha\mathbf{L})^L + (1-\alpha)(1 + (\alpha\mathbf{L}) + \ldots + (\alpha\mathbf{L})^{L-1}) \right) \mathbf{X}\mathbf{W}.
\tag{126}
$$

Therefore, the gradient with respect to weight $\mathbf{W}$ is computed as

$$
\begin{aligned}
\|\mathbf{G}^{(1)}\|_2 &\underset{(a)}{\leq} \frac{2}{\gamma} \max_i \left\| \frac{\partial f(\mathbf{h}_i)}{\partial \mathbf{W}} \right\|_2 \\
&= \frac{2}{\gamma} \max_i \left\| \left[ \left( (\alpha\mathbf{L})^L + (1-\alpha)(1 + (\alpha\mathbf{L}) + \ldots + (\alpha\mathbf{L})^{L-1}) \right) \mathbf{X} \right]_{i,:} \right\|_2 \\
&\leq \frac{2}{\gamma} \left( (\alpha\sqrt{d})^L + (1-\alpha)(1 + (\alpha\sqrt{d}) + \ldots + (\alpha\sqrt{d})^{L-1}) \right) B_x \\
&\leq \frac{2}{\gamma} B_d^\alpha B_x,
\end{aligned}
\tag{127}
$$

where inequality $(a)$ is due to the fact that the loss function is $\frac{2}{\gamma}$-Lipschitz continuous, and $B_d^\alpha = (1-\alpha)\sum_{\ell=1}^{L}(\alpha\sqrt{d})^{\ell-1} + (\alpha\sqrt{d})^L$. Besides, the gradient with respect to weight $v$ is computed as

$$
\begin{aligned}
\|\mathbf{G}^{(2)}\|_2 &= \left\| \frac{\partial \Phi_\gamma(-p(f(\mathbf{h}_i^{(L)}), y_i))}{\partial v} \right\|_2 \\
&\leq \frac{2}{\gamma} h_{\max}^{(L)} \\
&\leq \frac{2}{\gamma} B_d^\alpha B_x B_w.
\end{aligned}
\tag{128}
$$

Combine the result above, we have $\sum_{\ell=1}^{2}\|\mathbf{G}^{(\ell)}\|_2 \leq \frac{4}{\gamma}B_d^\alpha B_x B_w$. Then, the difference between two gradient computed on two different set of weight parameters is computed as

$$
\begin{aligned}
\|\mathbf{G}^{(1)} - \tilde{\mathbf{G}}^{(1)}\|_2 &\leq \frac{2}{\gamma} \max_i \left\| \frac{\partial f(\mathbf{h}_i^{(L)})}{\partial \mathbf{h}_i^{(L)}} \frac{\partial \mathbf{h}_i^{(L)}}{\partial \mathbf{W}} - \frac{\partial f(\tilde{\mathbf{h}}_i^{(L)})}{\partial \tilde{\mathbf{h}}_i^{(L)}} \frac{\partial \tilde{\mathbf{h}}_i^{(L)}}{\partial \tilde{\mathbf{W}}} \right\|_2 \\
&= \frac{2}{\gamma} B_x B_d^\alpha \left\| \frac{\partial f(\mathbf{h}_i^{(L)})}{\partial \mathbf{h}_i^{(L)}} - \frac{\partial f(\tilde{\mathbf{h}}_i^{(L)})}{\partial \tilde{\mathbf{h}}_i^{(L)}} \right\|_2 \\
&\underset{(a)}{\leq} \frac{2}{\gamma} B_x B_d^\alpha \left( \Delta h_{\max}^{(L)} + (h_{\max}^{(L)} + 1)\|\Delta v\|_2 \right) \\
&\underset{(b)}{\leq} \frac{2}{\gamma} B_x B_d^\alpha \left( B_x B_d^\alpha \|\Delta \mathbf{W}\|_2 + (B_x B_d^\alpha B_w + 1)\|\Delta v\|_2 \right) \\
&\leq \frac{2}{\gamma} B_x B_d^\alpha \left( B_x B_d^\alpha \max\{1, B_w\} + 1 \right)\|\Delta \boldsymbol{\theta}\|_2,
\end{aligned}
\tag{129}
$$

where inequality $(a)$ follows from Lemma 5 and inequality $(b)$ follows from Lemma 19 and Lemma 18.

Similarly, the difference of gradient w.r.t. weight $v$ is computed as

$$
\begin{aligned}
\|\mathbf{G}^{(2)} - \tilde{\mathbf{G}}^{(2)}\|_2 &= \left\| \frac{\partial \Phi_\gamma(-p(f(\mathbf{h}_i^{(L)}), y_i))}{\partial v} - \frac{\partial \Phi_\gamma(-p(\tilde{f}(\tilde{\mathbf{h}}_i^{(L)}), y_i))}{\partial \tilde{v}} \right\|_2 \\
&\leq \frac{2}{\gamma} \Delta h_{\max}^{(L)} \\
&\leq \frac{2}{\gamma} B_d^\alpha B_x \|\Delta \mathbf{W}\|_2.
\end{aligned}
\tag{130}
$$

Combining result above yields $\sum_{\ell=1}^{2}\|\mathbf{G}^{(\ell)} - \tilde{\mathbf{G}}^{(\ell)}\|_2 \leq \frac{4}{\gamma}B_x B_d^\alpha \left( B_x B_d^\alpha \max\{1, B_w\} + 1 \right)\|\Delta \boldsymbol{\theta}\|_2$ as desired.

# I   Generalization bound for GCNII

In the following, we provide generalization bound of the GCNII. Recall that the update rule of GCNII is defined as

$$
\mathbf{H}^{(\ell)} = \sigma\left( \left(\alpha \mathbf{L}\mathbf{H}^{(\ell)} + (1-\alpha)\mathbf{X}\right)\left(\beta \mathbf{W}^{(\ell)} + (1-\beta)\mathbf{I}_N\right) \right),
\tag{131}
$$

where $\sigma(\cdot)$ is ReLU activation function. Although ReLU function $\sigma(x)$ is not differentiable when $x = 0$, for analysis purpose, we suppose the $\sigma'(0) = 0$ which is also used in optimization.

The training of GCNII is an empirical risk minimization with respect to a set of parameters $\boldsymbol{\theta} = \{\mathbf{W}^{(1)}, \ldots, \mathbf{W}^{(L)}, v\}$, i.e.,

$$
\mathcal{L}(\boldsymbol{\theta}) = \frac{1}{m}\sum_{i=1}^{m} \Phi_\gamma(-p(f(\mathbf{h}_i^{(L)}), y_i)), \quad f(\mathbf{h}_i^{(L)}) = \tilde{\sigma}(v^\top \mathbf{h}_i^{(L)}),
\tag{132}
$$

where $\mathbf{h}_i^{(L)}$ is the node representation of the $i$th node at the final layer, $f(\mathbf{h}_i^{(L)})$ is the predicted label for the $i$th node, $\tilde{\sigma}(x) = \frac{1}{\exp(-x)+1}$ is the sigmoid function, and loss function $\Phi_\gamma(-p(z,y))$ is $\frac{2}{\gamma}$-Lipschitz continuous with respect to its first input $z$. For simplification, let $\mathrm{Loss}(z,y)$ denote $\Phi_\gamma(-p(z,y))$ in the proof.

To establish the generalization of GCNII as stated in Theorem 3, we utilize the result on transductive uniform stability from [17] (Theorem 8 in Appendix F). Then, in Lemma 21, we derive the uniform stability constant for GCNII, i.e., $\epsilon_{\mathrm{GCNII}}$.

**Lemma 21.** *The uniform stability constant for GCNII is computed as* $\epsilon_{\mathbf{GCNII}} = \frac{2\eta\rho_f G_f}{m}\sum_{t=1}^T(1+\eta L_F)^{t-1}$ *where*

$$
\begin{aligned}
&\rho_f = \beta C_1^L C_2, \; G_f = \beta\frac{2}{\gamma}(L+1)C_1^L C_2, \\
&L_f = \alpha\beta\frac{2}{\gamma}(L+1)\sqrt{d}C_1^L C_2\Big((\alpha\beta L + \beta B_w + 2)C_1^L C_2 + 2\beta\Big), \\
&C_1 = \max\{1, \alpha\sqrt{d}B_w^\beta\}, \; C_2 = \sqrt{d} + B_{\ell,d}^{\alpha,\beta}B_x, \\
&B_w^\beta = \beta B_w + (1-\beta), \; B_{\ell,d}^{\alpha,\beta} = \max\Big\{\beta\big((1-\alpha)L + \alpha\sqrt{d}\big), (1-\alpha)LB_w^\beta + 1\Big\}.
\end{aligned}
\tag{133}
$$

By plugging the result in Lemma 21 back to Theorem 8, we establish the generalization bound for GCNII.

The key idea of the proof is to decompose the change of the GCNII output into two terms (in Lemma 5) which depend on

- (Lemma 22) The maximum change of node representation $\Delta h_{\max}^{(\ell)} = \max_i \|[\mathbf{H}^{(\ell)} - \tilde{\mathbf{H}}^{(\ell)}]_{i,:}\|_2$,

- (Lemma 23) The maximum node representation $h_{\max}^{(\ell)} = \max_i \|[\mathbf{H}^{(\ell)}]_{i,:}\|_2$.

**Lemma 22** (Upper bound of $h_{\max}^{(\ell)}$ for **GCNII**). *Let suppose Assumption 1 hold. Then, the maximum node embeddings for any node at the $\ell$th layer is bounded by*

$$
h_{\max}^{(\ell)} \le \big((1-\alpha)\ell B_w^\beta + 1\big)B_x \cdot \big(\max\{1, \alpha\sqrt{d}B_w^\beta\}\big)^\ell,
\tag{134}
$$

*where $B_w^\beta = \beta B_w + (1-\beta)$ and $\Delta\mathbf{W}^{(\ell)} = \mathbf{W}^{(\ell)} - \tilde{\mathbf{W}}^{(\ell)}$.*

**Lemma 23** (Upper bound of $\Delta h_{\max}^{(\ell)}$ for **GCNII**). *Let suppose Assumption 1 hold. Then, the maximum change between the node embeddings on two different set of weight parameters for any node at the $\ell$th layer is bounded by*

$$
\Delta h_{\max}^{(\ell)} \le \beta B_x\big((1-\alpha)\ell + \alpha\sqrt{d}\big)\big(\max\{1, \alpha\sqrt{d}B_w^\beta\}\big)^{\ell-1}\Big(\|\Delta\mathbf{W}^{(1)}\|_2 + \ldots + \|\Delta\mathbf{W}^{(\ell)}\|_2\Big),
\tag{135}
$$

*where $B_w^\beta = \beta B_w + (1-\beta)$ and $\Delta\mathbf{W}^{(\ell)} = \mathbf{W}^{(\ell)} - \tilde{\mathbf{W}}^{(\ell)}$.*

Besides, in Lemma 24, we derive the upper bound on the maximum change of node embeddings before the activation function, i.e., $\Delta z_{\max}^{(\ell)} = \max_i \|[\mathbf{Z}^{(\ell)} - \tilde{\mathbf{Z}}^{(\ell)}]_{i,:}\|_2$, which will be used to derive the gradient related upper bounds.

**Lemma 24** (Upper bound of $\Delta z_{\max}^{(\ell)}$ for **GCNII**). *Let suppose Assumption 1 hold. Then, the maximum change between the node embeddings before the activation function on two different set of weight parameters for any node at the $\ell$th layer is bounded by*

$$
\Delta z_{\max}^{(\ell)} \le \beta B_x\big((1-\alpha)\ell + \alpha\sqrt{d}\big)(\max\{1, \alpha\sqrt{d}B_w^\beta\})^{\ell-1}\Big(\|\Delta\mathbf{W}^{(1)}\|_2 + \ldots + \|\Delta\mathbf{W}^{(\ell)}\|_2\Big),
\tag{136}
$$

*where $B_w^\beta = \beta B_w + (1-\beta)$ and $\Delta\mathbf{W}^{(\ell)} = \mathbf{W}^{(\ell)} - \tilde{\mathbf{W}}^{(\ell)}$.*

Then, in Lemma 25, we decompose the change of the model parameters into two terms which depend on

- The maximum change of gradient passing from $(\ell+1)$th layer to the $\ell$th layer $\Delta d_{\max}^{(\ell)} = \max_i \left\| \left[ \frac{\partial \mathbf{H}^{(\ell)}}{\partial \mathbf{H}^{(\ell-1)}} - \frac{\partial \tilde{\mathbf{H}}^{(\ell)}}{\partial \tilde{\mathbf{H}}^{(\ell-1)}} \right]_{i,:} \right\|_2$,

- The maximum gradient passing from $(\ell+1)$th layer to the $\ell$th layer $d_{\max}^{(\ell)} = \max_i \left\| \left[ \frac{\partial \mathbf{H}^{(\ell)}}{\partial \mathbf{H}^{(\ell-1)}} \right]_{i,:} \right\|_2$.

**Lemma 25** (Upper bound of $\mathbf{d}_{\max}^{(\ell)}, \Delta \mathbf{d}_{\max}^{(\ell)}$ for **GCNII**). *Let suppose Assumption 1 hold and let denote $C_1 = \max\{1, \alpha\sqrt{d}B_w^\beta\}$, $C_2 = \sqrt{d} + B_{\ell,d}^{\alpha,\beta} B_x$, $B_w^\beta = \beta B_w + (1-\beta)$, and $B_{\ell,d}^{\alpha,\beta} = \max\left\{ \beta\big((1-\alpha)L + \alpha\sqrt{d}\big), (1-\alpha)LB_w^\beta + 1 \right\}$. Then, the maximum gradient passing from layer $\ell+1$ to layer $\ell$ for any node is bounded by*

$$d_{\max}^{(\ell)} \leq \frac{2}{\gamma} C_1^{L-\ell+1} \|\Delta\boldsymbol{\theta}\|_2, \tag{137}$$

*and the maximum change between the gradient passing from layer $\ell+1$ to layer $\ell$ on two different set of weight parameters for any node is bounded by*

$$\Delta d_{\max}^{(\ell)} \leq \frac{2}{\gamma} C_1^{L-\ell}\Big( (\alpha\beta L + \beta B_w + 1)C_1^L C_2 + 1 \Big) \|\Delta\boldsymbol{\theta}\|_2, \tag{138}$$

*where $\|\Delta\boldsymbol{\theta}\|_2 = \|\boldsymbol{v} - \tilde{\boldsymbol{v}}\|_2 + \sum_{\ell-1}^L \|\mathbf{W}^{(\ell)} - \tilde{\mathbf{W}}^{(\ell)}\|_2$ denotes the change of two set of parameters.*

Finally, in Lemma 26, we decompose the change of the model parameters into two terms which depend on

- The change of gradient with respect to the $\ell$th layer weight matrix $\|\Delta\mathbf{G}^{(\ell)}\|_2$,
- The gradient with respect to the $\ell$th layer weight matrix $\|\mathbf{G}^{(\ell)}\|_2$,

where $\mathbf{G}^{(L+1)}$ denotes the gradient with respect to the weight $\boldsymbol{v}$ of the binary classifier and $\mathbf{G}^{(\ell)}$ denotes the gradient with respect to the weight matrices $\mathbf{W}^{(\ell)}$ of the graph convolutional layer. Notice that $\|\Delta\mathbf{G}^{(\ell)}\|_2$ reflect the smoothness of GCN model and $\|\mathbf{G}^{(\ell)}\|_2$ correspond the upper bound of gradient.

**Lemma 26** (Upper bound of $\mathbf{G}^{(\ell)}, \Delta\mathbf{G}^{(\ell)}$ for **GCNII**). *Let suppose Assumption 1 hold, and let $C_1 = \max\{1, \alpha\sqrt{d}B_w^\beta\}$, $C_2 = \sqrt{d} + B_{\ell,d}^{\alpha,\beta} B_x$, $B_w^\beta = \beta B_w + (1-\beta)$, and $B_{\ell,d}^{\alpha,\beta} = \max\big\{ \beta\big((1-\alpha)L + \alpha\sqrt{d}\big), (1-\alpha)LB_w^\beta + 1 \big\}$. Then, the gradient and the maximum change between gradients on two different set of weight parameters are bounded by*

$$\sum_{\ell=1}^{L+1} \|\mathbf{G}^{(\ell)}\|_2 \leq \frac{2}{\gamma}(L+1)\beta C_1^L C_2,$$

$$\sum_{\ell=1}^{L+1} \|\Delta\mathbf{G}^{(\ell)}\|_2 \leq \alpha\beta\frac{2}{\gamma}(L+1)\sqrt{d}C_1^L C_2 \Big( (\alpha\beta L + \beta B_w + 2)C_1^L C_2 + 2\beta \Big) \|\Delta\boldsymbol{\theta}\|_2, \tag{139}$$

*where $\|\Delta\boldsymbol{\theta}\|_2 = \|\boldsymbol{v} - \tilde{\boldsymbol{v}}\|_2 + \sum_{\ell-1}^L \|\mathbf{W}^{(\ell)} - \tilde{\mathbf{W}}^{(\ell)}\|_2$ denotes the change of two set of parameters.*

Equipped with above intermediate results, we now proceed to prove Lemma 21.

*Proof.* Recall that our goal is to explore the impact of different GCN structures on the uniform stability constant $\epsilon_{\text{GCNII}}$, which is a function of $\rho_f$, $G_f$, and $L_f$. Let denote $C_1 = \max\{1, \alpha\sqrt{d}B_w^\beta\}$, $C_2 = \sqrt{d} + B_{\ell,d}^{\alpha,\beta} B_x$, $B_w^\beta = \beta B_w + (1-\beta)$, and $B_{\ell,d}^{\alpha,\beta} = \max\big\{ \beta\big((1-\alpha)L + \alpha\sqrt{d}\big), (1-\alpha)LB_w^\beta + 1 \big\}$. By Lemma 23, we know that the function $f$ is $\rho_f$-Lipschitz continuous, with

$$\rho_f = \beta C_1^L C_2. \tag{140}$$

By Lemma 26, we know that the function $f$ is $L_f$-smoothness, and the gradient of each weight matrices is bounded by $G_f$, with

$$G_f = \frac{2}{\gamma}(L+1)\beta C_1^L C_2,$$

$$L_f = \alpha\beta\frac{2}{\gamma}(L+1)\sqrt{d}C_1^L C_2 \Big( (\alpha\beta L + \beta B_w + 2)C_1^L C_2 + 2\beta \Big). \tag{141}$$

where $B_{\ell,d}^{\alpha,\beta} = \max\left\{\beta\big((1-\alpha)L + \alpha\sqrt{d}\big), (1-\alpha)LB_w^\beta + 1\right\}$.

By plugging $\epsilon_{\text{GCNII}}$ into Theorem 3, we obtain the generalization bound of GCNII.

$\square$

### I.1 Proof of Lemma 22

By the definition of $h_{\max}^{(\ell)}$, we have

$$
\begin{aligned}
h_{\max}^{(\ell)} &= \max_i \|[\sigma\big((\alpha\mathbf{L}\mathbf{H}^{(\ell-1)} + (1-\alpha)\mathbf{X})(\beta\mathbf{W}^{(\ell)} + (1-\beta)\mathbf{I})\big)]_{i,:}\|_2 \\
&\underset{(a)}{\leq} \max_i \|[(\alpha\mathbf{L}\mathbf{H}^{(\ell-1)} + (1-\alpha)\mathbf{X})(\beta\mathbf{W}^{(\ell)} + (1-\beta)\mathbf{I})]_{i,:}\|_2 \\
&\leq \max_i \|[(\alpha\mathbf{L}\mathbf{H}^{(\ell-1)} + (1-\alpha)\mathbf{X})]_{i,:}\|_2 \|\beta\mathbf{W}^{(\ell)} + (1-\beta)\mathbf{I}\|_2 \\
&\leq (\beta B_w + (1-\beta)) \cdot \max_i \|[(\alpha\mathbf{L}\mathbf{H}^{(\ell-1)} + (1-\alpha)\mathbf{X})]_{i,:}\|_2 \\
&\leq (\beta B_w + (1-\beta)) \cdot \left(\alpha \max_i \|[\mathbf{L}\mathbf{H}^{(\ell-1)}]_{i,:}\|_2 + (1-\alpha) \max_i \|[\mathbf{X}]_{i,:}\|_2\right) \\
&\leq (\beta B_w + (1-\beta)) \cdot \left(\alpha\sqrt{d}h_{\max}^{(\ell-1)} + (1-\alpha)B_x\right) \\
&= \alpha\sqrt{d}(\beta B_w + (1-\beta))h_{\max}^{(\ell-1)} + (1-\alpha)B_x(\beta B_w + (1-\beta)),
\end{aligned}
\tag{142}
$$

where inequality $(a)$ is due to $\|\sigma(x)\|_2 \leq \|x\|_2$.

Let $B_w^\beta = \beta B_w + (1-\beta)$. By induction, we have

$$
h_{\max}^{(\ell)} \leq \left((1-\alpha)B_x B_w^\beta\right) \sum_{\ell'=1}^{\ell} \left(\alpha\sqrt{d}B_w^\beta\right)^{\ell'-1} + \left(\alpha\sqrt{d}B_w^\beta\right)^\ell B_x.
\tag{143}
$$

If $\alpha\sqrt{d}B_w^\beta \leq 1$ we have,

$$
h_{\max}^{(\ell)} \leq (1-\alpha)B_x B_w^\beta \ell + B_x = \left((1-\alpha)\ell B_w^\beta + 1\right)B_x,
\tag{144}
$$

and if $\alpha\sqrt{d}B_w^\beta > 1$ we have

$$
\begin{aligned}
h_{\max}^{(\ell)} &\leq \left((1-\alpha)B_x B_w^\beta\right)\ell\left(\alpha\sqrt{d}B_w^\beta\right)^{\ell-1} + \left(\alpha\sqrt{d}B_w^\beta\right)^\ell B_x \\
&\leq \left((1-\alpha)\ell B_w^\beta + 1\right)B_x \cdot \left(\alpha\sqrt{d}B_w^\beta\right)^\ell.
\end{aligned}
\tag{145}
$$

By combining results above, we have

$$
h_{\max}^{(\ell)} \leq \left((1-\alpha)\ell B_w^\beta + 1\right)B_x \cdot \left(\max\{1, \alpha\sqrt{d}B_w^\beta\}\right)^\ell.
\tag{146}
$$

## I.2 Proof of Lemma 23

By the definition of $\Delta h_{\max}^{(\ell)}$, we have

$$
\begin{aligned}
\Delta h_{\max}^{(\ell)} &= \max_i \|\mathbf{h}_i^{(\ell)} - \tilde{\mathbf{h}}_i^{(\ell)}\|_2 \\
&= \max_i \left\| \left[ \sigma\Big( \big(\alpha \mathbf{L}\mathbf{H}^{(\ell-1)} + (1-\alpha)\mathbf{X}\big)\big(\beta\mathbf{W}^{(\ell)} + (1-\beta)\mathbf{I}\big) \Big) \right.\right. \\
&\qquad \left.\left. - \sigma\Big( \big(\alpha \mathbf{L}\tilde{\mathbf{H}}^{(\ell-1)} + (1-\alpha)\mathbf{X}\big)\big(\beta\tilde{\mathbf{W}}^{(\ell)} + (1-\beta)\mathbf{I}\big) \Big) \right]_{i,:} \right\|_2 \\
&\leq \max_i \left\| \left[ \big(\alpha \mathbf{L}\mathbf{H}^{(\ell-1)} + (1-\alpha)\mathbf{X}\big)\big(\beta\mathbf{W}^{(\ell)} + (1-\beta)\mathbf{I}\big) \right.\right. \\
&\qquad \left.\left. - \big(\alpha \mathbf{L}\tilde{\mathbf{H}}^{(\ell-1)} + (1-\alpha)\mathbf{X}\big)\big(\beta\tilde{\mathbf{W}}^{(\ell)} + (1-\beta)\mathbf{I}\big) \right]_{i,:} \right\|_2 \\
&\underset{(a)}{\leq} \max_i \left\| \left[ \big(\alpha \mathbf{L}\mathbf{H}^{(\ell-1)} + (1-\alpha)\mathbf{X}\big)\big(\beta(\mathbf{W}^{(\ell)} - \tilde{\mathbf{W}}^{(\ell)})\big) \right]_{i,:} \right\|_2 \\
&\qquad + \max_i \left\| \left[ \big(\alpha\mathbf{L}(\mathbf{H}^{(\ell-1)} - \tilde{\mathbf{H}}^{(\ell-1)})\big)\big(\beta\tilde{\mathbf{W}}^{(\ell)} + (1-\beta)\mathbf{I}\big) \right]_{i,:} \right\|_2 \\
&\leq \max_i \|[\alpha\mathbf{L}\mathbf{H}^{(\ell-1)} + (1-\alpha)\mathbf{X}]_{i,:}\|_2 \cdot \beta\|\Delta\mathbf{W}^{(\ell)}\|_2 + \alpha(\beta B_w + (1-\beta)) \max_i \|[\mathbf{L}(\mathbf{H}^{(\ell-1)} - \tilde{\mathbf{H}}^{(\ell-1)})]_{i,:}\|_2 \\
&\leq \Big(\alpha\sqrt{d}h_{\max}^{(\ell-1)} + (1-\alpha)B_x\Big) \cdot \beta\|\Delta\mathbf{W}^{(\ell)}\|_2 + \alpha\big(\beta B_w + (1-\beta)\big)\sqrt{d}\Delta h_{\max}^{(\ell-1)},
\end{aligned}
$$
(147)

where $(a)$ is due to the fact that $\|\sigma(x)\|_2 \leq \|x\|_2$.

Let $B_w^\beta = \beta B_w + (1-\beta)$. Then, by induction we have

$$
\begin{aligned}
\Delta h_{\max}^{(\ell)} &\leq \alpha\sqrt{d}B_w^\beta \cdot \Delta h_{\max}^{(\ell-1)} + \beta\Big(\alpha\sqrt{d}h_{\max}^{(\ell-1)} + (1-\alpha)B_x\Big)\|\Delta\mathbf{W}^{(\ell)}\|_2 \\
&\leq (\alpha\sqrt{d}B_w^\beta)^2 \cdot \Delta h_{\max}^{(\ell-2)} + \beta\Big(\alpha\sqrt{d}h_{\max}^{(\ell-1)} + (1-\alpha)B_x\Big)\|\Delta\mathbf{W}^{(\ell)}\|_2 \\
&\qquad + \beta(\alpha\sqrt{d}B_w^\beta)\Big(\alpha\sqrt{d}h_{\max}^{(\ell-2)} + (1-\alpha)B_x\Big)\|\Delta\mathbf{W}^{(\ell-1)}\|_2 \\
&\cdots \\
&\leq \beta\Big(\alpha\sqrt{d}h_{\max}^{(\ell-1)} + (1-\alpha)B_x\Big)\|\Delta\mathbf{W}^{(\ell)}\|_2 \\
&\qquad + \beta(\alpha\sqrt{d}B_w^\beta)\Big(\alpha\sqrt{d}h_{\max}^{(\ell-2)} + (1-\alpha)B_x\Big)\|\Delta\mathbf{W}^{(\ell-1)}\|_2 + \ldots \\
&\qquad + \beta(\alpha\sqrt{d}B_w^\beta)^{\ell-1}\Big(\alpha\sqrt{d}h_{\max}^{(0)} + (1-\alpha)B_x\Big)\|\Delta\mathbf{W}^{(1)}\|_2.
\end{aligned}
$$
(148)

If $\alpha\sqrt{d}B_w^\beta \leq 1$, using Lemma 23 we have

$$
\begin{aligned}
\Delta h_{\max}^{(\ell)} &\leq \beta\Big(\alpha\sqrt{d}h_{\max}^{(\ell-1)} + (1-\alpha)B_x\Big)\|\Delta\mathbf{W}^{(\ell)}\|_2 + \beta\Big(\alpha\sqrt{d}h_{\max}^{(\ell-2)} + (1-\alpha)B_x\Big)\|\Delta\mathbf{W}^{(\ell-1)}\|_2 + \ldots \\
&\qquad + \beta\Big(\alpha\sqrt{d}h_{\max}^{(0)} + (1-\alpha)B_x\Big)\|\Delta\mathbf{W}^{(1)}\|_2 \\
&\leq \beta B_x\Big((1-\alpha)\ell + \alpha\sqrt{d}\Big)\Big(\|\Delta\mathbf{W}^{(1)}\|_2 + \ldots + \|\Delta\mathbf{W}^{(\ell)}\|_2\Big),
\end{aligned}
$$
(149)

and if $\alpha\sqrt{d}B_w^\beta > 1$, using Lemma 23 we have

$$
\begin{aligned}
\Delta h_{\max}^{(\ell)} &\leq \beta\Big(\alpha\sqrt{d}\big((1-\alpha)(\ell-1)B_w^\beta + 1\big)B_x \cdot \big(\alpha\sqrt{d}B_w^\beta\big)^{\ell-1} + (1-\alpha)B_x\Big)\|\Delta\mathbf{W}^{(\ell)}\|_2 \\
&\qquad + \beta(\alpha\sqrt{d}B_w^\beta)\Big(\alpha\sqrt{d}\big((1-\alpha)(\ell-2)B_w^\beta + 1\big)B_x \cdot \big(\alpha\sqrt{d}B_w^\beta\big)^{\ell-2} + (1-\alpha)B_x\Big)\|\Delta\mathbf{W}^{(\ell-1)}\|_2 + \ldots \\
&\qquad + \beta(\alpha\sqrt{d}B_w^\beta)^{\ell-1}\Big(\alpha\sqrt{d}B_x + (1-\alpha)B_x\Big)\|\Delta\mathbf{W}^{(1)}\|_2 \\
&\leq \beta B_x\big((1-\alpha)\ell + \alpha\sqrt{d}\big)(\alpha\sqrt{d}B_w^\beta)^{\ell-1}\Big(\|\Delta\mathbf{W}^{(1)}\|_2 + \ldots + \|\Delta\mathbf{W}^{(\ell)}\|_2\Big).
\end{aligned}
$$
(150)

By combining the result above, we have

$$\Delta h_{\max}^{(\ell)} \le \beta B_x\big((1-\alpha)\ell + \alpha\sqrt{d}\big)(\max\{1, \alpha\sqrt{d}B_w^\beta\})^{\ell-1}\Big(\|\Delta\mathbf{W}^{(1)}\|_2 + \ldots + \|\Delta\mathbf{W}^{(\ell)}\|_2\Big). \quad (151)$$

### I.3 Proof of Lemma 24

The proof is similar to the proof of Lemma 23.

### I.4 Proof of Lemma 25

Let $B_w^\beta = \beta B_w + (1-\beta)$. By the definition of $d_{\max}^{(\ell)}$, we have

$$\begin{aligned}
d_{\max}^{(\ell)} &= \max_i \|[\mathbf{D}^{(\ell)}]_{i,:}\|_2 \\
&= \alpha \max_i \|[\mathbf{L}^\top \sigma'(\mathbf{Z}^{(\ell)}) \odot \mathbf{D}^{(\ell+1)}\big(\beta\mathbf{W}^{(\ell)} + (1-\beta)\mathbf{I}\big)]_{i,:}\|_2 \\
&\le \alpha \max_i \|[\mathbf{L}^\top \sigma'(\mathbf{Z}^{(\ell)}) \odot \mathbf{D}^{(\ell+1)}]_{i,:}\|_2 \cdot B_w^\beta \\
&\underset{(a)}{\le} \alpha\sqrt{d}B_w^\beta \cdot d_{\max}^{(\ell+1)} \\
&\le \frac{2}{\gamma}(\max\{1, \alpha\sqrt{d}B_w^\beta\})^{L-\ell+1},
\end{aligned} \quad (152)$$

where inequality $(a)$ is due to the gradient of ReLU activation is element-wise either $0$ and $1$.

By the definition of $\Delta d_{\max}^{(\ell)}$, we have

$$\begin{aligned}
\Delta d_{\max}^{(\ell)} &= \max_i \|[\mathbf{D}^{(\ell)} - \tilde{\mathbf{D}}^{(\ell)}]_{i,:}\|_2 \\
&\le \max_i \alpha\|[\mathbf{L}^\top \sigma'(\mathbf{Z}^{(\ell)}) \odot \mathbf{D}^{(\ell+1)}\big(\beta\mathbf{W}^{(\ell)} + (1-\beta)\mathbf{I}\big) - \mathbf{L}^\top \sigma'(\tilde{\mathbf{Z}}^{(\ell)}) \odot \tilde{\mathbf{D}}^{(\ell+1)}\big(\beta\tilde{\mathbf{W}}^{(\ell)} - (1-\beta)\mathbf{I}\big)]_{i,:}\|_2 \\
&\underset{(a)}{\le} \max_i \alpha\|[\mathbf{L}^\top \mathbf{D}^{(\ell+1)}\big(\beta\mathbf{W}^{(\ell)} + (1-\beta)\mathbf{I}\big) - \mathbf{L}^\top \tilde{\mathbf{D}}^{(\ell+1)}\big(\beta\mathbf{W}^{(\ell)} + (1-\beta)\mathbf{I}\big)]_{i,:}\|_2 \\
&\quad + \max_i \alpha\|[\mathbf{L}^\top \tilde{\mathbf{D}}^{(\ell+1)}\big(\beta\mathbf{W}^{(\ell)} + (1-\beta)\mathbf{I}\big)]_{i,:} - \mathbf{L}^\top \tilde{\mathbf{D}}^{(\ell+1)}\big(\beta\tilde{\mathbf{W}}^{(\ell)} - (1-\beta)\mathbf{I}\big)]_{i,:}\|_2 \\
&\quad + \alpha \max_i \|[\mathbf{L}^\top \tilde{\mathbf{D}}^{(\ell+1)}\big(\beta\tilde{\mathbf{W}}^{(\ell)} - (1-\beta)\mathbf{I}\big)]_{i,:}\|_2 \max_i \|[\mathbf{Z}^{(\ell)} - \tilde{\mathbf{Z}}^{(\ell)}]_{i,:}\|_2 \\
&\underset{(b)}{\le} \max_i \alpha\|[\mathbf{L}^\top (\mathbf{D}^{(\ell+1)} - \tilde{\mathbf{D}}^{(\ell+1)})\big(\beta\mathbf{W}^{(\ell)} + (1-\beta)\mathbf{I}\big)]_{i,:}\|_2 \\
&\quad + \max_i \alpha\|[\mathbf{L}^\top \tilde{\mathbf{D}}^{(\ell+1)}\big(\beta(\mathbf{W}^{(\ell)} - \tilde{\mathbf{W}}^{(\ell)})\big)]_{i,:}\|_2 \\
&\quad + \alpha \max_i \|[\mathbf{L}^\top \tilde{\mathbf{D}}^{(\ell+1)}\big(\beta\tilde{\mathbf{W}}^{(\ell)} - (1-\beta)\mathbf{I}\big)]_{i,:}\|_2 \max_i \|[\mathbf{Z}^{(\ell)} - \tilde{\mathbf{Z}}^{(\ell)}]_{i,:}\|_2 \\
&\le \alpha\sqrt{d}B_w^\beta \cdot \Delta d_{\max}^{(\ell+1)} + \alpha\beta\sqrt{d}d_{\max}^{(\ell+1)}\|\Delta\mathbf{W}^{(\ell)}\|_2 + \alpha\sqrt{d}B_w^\beta d_{\max}^{(\ell+1)}\Delta z_{\max}^{(\ell)} \\
&= \alpha\sqrt{d}B_w^\beta \cdot \Delta d_{\max}^{(\ell+1)} + \underbrace{\alpha d_{\max}^{(\ell+1)}\Big(\beta\sqrt{d}\|\Delta\mathbf{W}^{(\ell)}\|_2 + \sqrt{d}B_w^\beta\Delta z_{\max}^{(\ell)}\Big)}_{(A)},
\end{aligned}$$
$$(153)$$

where inequality $(a)$ and $(b)$ is due to the gradient of ReLU activation is element-wise either $0$ and $1$.

Let $C_1 = \max\{1, \alpha\sqrt{d}B_w^\beta\}$ and $B_{\ell,d}^{\alpha,\beta} = \max\big\{\beta\big((1-\alpha)L + \alpha\sqrt{d}\big), (1-\alpha)LB_w^\beta + 1\big\}$. Recall that $d_{\max}^{(\ell+1)} \le \frac{2}{\gamma}(\max\{1, \alpha\sqrt{d}B_w^\beta\})^{L-\ell}$ and the upper bound of $\Delta z_{\max}^{(\ell)}$ in Lemma 24, we have

$$d_{\max}^{(\ell+1)} \le \frac{2}{\gamma}C_1^{L-\ell}, \quad \Delta z_{\max}^{(\ell)} \le \beta B_x B_{\ell,d}^{\alpha,\beta} C_1^{\ell-1}\big(\|\Delta\mathbf{W}^{(1)}\|_2 + \ldots + \|\Delta\mathbf{W}^{(\ell)}\|_2\big). \quad (154)$$

Therefore, we have upper bound $(A)$ by

$$
\alpha\sqrt{d}d_{\max}^{(\ell+1)}\Big(\beta\|\Delta\mathbf{W}^{(\ell)}\|_2 + B_w^\beta\Delta z_{\max}^{(\ell)}\Big)
$$

$$
\leq \alpha\beta\cdot\frac{2}{\gamma}C_1^{L-\ell}\cdot\Big(\sqrt{d}\|\Delta\mathbf{W}^{(\ell)}\|_2 + B_x B_{\ell,d}^{\alpha,\beta}C_1^\ell\big(\|\Delta\mathbf{W}^{(1)}\|_2 + \ldots + \|\Delta\mathbf{W}^{(\ell)}\|_2\big)\Big)
$$

$$
\leq \alpha\beta\frac{2}{\gamma}C_1^L\big(\sqrt{d} + B_{\ell,d}^{\alpha,\beta}B_x\big)\|\Delta\boldsymbol{\theta}\|_2 \tag{155}
$$

$$
\leq \alpha\beta\frac{2}{\gamma}C_1^L C_2\|\Delta\boldsymbol{\theta}\|_2,
$$

where $C_2 = \sqrt{d} + B_{\ell,d}^{\alpha,\beta}B_x$ and $B_{\ell,d}^{\alpha,\beta} = \max\Big\{\beta\big((1-\alpha)L + \alpha\sqrt{d}\big), (1-\alpha)LB_w^\beta + 1\Big\}$. By plugging it back and by induction, we have

$$
\Delta d_{\max}^{(\ell)} \leq C_1\Delta d_{\max}^{(\ell+1)} + \alpha\beta\frac{2}{\gamma}C_1^L C_2\|\Delta\boldsymbol{\theta}\|_2
$$

$$
\leq C_1^2\Delta d_{\max}^{(\ell+2)} + (1+C_1)\cdot\alpha\beta\frac{2}{\gamma}C_1^L C_2\|\Delta\boldsymbol{\theta}\|_2
$$

$$
\leq C_1^{L-\ell+1}\Delta d_{\max}^{(L+1)} + (1+C_1+\ldots+C_1^{L-\ell+1})\cdot\alpha\beta\frac{2}{\gamma}C_1^L C_2\|\Delta\boldsymbol{\theta}\|_2 \tag{156}
$$

$$
\leq C_1^{L-\ell}\Big(\Delta d_{\max}^{(L+1)} + \alpha\beta\frac{2}{\gamma}LC_1^L C_2\|\Delta\boldsymbol{\theta}\|_2\Big).
$$

Let first explicit upper bound $\Delta d_{\max}^{(L+1)}$. By definition, $\Delta d_{\max}^{(L+1)}$ as

$$
\Delta d_{\max}^{(L+1)} = \max_i\|\mathbf{d}_i^{(L+1)} - \tilde{\mathbf{d}}_i^{(L+1)}\|_2
$$

$$
= \max_i\left\|\frac{\partial\text{Loss}(f(\mathbf{h}_i^{(L)}), y_i)}{\partial\mathbf{h}_i^{(L)}} - \frac{\partial\text{Loss}(\tilde{f}(\tilde{\mathbf{h}}_i^{(L)}), y_i)}{\partial\tilde{\mathbf{h}}_i^{(L)}}\right\|_2
$$

$$
\leq \frac{2}{\gamma}\max_i\left\|\frac{\partial f(\mathbf{h}_i^{(L)})}{\partial\mathbf{h}_i^{(L)}} - \frac{\partial\tilde{f}(\tilde{\mathbf{h}}_i^{(L)})}{\partial\tilde{\mathbf{h}}_i^{(L)}}\right\|_2
$$

$$
\leq \frac{2}{\gamma}\Big(\Delta h_{\max}^{(L)} + (h_{\max}^{(L)} + 1)\Delta\boldsymbol{v}\Big)
$$

$$
= \frac{2}{\gamma}\Big[B_x B_{\ell,d}^{\alpha,\beta}C_1^{L-1}\big(\|\Delta\mathbf{W}^{(1)}\|_2 + \ldots + \|\Delta\mathbf{W}^{(L)}\|_2\big) + \big(B_x B_{\ell,d}^{\alpha,\beta}C_1^L + 1\big)\|\Delta\boldsymbol{v}\|_2\Big]
$$

$$
\leq \frac{2}{\gamma}\Big(B_\ell^\alpha B_x C_1^L + 1\Big)\|\Delta\boldsymbol{\theta}\|_2
$$

$$
= \frac{2}{\gamma}\Big(B_\ell^\alpha B_x C_1^L + 1\Big)\|\Delta\boldsymbol{\theta}\|_2, \tag{157}
$$

where $C_1 = \max\{1, \alpha\sqrt{d}B_w^\beta\}$, $B_w^\beta = \beta B_w + (1-\beta)$ and $B_{\ell,d}^{\alpha,\beta} = \max\Big\{\beta\big((1-\alpha)L + \alpha\sqrt{d}\big), (1-\alpha)LB_w^\beta + 1\Big\}$. By plugging it back, we have

$$
\Delta d_{\max}^{(\ell)} \leq C_1^{L-\ell}\Big(\frac{2}{\gamma}\big(B_{\ell,d}^{\alpha,\beta}B_x C_1^L + 1\big)\|\Delta\boldsymbol{\theta}\|_2 + \alpha\beta\frac{2}{\gamma}LC_1^L(\sqrt{d} + B_{\ell,d}^{\alpha,\beta}B_x)\|\Delta\boldsymbol{\theta}\|_2\Big)
$$

$$
\leq C_1^{L-\ell}\frac{2}{\gamma}\Big(B_{\ell,d}^{\alpha,\beta}B_x C_1^L + 1 + \alpha\beta LC_1^L(\sqrt{d} + B_{\ell,d}^{\alpha,\beta}B_x)\Big)\|\Delta\boldsymbol{\theta}\|_2 \tag{158}
$$

$$
\leq C_1^{L-\ell}\frac{2}{\gamma}\Big((\alpha\beta L + 1)(\sqrt{d} + B_{\ell,d}^{\alpha,\beta}B_x)C_1^L + 1\Big)\|\Delta\boldsymbol{\theta}\|_2
$$

$$
= C_1^{L-\ell}\frac{2}{\gamma}\Big((\alpha\beta L + 1)C_1^L C_2 + 1\Big)\|\Delta\boldsymbol{\theta}\|_2.
$$

## I.5  Proof of Lemma 26

By the definition of $\mathbf{G}^{(\ell)}$, we have

$$
\begin{aligned}
\|\mathbf{G}^{(\ell)}\|_2 &= \frac{1}{m}\left\|\beta\cdot[\alpha\mathbf{L}\mathbf{H}^{(\ell-1)}+(1-\alpha)\mathbf{X}]^\top\mathbf{D}^{(\ell)}\odot\sigma'(\mathbf{Z}^{(\ell)})\right\|_2 \\
&\leq \frac{1}{m}\beta\left\|[\alpha\mathbf{L}\mathbf{H}^{(\ell-1)}+(1-\alpha)\mathbf{X}]^\top\mathbf{D}^{(\ell)}\right\|_2 \\
&\leq \beta\Big(\alpha\sqrt{d}h_{\max}^{(\ell-1)}+(1-\alpha)B_x\Big)d_{\max}^{(\ell)}.
\end{aligned}
\tag{159}
$$

Plugging in the results from Lemma 22 and Lemma 25, we have

$$
\begin{aligned}
\|\mathbf{G}^{(\ell)}\|_2 &\leq \frac{2}{\gamma}\beta B_x\Big(\alpha\sqrt{d}\big((1-\alpha)(\ell-1)B_w^\beta+1\big)\cdot C_1^{\ell-1}+(1-\alpha)\Big)C_1^{L-\ell+1} \\
&\leq \frac{2}{\gamma}\beta B_x\big((1-\alpha)\ell+\alpha\sqrt{d}\big)C_1^L \leq \frac{2}{\gamma}\beta C_1^L C_2,
\end{aligned}
\tag{160}
$$

where $C_2 = \sqrt{d}+B_{\ell,d}^{\alpha,\beta}B_x$. Similarly, by the definition of $\mathbf{G}^{(L+1)}$, we have

$$
\|\mathbf{G}^{(L+1)}\|_2 \leq \frac{2}{\gamma}h_{\max}^{(L)} \leq \frac{2}{\gamma}C_1^L C_2.
\tag{161}
$$

Therefore, we have

$$
\sum_{\ell=1}^{L+1}\|\mathbf{G}^{(\ell)}\|_2 \leq \frac{2}{\gamma}(L+1)\beta C_1^L C_2.
\tag{162}
$$

Furthermore, we can upper bound the difference between gradient for $\ell\in[L]$ as

$$
\begin{aligned}
\|\mathbf{G}^{(\ell)}-\tilde{\mathbf{G}}^{(\ell)}\|_2 &= \frac{1}{m}\left\|\beta[\alpha\mathbf{L}\mathbf{H}^{(\ell-1)}+(1-\alpha)\mathbf{X}]^\top\mathbf{D}^{(\ell+1)}\odot\sigma'(\mathbf{Z}^{(\ell)})-\beta[\alpha\mathbf{L}\tilde{\mathbf{H}}^{(\ell-1)}+(1-\alpha)\mathbf{X}]^\top\tilde{\mathbf{D}}^{(\ell+1)}\odot\sigma'(\tilde{\mathbf{Z}}^{(\ell)})\right\|_2 \\
&\underset{(a)}{\leq} \frac{1}{m}\beta\|[\alpha\mathbf{L}\mathbf{H}^{(\ell-1)}+(1-\alpha)\mathbf{X}]^\top\mathbf{D}^{(\ell+1)}-[\alpha\mathbf{L}\tilde{\mathbf{H}}^{(\ell-1)}+(1-\alpha)\mathbf{X}]^\top\mathbf{D}^{(\ell+1)}\|_2 \\
&\quad + \frac{1}{m}\beta\|[\alpha\mathbf{L}\tilde{\mathbf{H}}^{(\ell-1)}+(1-\alpha)\mathbf{X}]^\top\mathbf{D}^{(\ell+1)}-[\alpha\mathbf{L}\tilde{\mathbf{H}}^{(\ell-1)}+(1-\alpha)\mathbf{X}]^\top\tilde{\mathbf{D}}^{(\ell+1)}\|_2 \\
&\quad + \frac{1}{m}\beta\|[\alpha\mathbf{L}\tilde{\mathbf{H}}^{(\ell-1)}+(1-\alpha)\mathbf{X}]^\top\tilde{\mathbf{D}}^{(\ell+1)}\odot\big(\sigma'(\mathbf{Z}^{(\ell)})-\sigma'(\tilde{\mathbf{Z}}^{(\ell)})\big)\|_2 \\
&\underset{(b)}{\leq} \beta\Big(\alpha\max_i\|[[\mathbf{L}(\mathbf{H}^{(\ell-1)}-\tilde{\mathbf{H}}^{(\ell-1)})]^\top\mathbf{D}^{(\ell+1)}]_{i,:}\|_2 \\
&\quad + \max_i\|[[\alpha\mathbf{L}\tilde{\mathbf{H}}^{(\ell-1)}+(1-\alpha)\mathbf{X}]^\top(\mathbf{D}^{(\ell+1)}-\tilde{\mathbf{D}}^{(\ell+1)})]_{i,:}\|_2+ \\
&\quad + \max_i\|[[\alpha\mathbf{L}\tilde{\mathbf{H}}^{(\ell-1)}+(1-\alpha)\mathbf{X}]^\top\tilde{\mathbf{D}}^{(\ell+1)}]_{i,:}\|_2\max_i\|[\mathbf{Z}^{(\ell)}-\tilde{\mathbf{Z}}^{(\ell)}]_{i,:}\|_2\Big) \\
&\leq \beta\Big(\underbrace{\alpha\sqrt{d}\Delta h_{\max}^{(\ell-1)}d_{\max}^{(\ell+1)}}_{(A)}+\underbrace{(\alpha\sqrt{d}h_{\max}^{(\ell-1)}+(1-\alpha)B_x)\Delta d_{\max}^{(\ell+1)}}_{(B)} \\
&\quad + \underbrace{(\alpha\sqrt{d}h_{\max}^{(\ell-1)}+(1-\alpha)B_x)d_{\max}^{(\ell+1)}\Delta z_{\max}^{(\ell)}}_{(C)}\Big),
\end{aligned}
\tag{163}
$$

where inequality $(a)$ and $(b)$ is due to the gradient of ReLU activation is element-wise either $0$ or $1$. We can bound $(A)$ as

$$
\alpha\sqrt{d}\Delta h_{\max}^{(\ell-1)}d_{\max}^{(\ell+1)} \leq \alpha\beta\sqrt{d}\frac{2}{\gamma}C_1^L C_2\|\Delta\boldsymbol{\theta}\|_2.
\tag{164}
$$

To upper bound $(B)$ and $(C)$, let first consider the upper bound of the following term

$$
\begin{aligned}
\alpha\sqrt{d}h_{\max}^{(\ell-1)}+(1-\alpha)B_x &\leq \alpha\sqrt{d}\Big((1-\alpha)(\ell-1)B_w^\beta+1\Big)B_x C_1^{\ell-1}+(1-\alpha)B_x \\
&\leq \alpha\sqrt{d}\Big((1-\alpha)\ell B_w^\beta+1\Big)B_x C_1^{\ell-1} \\
&\leq \alpha\sqrt{d}C_1^{\ell-1}C_2.
\end{aligned}
\tag{165}
$$

Therefore, we can bound $(B)$ as

$$\left(\alpha\sqrt{d}h_{\max}^{(\ell-1)} + (1-\alpha)B_x\right)\Delta d_{\max}^{(\ell+1)} \le \alpha\sqrt{d}C_1^{\ell-1}C_2 \cdot C_1^{L-\ell}\frac{2}{\gamma}\Big((\alpha\beta L+1)C_1^L C_2+1\Big)\|\Delta\boldsymbol{\theta}\|_2$$
$$\le \alpha\sqrt{d}\frac{2}{\gamma}C_1^L C_2\Big((\alpha\beta L+1)C_1^L C_2+1\Big)\|\Delta\boldsymbol{\theta}\|_2, \tag{166}$$

and we can bound $(C)$ by

$$\left(\alpha\sqrt{d}h_{\max}^{(\ell-1)} + (1-\alpha)B_x\right)d_{\max}^{(\ell+1)}\Delta z_{\max}^{(\ell)} \le \alpha\sqrt{d}C_1^{\ell-1}C_2 \cdot \frac{2}{\gamma}C_1^{L-\ell} \cdot \beta C_1^{\ell-1}C_2\|\Delta\boldsymbol{\theta}\|_2$$
$$\le \alpha\beta\sqrt{d}\frac{2}{\gamma}(C_1^L C_2)^2\|\Delta\boldsymbol{\theta}\|_2. \tag{167}$$

By combining result above, we have

$$\|\mathbf{G}^{(\ell)} - \tilde{\mathbf{G}}^{(\ell)}\|_2 \le \alpha\beta\frac{2}{\gamma}\sqrt{d}C_1^L C_2\Big(2\beta + (\alpha\beta L+1)C_1^L C_2\Big). \tag{168}$$

Similarly, we can upper bound the difference between gradient for the weight of the binary classifier as

$$\|\mathbf{G}^{(L+1)} - \tilde{\mathbf{G}}^{(L+1)}\|_2 \le \frac{2}{\gamma}\Delta h_{\max}^{(L)} \le \beta\frac{2}{\gamma}C_1^L C_2\|\Delta\boldsymbol{\theta}\|_2. \tag{169}$$

which is upper bound by the right hand side of the previous equation. Therefore, we have

$$\sum_{\ell=1}^{L+1}\|\mathbf{G}^{(\ell)} - \tilde{\mathbf{G}}^{(\ell)}\|_2 \le \alpha\beta\frac{2}{\gamma}(L+1)\sqrt{d}C_1^L C_2\Big(2\beta + (\alpha\beta L+1)C_1^L C_2\Big). \tag{170}$$

## J   Generalization bound for DGCN

In the following, we provide proof of the generalization bound of DGCN in Theorem 5. Recall that the update rule of DGCN is defined as

$$\mathbf{Z} = \sum_{\ell=1}^{L}\alpha_\ell\mathbf{H}^{(\ell)}, \ \mathbf{H}^{(\ell)} = \mathbf{L}^\ell\mathbf{X}(\beta_\ell\mathbf{W}^{(\ell)} + (1-\beta_\ell)\mathbf{I}). \tag{171}$$

The training of DGCN is an empirical risk minimization with respect to a set of parameters $\boldsymbol{\theta} = \{\mathbf{W}^{(1)}, \dots, \mathbf{W}^{(L)}, \mathbf{v}\}$, i.e.,

$$\mathcal{L}(\boldsymbol{\theta}) = \frac{1}{m}\sum_{i=1}^{m}\Phi_\gamma(-p(f(\mathbf{z}_i), y_i)), \ f(\mathbf{z}_i) = \tilde{\sigma}(\mathbf{v}^\top\mathbf{z}_i), \tag{172}$$

where $\mathbf{h}_i^{(L)}$ is the node representation of the $i$th node at the final layer, $f(\mathbf{h}_i^{(L)})$ is the predicted label for the $i$th node, $\tilde{\sigma}(x) = \frac{1}{\exp(-x)+1}$ is the sigmoid function, and loss function $\Phi_\gamma(-p(z,y))$ is $\frac{2}{\gamma}$-Lipschitz continuous with respect to its first input $z$. For simplification, let $\text{Loss}(z,y)$ denote $\Phi_\gamma(-p(z,y))$ in the proof.

For analysis purpose, we suppose $\alpha_\ell$ and $\beta_\ell$ are hyper-parameters that are pre-selected before training and fixed during the training phase. However, in practice, these two parameters are tuned during the training phase.

To establish the generalization of DGCN as stated in Theorem 3, we utilize the result on transductive uniform stability from [17] (Theorem 8 in Appendix F). Then, in Lemma 27, we derive the uniform stability constant for DGCN, i.e., $\epsilon_{\text{DGCN}}$.

**Lemma 27.** *The uniform stability constant for DGCN is computed as* $\epsilon_{DGCN} = \frac{2\eta\rho_f G_f}{m}\sum_{t=1}^{T}(1+\eta L_F)^{t-1}$ *where*

$$\rho_f = (\sqrt{d})^L B_x, \ G_f = \frac{2}{\gamma}(L+1)(\sqrt{d})^L B_x,$$
$$L_f = \frac{2}{\gamma}(L+1)(\sqrt{d})^L B_x\Big((\sqrt{d})^L B_x \max\{1, B_w\} + 1\Big). \tag{173}$$

By plugging the result in Lemma 27 back to Theorem 8, we establish the generalization bound for DGCN.

The key idea of the proof is to decompose the change of the network output into two terms (in Lemma 5) which depend on

- (Lemma 28) The maximum change of node representation $\Delta z_{\max}^{(\ell)} = \max_i \|[\mathbf{Z} - \tilde{\mathbf{Z}}]_{i,:}\|_2$,

- (Lemma 29) The maximum node representation $z_{\max}^{(\ell)} = \max_i \|[\mathbf{Z}]_{i,:}\|_2$.

**Lemma 28** (Upper bound of $z_{\max}$ for **DGCN**). *Let suppose Assumption 1 hold. Then, the maximum node embeddings for any node at the $\ell$th layer is bounded by*

$$z_{\max} \leq (\sqrt{d})^L B_x \max\{1, B_w\}. \tag{174}$$

**Lemma 29** (Upper bound of $\Delta z_{\max}$ for **DGCN**). *Let suppose Assumption 1 hold. Then, the maximum change between the node embeddings on two different set of weight parameters for any node at the $\ell$th layer is bounded by*

$$\Delta z_{\max} \leq (\sqrt{d})^L B_x(\|\Delta \mathbf{W}^{(1)}\|_2 + \ldots + \|\Delta \mathbf{W}^{(L)}\|_2), \tag{175}$$

*where $\Delta \mathbf{W}^{(\ell)} = \mathbf{W}^{(\ell)} - \tilde{\mathbf{W}}^{(\ell)}$.*

Then, in Lemma 30, we decompose the change of the model parameters into two terms which depend on

- The change of gradient with respect to the $\ell$th layer weight matrix $\|\Delta \mathbf{G}^{(\ell)}\|_2$,

- The gradient with respect to the $\ell$th layer weight matrix $\|\mathbf{G}^{(\ell)}\|_2$,

where $\|\Delta \mathbf{G}^{(\ell)}\|_2$ reflect the smoothness of APPNP model and $\|\mathbf{G}^{(\ell)}\|_2$ correspond the upper bound of gradient.

**Lemma 30** (Upper bound of $\|\mathbf{G}^{(\ell)}\|_2, \|\Delta \mathbf{G}^{(\ell)}\|_2$ for **DGCN**). *Let suppose Assumption 1 hold. Then, the gradient and the maximum change between gradients on two different set of weight parameters are bounded by*

$$
\begin{aligned}
\sum_{\ell=1}^{L+1} \|\mathbf{G}^{(\ell)}\|_2 &\leq \frac{2}{\gamma}(L+1)(\sqrt{d})^L B_x, \\
\sum_{\ell=1}^{L+1} \|\Delta \mathbf{G}^{(\ell)}\|_2 &\leq \frac{2}{\gamma}(L+1)(\sqrt{d})^L B_x\Big((\sqrt{d})^L B_x \max\{1, B_w\} + 1\Big)\|\Delta \boldsymbol{\theta}\|_2,
\end{aligned}
\tag{176}
$$

*where $\|\Delta \boldsymbol{\theta}\|_2 = \|\mathbf{v} - \tilde{\mathbf{v}}\|_2 + \sum_{\ell-1}^{L} \|\mathbf{W}^{(\ell)} - \tilde{\mathbf{W}}^{(\ell)}\|_2$ denotes the change of two set of parameters.*

Equipped with above intermediate results, we now proceed to prove Lemma 27.

*Proof.* Recall that our goal is to explore the impact of different GCN structures on the uniform stability constant $\epsilon_{\text{DGCN}}$, which is a function of $\rho_f$, $G_f$, and $L_f$. By Lemma 29, we know that the function $f$ is $\rho_f$-Lipschitz continuous, with $\rho_f = (\sqrt{d})^L B_x$. By Lemma 30, we know the function $f$ is $L_f$-smoothness, and the gradient of each weight matrices is bounded by $G_f$, with

$$G_f \leq \frac{2}{\gamma}(L+1)(\sqrt{d})^L B_x, \ L_f \leq \frac{2}{\gamma}(L+1)(\sqrt{d})^L B_x\Big((\sqrt{d})^L B_x \max\{1, B_w\} + 1\Big). \tag{177}$$

By plugging $\epsilon_{\text{DGCN}}$ into Theorem 3, we obtain the generalization bound of DGCN.

$\square$

### J.1 Proof of Lemma 28

By the definition of $z_{\max}$, we have

$$
\begin{aligned}
z_{\max} &= \max_i \left\| \left[ \sum_{\ell=1}^{L} \alpha_\ell \mathbf{L}^\ell \mathbf{X} \big( \beta_\ell \mathbf{W}^{(\ell)} + (1-\beta_\ell)\mathbf{I} \big) \right]_{i,:} \right\|_2 \\
&\leq \sum_{\ell=1}^{L} \alpha_\ell \max_i \| [\mathbf{L}^\ell \mathbf{X} \big( \beta_\ell \mathbf{W}^{(\ell)} + (1-\beta_\ell)\mathbf{I} \big)]_{i,:} \|_2 \\
&\overset{(a)}{\leq} \sum_{\ell=1}^{L} \alpha_\ell (\sqrt{d})^\ell B_x \big( \beta_\ell B_w + (1-\beta_\ell) \big) \\
&\overset{(b)}{\leq} (\sqrt{d})^L B_x \max\{1, B_w\},
\end{aligned}
\tag{178}
$$

where inequality $(a)$ follows from Lemma 31 and inequality $(b)$ is due to the fact that $\sum_{\ell=1}^{L} \alpha_\ell = 1$ and $\alpha_\ell \in [0,1]$.

### J.2 Proof of Lemma 29

By the definition of $\Delta z_{\max}$, we have

$$
\begin{aligned}
\Delta z_{\max} &= \max_i \| [\mathbf{Z} - \tilde{\mathbf{Z}}]_{i,:} \|_2 \\
&= \max_i \left\| \left[ \sum_{\ell=1}^{L} \alpha_\ell \mathbf{L}^\ell \mathbf{X} \big( \beta_\ell \mathbf{W}^{(\ell)} + (1-\beta_\ell)\mathbf{I} \big) - \sum_{\ell=1}^{L} \alpha_\ell \mathbf{L}^\ell \mathbf{X} \big( \beta_\ell \tilde{\mathbf{W}}^{(\ell)} + (1-\beta_\ell)\mathbf{I} \big) \right]_{i,:} \right\|_2 \\
&\leq \sum_{\ell=1}^{L} \alpha_\ell \max_i \left\| \left[ \mathbf{L}^\ell \mathbf{X} \big( \beta_\ell \mathbf{W}^{(\ell)} + (1-\beta_\ell)\mathbf{I} \big) - \mathbf{L}^\ell \mathbf{X} \big( \beta_\ell \tilde{\mathbf{W}}^{(\ell)} + (1-\beta_\ell)\mathbf{I} \big) \right]_{i,:} \right\|_2 \\
&= \sum_{\ell=1}^{L} \alpha_\ell \beta_\ell \max_i \left\| \left[ \mathbf{L}^\ell \mathbf{X} (\mathbf{W}^{(\ell)} - \tilde{\mathbf{W}}^{(\ell)}) \right]_{i,:} \right\|_2 \\
&\overset{(a)}{\leq} \sum_{\ell=1}^{L} \alpha_\ell \beta_\ell (\sqrt{d})^\ell B_x \| \Delta \mathbf{W}^{(\ell)} \|_2 \\
&\leq (\sqrt{d})^L B_x \Big( \| \Delta \mathbf{W}^{(1)} \|_2 + \ldots + \| \Delta \mathbf{W}^{(L)} \|_2 \Big),
\end{aligned}
\tag{179}
$$

where inequality $(a)$ follows from Lemma 31 and inequality $(b)$ is due to the fact that $\sum_{\ell=1}^{L} \alpha_\ell = 1$ and $\alpha_\ell \in [0,1]$.

## J.3 Proof of Lemma 30

Recall that $\mathbf{G}^{(\ell)}$ is defined as the gradient of loss with respect to $\mathbf{W}^{(\ell)}$, which can be formulated as

$$
\begin{aligned}
\|\mathbf{G}^{(\ell)}\|_2 &= \left\| \frac{1}{m} \sum_{i=1}^{m} \frac{\partial \mathrm{Loss}(f(\mathbf{z}_i), y_i)}{\partial \mathbf{z}_i} \frac{\partial \mathbf{z}_i}{\partial \mathbf{h}_i^{(\ell)}} \frac{\partial \mathbf{h}_i^{(\ell)}}{\partial \mathbf{W}^{(\ell)}} \right\|_2 \\
&\leq \max_i \left\| \frac{\partial \mathrm{Loss}(f(\mathbf{z}_i), y_i)}{\partial \mathbf{z}_i} \frac{\partial \mathbf{z}_i}{\partial \mathbf{h}_i^{(\ell)}} \frac{\partial \mathbf{h}_i^{(\ell)}}{\partial \mathbf{W}^{(\ell)}} \right\|_2 \\
&\leq \alpha_\ell \beta_\ell (\sqrt{d})^\ell B_x \max_i \left\| \frac{\partial \mathrm{Loss}(f(\mathbf{z}_i), y_i)}{\partial \mathbf{z}_i} \right\|_2 \\
&\underset{(a)}{\leq} \frac{2}{\gamma} \alpha_\ell \beta_\ell (\sqrt{d})^\ell B_x \\
&\leq \frac{2}{\gamma} (\sqrt{d})^L B_x,
\end{aligned}
\tag{180}
$$

where $(a)$ is due to the fact that loss function is $\frac{2}{\gamma}$-Lipschitz conitnous.

Besides, recall that $\mathbf{G}^{(L+1)}$ denotes the gradient with respect to the weight of binary classifier. Therefore, we have

$$
\|\mathbf{G}^{(L+1)}\|_2 \leq \frac{2}{\gamma} h_{\max}^{(L)} \underset{(a)}{\leq} \frac{2}{\gamma} (\sqrt{d})^L B_x,
\tag{181}
$$

where inequality $(a)$ follows from Lemma 29.

Therefore, combining the results above, we have

$$
\sum_{\ell=1}^{L+1} \|\mathbf{G}^{(\ell)}\|_2 \leq \frac{2}{\gamma} (L+1)(\sqrt{d})^L B_x.
\tag{182}
$$

Furthermore, by the definition of $\|\mathbf{G}^{(\ell)} - \tilde{\mathbf{G}}^{(\ell)}\|_2$, we have

$$
\begin{aligned}
\|\mathbf{G}^{(\ell)} - \tilde{\mathbf{G}}^{(\ell)}\|_2 &\leq \frac{2}{\gamma} \max_i \left\| \frac{\partial f(\mathbf{z}_i)}{\partial \mathbf{z}_i} \frac{\partial \mathbf{z}_i}{\partial \mathbf{W}} - \frac{\partial f(\tilde{\mathbf{z}}_i)}{\partial \tilde{\mathbf{z}}_i} \frac{\partial \tilde{\mathbf{z}}_i}{\partial \tilde{\mathbf{W}}} \right\|_2 \\
&= \frac{2}{\gamma} \alpha_\ell \beta_\ell (\sqrt{d})^\ell B_x \left\| \frac{\partial f(\mathbf{z}_i)}{\partial \mathbf{z}_i} - \frac{\partial f(\tilde{\mathbf{z}}_i)}{\partial \tilde{\mathbf{z}}_i} \right\|_2 \\
&\leq \frac{2}{\gamma} \alpha_\ell \beta_\ell B_x (\sqrt{d})^\ell \left( \Delta z_{\max} + (z_{\max}+1) \|\Delta \mathbf{v}\|_2 \right) \\
&\leq \frac{2}{\gamma} (\sqrt{d})^L B_x \left( (\sqrt{d})^L B_x \max\{1, B_w\} + 1 \right) \|\Delta \boldsymbol{\theta}\|_2.
\end{aligned}
\tag{183}
$$

Similarly, we can upper bound the difference between gradient for the weight of the binary classifier as

$$
\begin{aligned}
\|\mathbf{G}^{(L+1)} - \tilde{\mathbf{G}}^{(L+1)}\|_2 &\leq \frac{2}{\gamma} \Delta h_{\max}^{(L)} \\
&\leq \frac{2}{\gamma} (\sqrt{d})^L B_x B_w,
\end{aligned}
\tag{184}
$$

which is bounded by the right hand side of the last equation. Therefore, we have

$$
\sum_{\ell=1}^{L+1} \|\mathbf{G}^{(\ell)} - \tilde{\mathbf{G}}^{(\ell)}\|_2 \leq \frac{2}{\gamma} (L+1)(\sqrt{d})^L B_x \left( (\sqrt{d})^L B_x \max\{1, B_w\} + 1 \right) \|\Delta \boldsymbol{\theta}\|_2.
\tag{185}
$$

# K  Omitted proofs of useful lemmas

## K.1  Proof of Lemma 1

Let $f_k(\boldsymbol{\theta})$ denote applying function $f$ with parameter $\boldsymbol{\theta}$ on the $k$th data point and $\boldsymbol{\theta}_t$ as the weight parameter at the $t$th iteration. Because function $f_k(\boldsymbol{\theta})$ is $\rho_f$-Lipschitz with respect to parameter $\boldsymbol{\theta}$, we

can bound the difference between model output by the difference between parameters $\boldsymbol{\theta}_T^{ij}, \boldsymbol{\theta}_T$, i.e.,

$$\epsilon = \max_k |f_k(\boldsymbol{\theta}_T^{ij}) - f_k(\boldsymbol{\theta}_T)| \leq \rho_f \|\boldsymbol{\theta}_T^{ij} - \boldsymbol{\theta}_T\|_2. \tag{186}$$

Let $\nabla f(\boldsymbol{\theta}) = \frac{1}{m} \sum_{k=1}^m \nabla f_k(\boldsymbol{\theta})$ and $\nabla f(\boldsymbol{\theta}^{ij}) = \frac{1}{m} \sum_{k=1}^m \nabla f_k(\boldsymbol{\theta}^{ij})$ denote the full-batch gradient computed on the original and perturbed dataset respectively.

Recall that both models have the same initialization $\boldsymbol{\theta}_0 = \boldsymbol{\theta}_0^{ij}$. At each iteration the full-batch gradient are computed on $(m-1)$ data points that identical in two dataset, and only 1 data point that are different in two dataset. Therefore, we can bound the change of model parameters $\boldsymbol{\theta}_t^{ij}, \boldsymbol{\theta}_t$ after one gradient update step as

$$\begin{aligned} \|\boldsymbol{\theta}_{t+1}^{ij} - \boldsymbol{\theta}_{t+1}\|_2 &\leq \left(1 + (1 - \frac{1}{m})\eta L_f\right) \|\boldsymbol{\theta}_t^{ij} - \boldsymbol{\theta}_t\|_2 + \frac{2\eta G_f}{m} \\ &\leq (1 + \eta L_f)\|\boldsymbol{\theta}_t^{ij} - \boldsymbol{\theta}_t\|_2 + \frac{2\eta G_f}{m}. \end{aligned} \tag{187}$$

Then after $T$ iterations, we can bound the different between two parameters as

$$\|\boldsymbol{\theta}_T^{ij} - \boldsymbol{\theta}_T\|_2 \leq \frac{2\eta G_f}{m} \sum_{t=1}^T (1 + \eta L_F)^{t-1}. \tag{188}$$

By plugging the result back to Eq. 186, we have

$$\epsilon = \frac{2\eta \rho_f G_f}{m} \sum_{t=1}^T (1 + \eta L_F)^{t-1}. \tag{189}$$

## K.2 Upper bound on the $\ell_1$-norm of Laplacian matrix

**Lemma 31** (Lemma A.3 in [36])**.** *Let $\mathbf{A}$ denote the adjacency matrix of a self-connected undirected graph $\mathcal{G}(\mathcal{V}, \mathcal{E})$, and $\mathbf{D}$ denote its corresponding degree matrix, and $d$ denote the maximum number of node degree. We define the graph Laplacian matrix as $\mathbf{L} = \mathbf{D}^{-1/2} \mathbf{A} \mathbf{D}^{-1/2}$. Then we have*

$$\max_i \|[\mathbf{L}]_{i,:}\|_2 \leq \sqrt{d}. \tag{190}$$

*Proof.* By the definition of Laplacian matrix, we know that the $i$th row and $j$th column of $\mathbf{L}$ is defined as $L_{i,j} = \frac{1}{\sqrt{\deg(i)}\sqrt{\deg(j)}}$. Therefore, we have

$$\begin{aligned} \max_i \sum_{j=1}^N L_{i,j} &= \max_i \sum_{j \in \mathcal{N}(i)} \frac{1}{\sqrt{\deg(i)}\sqrt{\deg(j)}} \\ &\overset{\leq}{_{(a)}} \max_i \sum_{j \in \mathcal{N}(i)} \frac{1}{\sqrt{\deg(i)}} \\ &= \max_i \sqrt{\deg(i)} \leq \sqrt{d}, \end{aligned} \tag{191}$$

where $(a)$ is due to $\frac{1}{\sqrt{\deg(i)}} \leq 1$ for any node $i$.

$\square$