# OpenReview forum: "On Provable Benefits of Depth in Training Graph Convolutional Networks"
_NeurIPS.cc/2021/Conference — NeurIPS 2021 Poster_

### Official Review · Reviewer_aKcs · 2021-07-14

**Rating:** 7
**Confidence:** 4

**Summary:**

The authors argue that over-smoothing does not necessarily happen in practice and prove that a deep model is expressive as long as properly trained and can converge to global optimal with linear convergence rate. They also analyze the generalization capability of GCNs  and propose a decoupled structure for GCNs to preserve the expressive power and ensure  good generalization performance.

**Limitations And Societal Impact:**

Yes, the authors adequately addressed the limitations and potential negative societal impact .

**Main Review:**

Strength:
1. Understanding over-smoothing problem is important for GNNs and it’s still there although there are many deep GNN models can empirically get good results.
2. I personally very like some points in this paper, e.g. “…over-smoothing is mainly an artifact of theoretical analysis and simplifications made in analysis…”; “Existing results expressive power are mixed, [38, 36, 7, 5] argue that deeper
 model has higher expressive power but [40, 26, 25] has the completely contradict result”
3. The authors provide detailed theoretical analysis of over-smoothing and it’s important for the whole ML community.


Weakness&Advice:

1. In theorem 2, why do you need $d_L \geq N$. This condition seems impractical.
2. Page 7 is the worst part of this paper because of writing problem. The writing is understandable but not reader friendly.
3. The font size in all figures need to be enlarged.
4. Lemma 1 can be moved to appendix and theorem 4 need to be elaborated. The content in page 7 is not reader friendly.
4. In Figure 4, some curves stops before iteration 200. Need more explanation. What is the results on test set.
5. Line 335-337, “…DGCN assigns smaller weight to each layerwise function $f^l(X)$ with more propagation steps at the beginning of training. Throughout the training, DGCN gradually adjusts the weight to leverage more useful large receptive field information.” It’s better to provide a figure on how $\alpha_l$ changes during training to confirm this claim.

Some grammar problems:

Line 5, “…over-smoothing not necessarily  happen…” —> “…over-smoothing does not necessarily  happen…”.
Line 226, “a deeper GCNs has worse performance” —> “a deeper GCN has worse performance”. There are some misuses of ‘GCN’ and ‘GCNs’ in other places, please double check.

I will raise my score if the writing can be improved. Overall, I would like to see this paper to be published.


**Time Spent Reviewing:**

8

---

> ### Author Response · Authors · 2021-08-10
> **Response to Reviewer (aKcs)**
>
> *Q1: Why need $d_L > N$?*
>
> A1: Thank you for asking. To show training loss is exponentially decreasing during training, we have to show the model parameters are not changing significantly during training, which requires to lower bound $\lambda_{\min}\left(\mathbf{H}^{(L)} (\mathbf{H}^{(L)})^\top \right)$ mathematically. However, bounding $\lambda_{\min}\left(\mathbf{H}^{(L)} (\mathbf{H}^{(L)})^\top \right)$ is non-trivial. Fortunately, when $d_L > N$ we have the transpose of node representation $(\mathbf{H}^{(L)})^\top \in \mathbb{R}^{d_L \times N}$ as a tall-thin row matrix, such that the equality $\lambda_{\min}\left(\mathbf{H}^{(L)} (\mathbf{H}^{(L)})^\top \right) = \lambda_{\min}^2(\mathbf{H}^{(L)})$ holds, then we can lower bound $\lambda_{\min}\left(\mathbf{H}^{(L)} (\mathbf{H}^{(L)})^\top \right)$ using $\lambda_{\min}^2(\mathbf{H}^{(L)})$, which is a function of largest singular value of weight matrices and input feature matrix. Bridging the gap is an interesting question that we might work on in the future.
>
> *Q2 and Q3: Page 7 is understandable but not reader-friendly. Font size in figures.*
>
> A2: We appreciate your suggestions and we will do our best to improve the readability of the paper.
>
> *Q4: Move Lemma 1 to appendix and more elaborate on Theorem 4.*
>
> A4: Thank you for the suggestions and will definitely take these into considerations. However, we think we need Lemma 1 right before Theorem 4 since Lemma 1 shows what type of role each individual $\rho_f, L_f, G_T, T$ plays in our final results in Table 1 and Theorem 4. In the current version, many details are shortened in the main body due to space constraints, we will elaborate more and fix this issue in the updated draft.
>
> *Q5: (1) Why figure 4 stops at 200? (2) How about the test scores?*
>
> A5: (1) The curve stops early at the largest training accuracy iteration (as one can see from the second row figures they reached 100\% training f1-score). (2) Thank you for pointing this out. We will add the test scores in the figure. Since the nodes in the synthetic dataset are randomly splitted into training/validation/testing set, the testing results is expected to be very close to the best validation results.
>
> *Q6: Better provide figures on the evolving dynamic of weight $\alpha_\ell$.*
>
> A6: Thank you for the suggestions, we will add this in the appendix for the following draft.

---

> > ### Comment · Reviewer_aKcs · 2021-08-25
> > **I'll Keep My Rating**
> >
> > Based on the response from the authors and the comments from other reviewers, I'll keep my rating and recommend this paper to be accepted.

---

### Official Review · Reviewer_btGR · 2021-07-16

**Rating:** 7
**Confidence:** 4

**Summary:**

This paper analyzed GCN model for node classification in depth. The paper first analyzed the deep model's performance degeneration problem and argued that the deep model's expressiveness is not worse than shadow model so it does not suffer from oversmoothing problem. In fact, the degeneration of deep model comes from the hardness of training instead of oversmoothing, and the papers theoretically proved that deep model needs more iterations to converge, under a strong assumption. The above explanation explained the training loss but not characterize the final test performance. The author further explored the generalization error (the gap between training loss and test loss) and proved that under transductive setting the generalization error is affected by the number of training epochs, the number of layers, singluar value of weight matrix and largest node degrees, which explained why deep models have degenerated test performance. The author finally proposed a weight to decouple weight matrix and feature propagation operations, which keeps the expressiveness and also improves generalization ability.

**Main Review:**

1. Originality: the analysis of performance of deep models is comprehensive in terms of training and test performance, and is novel. The author gives enough empirical observations as well as solid theoretical analysis (although some parts are not clear enough, and I'm not sure whether the proof contains error or not). Overall the novelty is great.

2. Quality: in general I buy the story as all explanations align with my own experience and also greatly explains their observations. However I do have several questions. And feel some claims seem too aggressive without solid enough theoretical results.
 - Line 120 defines the normalized Laplacian matrix and DAD which is not correct, I-DAD is the normalized Laplacian matrix.
 - Theorem 1 charaterize node feature richness by using graph-level result, and this needs to view each node embedding as a graph embedding of the computational tree structure. I want to know whether the result rely on independent assumption of very computational subgraph (in definition 1). In general N computational subgraphs in original GNN are coupled together instead of independent, which is true for graph level tasks.
 - Theorem 2 needs dL >= N, which is not valid in real world, (however you can assume a wide enough representation with NxN matrix size in theory) , I would like to know how does this affect the result of theorem 2 if dL is a constant like 512.
 - Statement from line 308 to line 310 seems to bold and incorrect to me, as you can also pickup hyper-parameter to make the order incorrect. I hope the author can provide more solid analysis instead of this vague argument. This argument is intended to match empirical observation in my understanding.
- There are conflicted statement about resnet, the line 292-295 is very different from line 354-357. One argument said residual connection worse the generalization while the other said that resnet is very similar to DGCN. I hope the author provide more analysis to explain why adding residual worse the generalization ability.
 - The Remark 1 is great empirical observation, however the statement sounds too confirm to me. If you can give more theoertical analysis out of empirical observation the remark 1 will become solid to me.
 - The generalization error is showed only on synthetic dataset and cora. I would like to see the empirical analysis over OGB dataset.
 - Table 1 is hard to read as it involves so many equations, if you would provide some intuition of them on the title of the table it will be a lot easier to follow.

3. Clarity: in general the paper is well-written and easy to follow.

4. Significance: the DGCN performance is not solid enough. This is acceptable as all previous analysis is novel and comprehensive enough. Nevertheless, I do expect that DGCN can help better generalization ability as it is specially designed. The similar performance between DGCN and APPNP suggests the generalization gap is not fully charaterized by the theory presented in this paper (whether upper bound is tight or not).





**Time Spent Reviewing:**

5

---

> ### Author Response · Authors · 2021-08-10
> **Response to Reviewer (btGR)**
>
> *Q1: $\mathbf{D}^{-1/2} \mathbf{A} \mathbf{D}^{-1/2}$ is not named as a normalized Laplacian matrix.*
>
> A1: Thank you for pointing this out and sorry for the confusion. We are actually referring to the "graph propagation matrix" defined in [1], we will clarify this in the revision.
>
> *Q2: Whether the result relies on the independent assumption of every computational sub-graph in Theorem 1?*
>
> A2: We appreciate the reviewer for bringing up this question. To achieve the upper bound in Theorem 1, we need an implicit assumption on the disjointedness of the computation tree of two node (which is similar to the independent assumption as mentioned by the reviewer). However, this does not affect the correctness of the theorem. We will clarify this point in the revision.
>
> *Q3: Theorem 2 requires a network dimension $d_L$ larger than the number of data $N$, which is not valid in the real world. Why do we need $d_L>N$, can $d_L$ be a constant?*
>
> A3: We totally agree that this assumption does not align with practice, but as far as theoretical results are concerned it is necessary to have this assumption. The reason is summarized as follows. To show training loss is exponentially decreasing during training, we have to show the model parameters are not changing significantly during training, which requires to lower bound $\lambda_{\min}\left( \mathbf{H}^{(L)}(\mathbf{H}^{(L)})^\top \right)$ mathematically. However, bounding $\lambda_{\min}\left( \mathbf{H}^{(L)} (\mathbf{H}^{(L)})^\top \right)$ is non-trivial. Fortunately, when $d_L > N$ we have the transpose of node representation $(\mathbf{H}^{(L)})^\top \in \mathbb{R}^{d_L \times N}$ as a tall-thin row matrix, such that the equality $\lambda_{\min}\left( \mathbf{H}^{(L)} (\mathbf{H}^{(L)})^\top \right) = \lambda_{\min}^2(\mathbf{H}^{(L)})$ holds, then we can lower bound $\lambda_{\min}\left(\mathbf{H}^{(L)} (\mathbf{H}^{(L)})^\top \right)$ using $\lambda_{\min}^2(\mathbf{H}^{(L)})$, which is a function of largest singular value of weight matrices and input feature matrix. Bridging this gap is an interesting question that we might work on in the future.
>
> *Q4: Generalization result order in line 308-310.*
>
> A4: We completely agree with the reviewer about the order of generalization gap here, as one might be able to generate another order using very extreme hyper-parameters. However note that we are using the commonly used hyper-parameters from the original GCN paper [1] for our experiments, (i.e., model dimension $64$, learning rate $0.001$, no augmentations or other hyper-parameters). Also, we tried various model dimensions (i.e 32, 128 and 256) and achieve almost the results. We would like to make this point clear in the following draft and include a discussion and further result on this manner.
>
> *Q5: Conflict statements in line 292-295 and line 354-357.*
>
> A5: Thank you for pointing this out. Actually the two statements are not in conflict. ResNet can be seen as using $\alpha_\ell=1, \ell\in[L]$ for each layer model, which can be regarded as the "*summation* of the model complexity" of $L$-layer. However, DGCN is using $\sum_{\ell=1}^L \alpha_\ell = 1$, which can be thought of as a "*weighted average* of model complexity". With just a simple change on the ResNet structure, our model DGCN is both easy to train and good to generalize.
>
> *Q6: More empirical validations for Remark 1.*
>
> A6: Thank you for your suggestion. We provided some results in Appendix D3 and D4, and we would also like to provide more empirical results and discussion on the convergence and generalization analysis.
>
> *Q7: More generalization error results on OGB and more intuitive introduction on Table 1.*
>
> A7: Thank you for your suggestions, we will definitely consider your suggestion to improve the readability!
>
> [1] Kipf, Thomas N., and Max Welling. Semi-supervised classification with graph convolutional networks.

---

### Official Review · Reviewer_HPrn · 2021-07-17

**Rating:** 7
**Confidence:** 3

**Summary:**

The paper provides a novel theoretical understanding of over-smoothing and point out that the major impact of poor performance is due to the generation gap. The paper brings a new point about why deep GCN fails. Moreover, the paper designs a new framework according to their theorem although the proposed framework is just a dense version of GCNII.

**Limitations And Societal Impact:**

Yes

**Main Review:**

Pros:
1. The paper point out the issues that the previous papers may neglect some other reasons that cause poor performance besides over-smoothing. Many papers assumes that the singular values of weights <1 and the second largest eigenvalue <1. However, the paper finds that the singular values of weights is not always <1 and the second largest eigenvalue is very close to 1 in the rea-world datasets. Consequently, it may not cause the vanishing of the layer outputs along infinite layers.
2. The paper theoretically analyzes another fact of poor performance, the generation gap. Lemma 1 formulates the generation gap in terms of Lipschitz continuity, smoothness and gradient scale.

Cons:
1. In Figure1, the paper claims that GCN would reach a high training accuracy on Cora with efficient number of iterations. However, I tested GCN on Cora with lr=[0.01~$10^-10$], dropout=0.5, hidden=128, epochs=6000, layers=8 (Basic modules from dgl). I did not get neither converged training loss nor high training accuracy. So how does the authors train the GCN to get high training accuracy?
2.  The experimental results does not show the results with very deep GCNs such as 32 layers or 64 layers. And the paper did not compare with some other over-smoothing-coping methods such as dropedge, Pair norm, etc. Or I might miss them in the Appendix.
3. The main theoretical results are not related to the Laplacian matrix. It seems that the paper relax the Laplacian matrix to be no larger than 1 (The largest eigenvalue). It could be a tighter bound with the second largest eigenvalue. Moreover, it is little confused between layer number $L$ and Laplacian matrix $\mathbf{L}$.
4. In [26], it involves the analysis for GCN with bias. The paper does not discuss the situation with bias. It might be a good future work.
5. The fonts of figures are too small.

**Time Spent Reviewing:**

12

---

> ### Author Response · Authors · 2021-08-10
> **Response to Reviewer (HPrn)**
>
> *Q1: How to achieve very high training accuracy?*
>
> A1. Thank you for testing the correctness of the experiments. High training score can be achieved by removing both dropout and weight decay. These two augmentations are designed to improve the generalization ability (i.e., validation/testing accuracy) of the model but might hurt the training accuracy, because of the randomness and also an extra regularization term on the model parameters are introduced during training. Please refer to our test on DGL [`DGL_code.ipynb`](https://anonymous.4open.science/r/1bf5e29d-c71c-44f4-b55f-4bae8e266029/DGL_code.ipynb).
> Please also feel free to refer to the provided code, the link to the anonymous repository at the Line 606 of appendix. Besides, the reviewer can find a very clean implementation of GCN using PyTorch's sparse matrix multiplication in [`model.py`](https://anonymous.4open.science/r/1bf5e29d-c71c-44f4-b55f-4bae8e266029/model.py) and [`layers.py`](https://anonymous.4open.science/r/1bf5e29d-c71c-44f4-b55f-4bae8e266029/layers.py). The code to reproduce the training F1-score results can be found in [`example.sh`](https://anonymous.4open.science/r/1bf5e29d-c71c-44f4-b55f-4bae8e266029/example.sh) and [`GCN_train_f1_score.ipynb`](https://anonymous.4open.science/r/1bf5e29d-c71c-44f4-b55f-4bae8e266029/GCN_train_f1_score.ipynb).
>
> *Q2: (1) Compare with other over-smoothing implementations, e.g., PairNorm and DropEdge, and (2) test on very deep GCN, e.g., 32/64 layers.*
>
> A2: Thank you for your suggestions. (1) Due to space limit, we put the empirical results of DropEdge and PairNorm in the supplementary and kept the more relevant experiments in the main body to ensure the integrity and fluency of the text. Please refer to the Appendix E3, E4 on page 28 in the supplementary material. (2) Training very deep GCN is also affected by the gradient stability issue (as discussed on Lines 217-233 with empirical evidence in Appendix E.5). Unfortunately, we cannot  train a very deep GCN model due to the gradient stability issue. Please feel free to check our discussion in the main body and the referred supplementary experiment evidence.
>
> *Q3: (1) Tighter results with second largest eigenvalues. (2) Confusion due to notation layer $L$ and Laplacian $\mathbf{L}$.*
>
> A3: (1) We greatly appreciate the suggestions. Indeed, we can get a tighter bound by using second largest eigenvalues. However, since we are using a uniformed generalization analysis framework on all GCNs and their variants, our current relaxation of Laplacian is enough for comparison and we believe is more reader friendly due to its simplicity. (2) Sorry for the confusion. Here we use the most standard mathematics notations: unbold (both capital and non-capital) characters for scalars (e.g., $A \in \mathbb{R}$),  and the boldface of capital character stands for matrix (e.g., $\mathbf{A} \in \mathbb{R}^{m\times n}$), and boldface of non-capital character for vectors (e.g., $\mathbf{a} \in \mathbb{R}^{d}$).
> However, we would love to change notation Laplacian $\mathbf{L}$ to propagation matrix $\mathbf{P}$ in order to alleviate the potential confusion in the revised version.
>
> *Q4: GCN with bias. Font in the figure is too small.*
>
> A4: Thank you for your thoughtful feedback, we will definitely take your suggestions into considerations and also make sure all figures are readable in the following draft.

---

> > ### Comment · Reviewer_HPrn · 2021-08-12
> > **Good reproducing results for Deep GraphConv**
> >
> > The feedback from the authors addressed my questions. I missed the code linking in the appendix because I thought the codes should be submitted as the supplementary materials. The dropout and weight decay  are really the key reasons against convergence after re-checking my code. I will raise my ranking to 7.

---

### Official Review · Reviewer_Z7vj · 2021-07-18

**Rating:** 5
**Confidence:** 3

**Summary:**

The deals with semi-supervised learning with graphs using graph neural networks. Specifically, it investigates the benefits of depth both from a (learning) theory as well as an empirical standpoint.

The key contributions are:
1. Questioning the common folklore belief that deep GNNs are not expressive due to "over-smoothing"
2. Studying the generalization performance of GNNs using uniform stability theory
3. Introducing  "Decoupled GCNs" which learn linear combinations of single-layer GNNs (using powers of the Laplacian).
4. Experimental study validating the theoretical results.

**Ethical Concerns:**

Not applicable.

**Limitations And Societal Impact:**

Not applicable.

**Main Review:**

**Contribution/Originality:** The paper aims at demystifying the role of depth for the expressivity/generalization properties of GCNs/GNNs. To that, they first try to show that the number of expressible equivariant functions grows exponentially with the depth (Thm 1). They then show that deeper GCNs need more iterations to converge while achieving the same training error (Thm 2, *the exact setup of the statement, seems to be missing in the main text*. It is not clear to me.). They then leverage known results from learning theory, namely the "transductive uniform stability bound", to study the generalization performance of different variants of GCNs, which, as far as I know, has not been done so far.
Moreover, they introduce "decoupled GCN", see lines 324/5, which learn linear combinations of single layer GNNs (using powers of the Laplacian) and relate their generalization performance to other GCN variants.
The theory is complemented by an experimental study validating the theoretical results.

**Clarity:** Unfortunately, the paper is **not clear**, it seems to be written up in a hasty manner. For example:
In the statement of Thm 1 the meaning of "richness" is not clear, although it could have been made clear as the result is just a simple combinatorial insight. This sloppiness continues throughout the whole paper making some theorems almost non-understandable (e.g., Thm. 2).

**Significance:** The findings are significant to the GNN community.

Pros:
+ First (real) principled attempt to understand the role of depth in the generalization performance of GNNs
+ Good coverage of (relevant) related work
+ Reasonably well-executed experimental study

Cons:
+ Unclear statements and sloppiness throughout. The presentation lacks clarity overall.

Minor remarks:
There are many spelling and grammar mistakes throughout the paper that could have been found easily by using a standard spell/grammar checker.

Suggestion:
Put in the work and make everything mathematically precise and clear, while also making it easier for the reader to understand the significance of the results. I am looking forward to a revised, cleaned-up version of this paper.

**Time Spent Reviewing:**

4h

---

> ### Author Response · Authors · 2021-08-10
> **Response to Reviewer (Z7vj)**
>
> We are deeply sorry for the inconvenience that the readability of the paper caused, we will surely do our best to improve the quality of writing and address the typos and grammatical errors. It would be very appreciated if the reviewer can update the score considering the contributions, originality and significance of this paper mentioned by the reviewer.
>
> *Q1: The meaning of *Richness* in Theorem 1.*
>
> A1: Sorry for the confusion. The term *richness* refers to the number of computation graphs a model can encode (Lines 195-197), where we use the notation $|L\text{-GCN}(T^L)|$ to show this quantity in Theorem 1. We will add this note in the revised version.
>
> *Q2: Clarification on Theorem 2*
>
> A2: We apologize that the reviewer thinks the theorem is non-understandable, and we will provide more intuitive discussions to improve its readability. In the following, we try to rephrase Theorem 2 in a more reader friendly manner.
> *Given an $L$-layer GCN that is parameterized by $\boldsymbol{\theta}$, let define $\boldsymbol{\theta}_t$ as the weight parameters at the $t$-th iteration, and we define the loss $\mathcal{L}(\boldsymbol{\theta}_t)$ at the $t$-th iteration using square loss. The theorem claims that if the dimension of the last layer of GCN $d_L$ is larger than the number of data $N$ (which is required and commonly used in other neural network convergence analysis to guarantee that the model parameters do not change significantly during training), then we can guarantee the loss $\mathcal{L}(\boldsymbol{\theta}_T) \leq \varepsilon$ after $T=\mathcal{O}(1/\varepsilon)$ iterations of the gradient updates. The deeper the GCN model becomes (i.e., the larger the $L$), the more number of iterations $T$ is required.*

---

> > ### Comment · Reviewer_Z7vj · 2021-08-18
> > **Response ot authors**
> >
> > Thank you, for the clarification. I increased my score, under the author's promise of sufficiently polishing the final version of the paper.

---

### Decision · Program_Chairs · 2021-09-27

**Decision:**

Accept (Poster)

**Comment:**

This work provides theoretical and empirical evidence that over-smoothing does not necessarily happen in practice: it is shown that a deep GCN is expressive as long as properly trained, as well as that it can converge to a globally optimal solution. The paper also discusses the generalization capability of GCNs.

The reviewers and AC agree that these contributions are valuable to the GNN community and non-trivial.

The paper contained some small bugs, but these should be easily fixable in the camera-ready version.